# SEL1L–HRD1 endoplasmic reticulum-associated degradation controls STING-mediated innate immunity by limiting the size of the activable STING pool

Yewei Ji [1,2,3,11] ✉, Yuan Luo[1,3,11], Yating Wu[3], Yao Sun[3], Lianfeng Zhao[3], Zhen Xue[4], Mengqi Sun[3], Xiaoqiong Wei[2], Zinan He[3], Shuangcheng Alivia Wu[2], Liangguang Leo Lin[2], You Lu[2], Lei Chang[3], Fei Chen[3], Siyu Chen[5], Wei Qian[6], Xiaoxi Xu[7], Shengnuo Chen[3], Dongli Pan [5], Zhangsen Zhou [2,8], Sheng Xia[6], Chih-Chi Andrew Hu [9], Tingbo Liang [1,3] & Ling Qi [2,7,10] ✉

Stimulator of interferon genes (STING) orchestrates the production of proinflammatory cytokines in response to cytosolic double-stranded DNA; however, the pathophysiological significance and molecular mechanism underlying the folding and maturation of nascent STING protein at the endoplasmic reticulum (ER) remain unknown. Here we report that the SEL1L–HRD1 protein complex—the most conserved branch of ER-associated degradation (ERAD)—is a negative regulator of the STING innate immunity by ubiquitinating and targeting nascent STING protein for proteasomal degradation in the basal state. *SEL1L* or *HRD1* deficiency in macrophages specifically amplifies STING signalling and immunity against viral infection and tumour growth. Mechanistically, nascent STING protein is a bona fide substrate of SEL1L–HRD1 in the basal state, uncoupled from ER stress or its sensor inositol-requiring enzyme 1α. Hence, our study not only establishes a key role of SEL1L–HRD1 ERAD in innate immunity by limiting the size of the activable STING pool, but identifies a regulatory mechanism and therapeutic approach to targeting STING.

The stimulator of interferon genes (STING) signalling cascade plays an essential role in orchestrating innate immunity against pathogenic double-stranded DNA (dsDNA) and autoimmunity[1–3]. Pathogen-derived cytosolic dsDNA is recognized by cyclic GMP–AMP synthase (cGAS), which converts ATP and GTP to cyclic GMP–AMP (cGAMP)[4]. cGAMP then binds to a four-span transmembrane protein known as STING on the endoplasmic reticulum (ER), triggering its conformational change and activation[1,5]. Activated STING exits the ER and translocates to the *trans*-Golgi network[6,7], where STING recruits and activates downstream kinase TANK-binding kinase 1

(TBK1) and the transcription factor interferon regulatory factor 3 (IRF3), leading to the induction of key inflammatory cytokine genes involved in innate immunity, such as type I interferon (IFN)[8,9]. In addition to pathogen-derived dsDNA, cytosolic self-DNA, originated from either damaged mitochondrial or unstable genome, can also lead to STING activation and the onset of autoimmune diseases in various pathologies, such as Aicardi–Goutieres syndrome, systemic lupus erythematosus and other type I interferonopathies[10,11]. Indeed, constitutively active STING mutations have been identified in patients with STING-associated vasculopathy with onset in infancy and lupus-like

symptoms[3,12]. Thus, STING activity needs to be tightly regulated to maintain immune homeostasis.

Recent studies have shown that activated STING is negatively regulated in the post-ER compartments by proteasomal- or lysosomal-mediated degradation[6,13–15]. Two E3 ubiquitin ligases—ring finger protein 5 (RNF5, also known as RMA1) and tripartite motif-containing protein 30α (TRIM30α)—may be involved in the degradation of activated STING, serving as a negative feedback regulatory mechanism to attenuate STING-mediated response following viral infection[13,14]. In addition, activated STING can be sorted into acidic endolysosomes for degradation. This membrane trafficking process may involve adaptor protein complex 1-mediated delivery from the Golgi apparatus to endolysosomes via clathrin-coated transport vesicles, the lysosomal membrane protein Niemann–Pick type C1 or p62/SQSTM1-dependent autophagy[6,15–18]. However, how STING is regulated under basal (resting) conditions remains largely unclear. Recent studies have shown that STING may interact with $Ca^{2+}$ sensor stromal interaction molecule 1 (STIM1)[19] or Toll-interacting protein[20] in the basal state, which prevents either its activation or degradation. Intriguingly, in the latter case, lysosomal degradation of STING requires the activity of inositol-requiring enzyme 1α (IRE1α), an ER-resident sensor of ER stress or unfolded protein response (UPR), although the mechanism remains unclear[20].

The observations that cGAMP binding triggering conformational change of STING on the ER is a prerequisite of its ER exit[1,2,5] and that constitutively active STING disease mutation products in STING-associated vasculopathy with onset in infancy readily exit the ER[21,22] point to the importance of ER retention in STING activation. However, molecular events occurring at the ER remain largely unexplored. ER-associated degradation (ERAD) is required for the proteasomal degradation of misfolded proteins in the ER[23–25]. The suppressor of lin-12-like (SEL1L)–HMG-CoA reductase degradation 1 (HRD1) complex is the most conserved branch of ERAD from yeast to humans[26–29]. Using various global, inducible or cell type-specific *Sel1L*- or *Hrd1*-deficient mouse models, we and others have revealed the vital importance of this protein complex in vivo and in many cell types[30–47]. Moreover, in mature B cells, a recent study showed that activated STING engages SEL1L–HRD1 ERAD to degrade the B cell receptor[48]; however, the role of SEL1L–HRD1 ERAD in STING biology remains unexplored.

In this article, we report that nascent STING interacts with and is ubiquitinated by SEL1L–HRD1 ERAD in the ER—an event that precedes ligand binding and regulates STING signalling potential. This SEL1L–HRD1 ERAD–STING axis in myeloid cells plays an important role in innate immunity against DNA viruses and tumorigenesis in a transplant model of pancreatic cancer. In contrast, our data show that SEL1L–HRD1 ERAD plays no role in TLR4 signalling and that ER stress, IRE1α and autophagy are all dispensable for STING signalling.

## Results

### Generation of myeloid cell-specific *Sel1L*-deficient mice

To investigate the role of SEL1L–HRD1 ERAD in innate immune signalling, we crossed *Sel1L*[flox/flox] (*Sel1L*[f/f]) mice[31] with *Lyz2*-Cre mice to generate myeloid cell-specific *Sel1L*-deficient mice (*Sel1L*[Lyz2]) (Extended Data Fig. 1a). Western blot analysis revealed a significant decrease of SEL1L protein in bone marrow-derived macrophages and thioglycollate-elicited peritoneal macrophages of *Sel1L*[Lyz2] mice compared with *Sel1L*[f/f] littermates, but not in other tissues such as the liver (Extended Data Fig. 1b). In line with the notion that SEL1L is required for HRD1 protein stability[31], HRD1 protein levels were significantly decreased in *Sel1L*[Lyz2] versus *Sel1L*[f/f] macrophages (Fig. 1a).

*Sel1L*[Lyz2] mice appeared normal compared with *Sel1L*[f/f] littermates (Fig. 1b). The percentages of splenic macrophages, CD11b[+]Gr1[+] neutrophils, B220[+] B cells and CD4[+] and CD8[+] T cells were comparable (Fig. 1c and Extended Data Fig. 1c,d). To demonstrate the functional consequence of SEL1L–HRD1 ERAD deficiency, we next assessed the status of ER homeostasis. Transmission electron microscopy (TEM)

analysis revealed dilated balloon-like ER morphology in *Sel1L*[Lyz2] macrophages versus sheet-like structures in *Sel1L*[f/f] mice (arrows in Fig. 1d). Chaperones such as BiP, PDI and ERP44 were mildly elevated in *Sel1L*[Lyz2] macrophages (Fig. 1a and Extended Data Fig. 1f). Consistent with the notion that IRE1α of the UPR sensor is an ERAD substrate[32], IRE1α protein was increased sevenfold in *Sel1L*[Lyz2] macrophages (Fig. 1a). However, Phos-tag-based western blot[49,50] failed to detect hyperphosphorylation of IRE1α in *Sel1L*[Lyz2] macrophages compared with that in *Sel1L*[f/f] macrophages under basal conditions (lane 6 versus lane 1 in Fig. 1e). Moreover, cell death was comparable between the two cohorts, as measured by levels of cleaved caspase-3 (lane 1 versus lane 7 in Fig. 1f) and annexin V staining (Extended Data Fig. 1e). Taken together, these data indicate that *SEL1L* deficiency is well tolerated by macrophages in vivo under basal conditions, as demonstrated by a subtle UPR and the lack of any detectable changes in immune cell composition and survival.

### Intact lipopolysaccharide response in *Sel1L*[Lyz2] mice

As previous studies have implicated the UPR pathway[51–53] and HRD1 in Toll-like receptor (TLR) signalling[54], we examined the lipopolysaccharide (LPS) response in *Sel1L*[Lyz2] macrophages. LPS treatment failed to enhance IRE1α phosphorylation in both *Sel1L*[f/f] and *Sel1L*[Lyz2] macrophages (Fig. 1e). Moreover, LPS-stimulated inflammation and cell death were comparable between *Sel1L*[f/f] and *Sel1L*[Lyz2] macrophages in vitro (Fig. 1f,g and Extended Data Fig. 1g). In vivo, LPS injection, which induces endotoxic shock through macrophages[55,56], increased circulating tumour necrosis factor α (TNFα) and interleukin-6 (IL-6) (Fig. 1h) and lethality at similar rates for both cohorts (Fig. 1i). Thus, our data demonstrate that SEL1L–HRD1 ERAD in macrophages plays no role in LPS response.

### Intact major histocompatibility complex antigen presentation in *Sel1L*[Lyz2] macrophages

Macrophages are professional antigen-presenting cells that process and present peptide and lipid antigens in complex with major histocompatibility complex (MHC) class I/II and CD1d proteins to activate CD8[+]/CD4[+] T and natural killer T (NKT) cells, respectively[57]. Biosynthesis, folding and assembly of MHC class I/II and CD1d protein complexes all occur in the ER, and orphan or mutated MHC class I heavy chains are reportedly SEL1L–HRD1 substrates[58]. Unexpectedly, surface MHC class I and II levels were unaffected by *Sel1L* deficiency in macrophages (Extended Data Fig. 1h). Moreover, there was no difference in the activation of OT-1 T cell receptor transgenic CD8[+] T cells following a coculture with primary macrophages loaded with ovalbumin peptide SIINFEKL, as measured by secreted IL-2 levels (Extended Data Fig. 1i). Similar results were obtained in the activation of the NKT cell line DN32.D3 when cocultured with primary macrophages pulsed with the CD1d lipid ligand α-galactoceramide (Extended Data Fig. 1j). Thus, we conclude that SEL1L–HRD1 ERAD is dispensable for the maturation and antigen presentation function of MHC complexes.

### Macrophage ERAD is dispensable for diet-induced obesity

Given that macrophage infiltration into white adipose tissue (WAT) may be important for the development of inflammatory tone in obesity and type 2 diabetes[59–61], we next explored the role of myeloid SEL1L–HRD1 ERAD in the pathogenis of diet-induced obesity. Following 60% high-fat diet (HFD) feeding for 20 weeks, there was no difference in body and tissue weights between *Sel1L*[f/f] and *Sel1L*[Lyz2] mice (Extended Data Fig. 2a–c). Histologically, no difference was observed in the liver and WAT between the two (Extended Data Fig. 2d). The expression levels of most M1 and M2 macrophage markers (that is, *Arg1*, *Pcd1lg2*, *Mrc1* and *Retn1a*) were comparable in WATs (Extended Data Fig. 2e), as were serum cytokine levels of IL-6 and TNFα (Extended Data Fig. 2f). Flow cytometric analysis of immune cell compositions in WAT showed comparable numbers and percentages of NKT, CD4[+] T and B220[+] cells

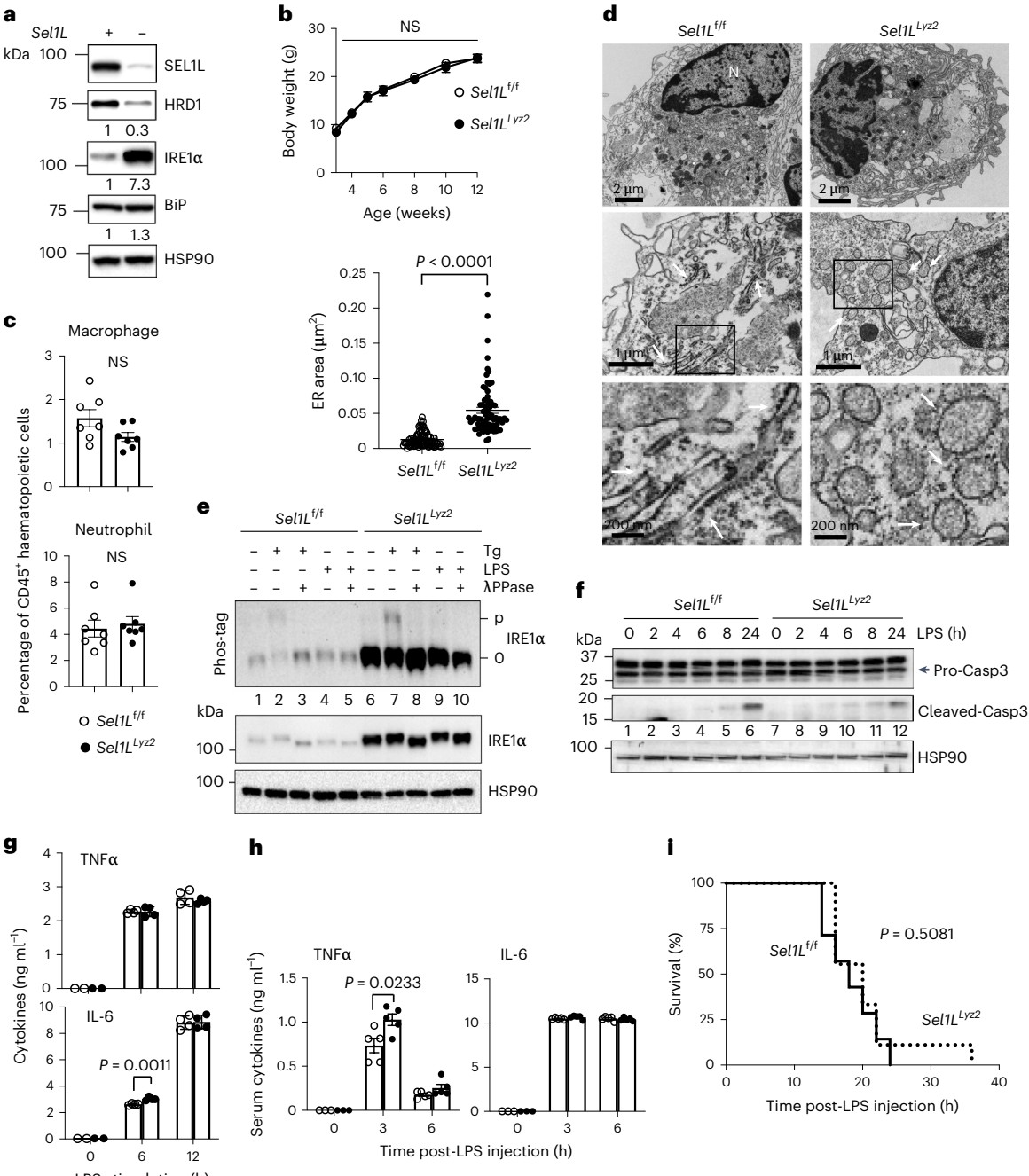

**Fig. 1 | Normal growth and intact TLR4 innate immune response in *Sel1L^Lyz2* mice. a**, Immunoblot in primary macrophages, with the relative band intensity (normalized to HSP90) shown below each gel, representative of three independent biological repeats. Each lane shows the results of pooled macrophages from three mice. **b**, Growth curves for male littermates (*n* = 7 mice each). NS, not significant. **c**, Quantitation of flow cytometric analysis for F4/80⁺CD11b⁺ macrophages and CD11b⁺Gr1⁺ neutrophils in spleens (*n* = 7 mice each from two independent repeats). Original data are shown in Extended Data Fig. 1c. **d**, Representative TEM images showing the ultrastructure of peritoneal macrophages. Quantitation of ER area is shown on the left (*n* = 93 ER areas from 14 *Sel1L^f/f* macrophages and 74 ER areas from 13 *Sel1L^Lyz2* macrophages, pooled from two mice per genotype). The arrows point to ER. N, nucleus. **e**, Immunoblot of IRE1α protein levels and phosphorylation (phos-tag gel) in macrophages treated with vehicle, thapsigargin (Tg; 300 nM) or LPS (1,000 ng ml⁻¹) for 4 h.

Protein lysate treated with λPPase was included as a control. p, phosphorylated; 0, non-phosphorylated. **f**, Immunoblot of pro- and cleaved caspase-3 in primary macrophages treated with LPS (1 μg ml⁻¹) for the indicated times. The results in **e** and **f** are representative of two independent biological repeats. **g**, ELISA analysis of TNFα and IL-6 in the culture supernatants of LPS-treated macrophages at different time points (*n* = 2 each for 0 h (a statistical test was not used for this time point) and *n* = 4 each for 6 and 12 h). **h**, ELISA analysis of the serum cytokines TNFα and IL-6 in mice at different time points after LPS injection (*n* = 3 mice each for 0 h and *n* = 5 mice each for 3 and 6 h; combined from two independent repeats). **i**, Survival curves post-LPS injection (*n* = 7 and 9 for *Sel1L^f/f* and *Sel1L^Lyz2* mice, respectively). The results are representative of two independent repeats and source data are provided for all repeats. All values represent means ± s.e.m. Statistical significance was determined by unpaired, two-tailed Student's *t*-test (**b**–**d**, **g** and **h**) and log-rank (Mantel–Cox) test (**i**).

and macrophages between the cohorts, whereas the percentages of CD8[+] T cells were elevated in *Sel1L^Lyz2* mice (Extended Data Fig. 2g,h). Moreover, metabolic parameters such as fasting glucose and serum insulin and glucose and insulin tolerance were all comparable between the two cohorts (Extended Data Fig. 2i–k). Taken together, we conclude that myeloid SEL1L–HRD1 ERAD is dispensable for WAT inflammation and insulin resistance in diet-induced obesity.

### *Sel1L* deficiency augments cGAS–STING signalling

Next, we screened a number of innate immunity agonists, including Pam3Cys–Ser–Lys (Pam3) for TLR2, polyinosinic:polycytidylic acid (poly(I:C)) for retinoic acid-inducible gene I (RIG-1) and cGAMP for STING, to determine their ability to stimulate inflammation in primary macrophages. Surprisingly, among these agonists, only cGAMP stimulation consistently triggered significantly higher expression of several inflammatory cytokine genes (*Tnfa*, *Il6*, *Ifnb* and *Cxcl10*) in *Sel1L^Lyz2* compared with *Sel1L^f/f* macrophages (Fig. 2a). Similar results were obtained with another STING agonist, cyclic diadenylate (c-di-AMP) (Extended Data Fig. 3a). In keeping with these findings, protein levels for secreted IFNβ and TNFα in stimulated *Sel1L^Lyz2* macrophages were significantly higher than those of *Sel1L^f/f* macrophages (Fig. 2b). Moreover, while protein levels of the STING downstream effectors TBK1 and IRF3 were comparable, both were hyper-phosphorylated in *Sel1L^Lyz2* macrophages compared with *Sel1L^f/f* macrophages upon cGAMP treatment (Fig. 2c). Similar observations were obtained using another STING agonist, 5,6-dimethylxanthenone-4-acetic acid (DMXAA) (Fig. 2d). In direct contrast, treatment with the RIG-1 agonist double-stranded RNA poly(I:C) triggered a similar response between *Sel1L^Lyz2* and *Sel1L^f/f* macrophages, including the secretion of IFNβ and TNFα, and phosphorylation of both TBK1 and IRF3 (Fig. 2e,f). As RIG-1 and STING pathways converge on TBK1 and IRF3 (refs. 9,62), these data suggest that SEL1L–HRD1 ERAD may specifically regulate an early event (or multiple early events) in the STING pathway.

### Stabilization of STING protein in the absence of ERAD

Considering that STING is an ER transmembrane protein[2], we then tested whether SEL1L–HRD1 ERAD may directly regulate the abundance of STING protein. Indeed, the STING protein level was increased nearly threefold in *Sel1L^Lyz2* versus *Sel1L^f/f* macrophages (Fig. 3a). In contrast, other proteins in its pathway, such as the dsDNA sensor cGAS, the STING interactor STIM1 (ref. 19) and the downstream effectors TBK and IRF3, were unchanged (Fig. 3a and Extended Data Fig. 3b). Moreover, the levels of other ER-resident membrane proteins, such as the ER chaperone CALNEXIN, the metabolic enzymes acyl-CoA synthetase 4 (FACL4) and sterol *O*-acyltransferase (SOAT1) and the immune receptors TLR2 and TLR4, as well as programmed death-ligand 1 (PD-L1), MHC I (H-2Kb/H-2Db) and MHC II (I-A/I-E), were all unaffected by *Sel1L* deficiency (Fig. 3a and Extended Data Fig. 3b,c). These data pointed to a STING-specific effect of SEL1L–HRD1 ERAD in macrophages. This effect was regulated at a post-transcriptional level because *Sting* transcript was unchanged in *Sel1L^Lyz2* macrophages (Fig. 3b). Pretreatment with the STING inhibitor H151 reversed the ligand-dependent hyper-activation of the STING pathway in *Sel1L^Lyz2* macrophages (Extended Data Fig. 3d,e). To further test the role of HRD1 in the regulation of STING function, we generated the *Hrd1*-deficient macrophage cell RAW 264.7 using the CRISPR–Cas9 system. Similar to *Sel1L*-deficient macrophages, deletion of *Hrd1* led to a nearly twofold accumulation of STING protein (Fig. 3c) and enhanced the phosphorylation of both STING and TBK1 upon agonist stimulation (Fig. 3d). Hence, SEL1L–HRD1 ERAD regulates cGAS/STING signalling via STING.

In translation shut-off assay with cycloheximide (CHX), STING protein levels decreased by over 30% in 6 h in *Sel1L^f/f* macrophages (lane 1 versus lane 3), but became stabilized in *Sel1L^Lyz2* macrophages (lane 5 versus lane 7) (Fig. 3e). A similar observation was obtained in *Hrd1*-deficient RAW 264.7 cells (Extended Data Fig. 4a,b). Treatment with the proteasomal inhibitor MG132 blocked the decay of STING protein in *Sel1L^f/f* macrophages (lane 3 versus lane 4 in Fig. 3e), pointing to the involvement of proteasomes in STING protein turnover. The SEL1L–HRD1 ERAD effect on STING was not limited to macrophages as STING protein was stabilized in *Hrd1^−/−* mouse embryonic fibroblasts (MEFs) as well in the basal state (Fig. 3f). Hence, the SEL1L–HRD1 ERAD–STING axis may represent a general mechanism.

To further demonstrate the importance of SEL1L–HRD1 ERAD in STING biology, we generated myeloid-specific *Atg7* knockout (*Atg7^Lyz2*) mice with defects in macroautophagy—another major intracellular proteolytic pathway[63]. Surprisingly, there were no differences in STING protein levels or its signalling pathway (Fig. 3g), nor gene expression or secretion of IFNβ (Fig. 3h,i) in *Atg7^Lyz2* versus *Atg7^f/f* macrophages in both basal and active states, pointing to a dispensable role of macroautophagy in STING activation. However, in line with previous reports[6,15], treatment with bafilomycin A1 (a compound that inhibits lysosomal acidification and degradation[64,65]) increased STING protein levels in DMXAA-stimulated cells while having no effect in the basal state (Fig. 3j). Taken together, we conclude that unlike active STING, which is degraded in the endolysosomes independent of macroautophagy, degradation of STING in the basal state occurs in the ER and is mediated by SEL1L–HRD1 ERAD.

### The effect of ERAD on STING is uncoupled from UPR and IRE1α

As ERAD and UPR are two tightly linked pathways[24], we next tested the interplay between ER stress/UPR and STING, as previously suggested[66–68]. Surprisingly, cGAMP treatment of wild-type macrophages failed to induce the expression of common UPR markers such as *Xbp1*, *Grp78*, *Grp94* and *Chop*, while the ER stressor thapsigargin had no impact on the expression of inflammatory genes such as *Sting*, *Ifnb* and *Cxcl10* (Fig. 4a). Moreover, thapsigargin treatment failed to stimulate IRF3 phosphorylation in macrophages (Fig. 4b). Hence, UPR is not sufficient to activate the STING pathway in macrophages, and vice versa.

Next, we asked whether UPR links ERAD dysfunction to STING activation. Thapsigargin treatment in *Sel1L^Lyz2* macrophages had no additional effect on STING protein levels or its downstream signalling (Fig. 4c). Pretreatment of macrophages with the chemical chaperone tauroursodeoxycholic acid, which partially decreased the expression of UPR genes (Extended Data Fig. 5a), failed to alter DMXAA-induced phosphorylation of STING and TBK1 (lane 8 versus lane 6 in Extended Data Fig. 5b) or the expression of the genes *Ifnb*, *Cxcl10* and *Il6* in *Sel1L^Lyz2* macrophages (Extended Data Fig. 5c). Hence, these data exclude a major role of UPR in *Sel1L* deficiency-induced hyper-responsive STING.

As the UPR sensor IRE1α is highly enriched in *Sel1L^Lyz2* macrophages (Fig. 1) and as IRE1α has been implicated in STING protein levels and its signalling[20], we next asked whether it may link SEL1L–HRD1 ERAD to STING. We treated primary macrophages with an IRE1α-specific inhibitor, 4μ8c[69]. While it reduced *Xbp1* messenger RNA splicing (Fig. 4d), 4μ8c treatment did not rescue the difference of DMXAA-induced STING activation in *Sel1L^Lyz2* versus *Sel1L^f/f* macrophages, as measured by the expression of inflammatory genes (Fig. 4e) and phosphorylation of STING and TBK1 (Fig. 4f). Moreover, genetic deletion of IRE1α alone in macrophages had no effect on STING protein levels (Fig. 4g), and a decrease of IRE1α protein levels in *Hrd1^−/−* RAW 264.7 cells to a level similar to that of wild-type cells did not reverse STING protein accumulation (Fig. 4h). Taken together, these data demonstrate that the UPR and IRE1α have no effect on STING signalling and that the SEL1L–HRD1 ERAD effect on the STING pathway is UPR and IRE1α independent.

### STING is an endogenous substrate of SEL1L–HRD1 ERAD

Next, we tested how SEL1L–HRD1 ERAD regulates STING protein stability. As STING physically interacted with SEL1L and HRD1 in B cells[48], we first tested whether SEL1L–HRD1 ERAD interacts with STING in

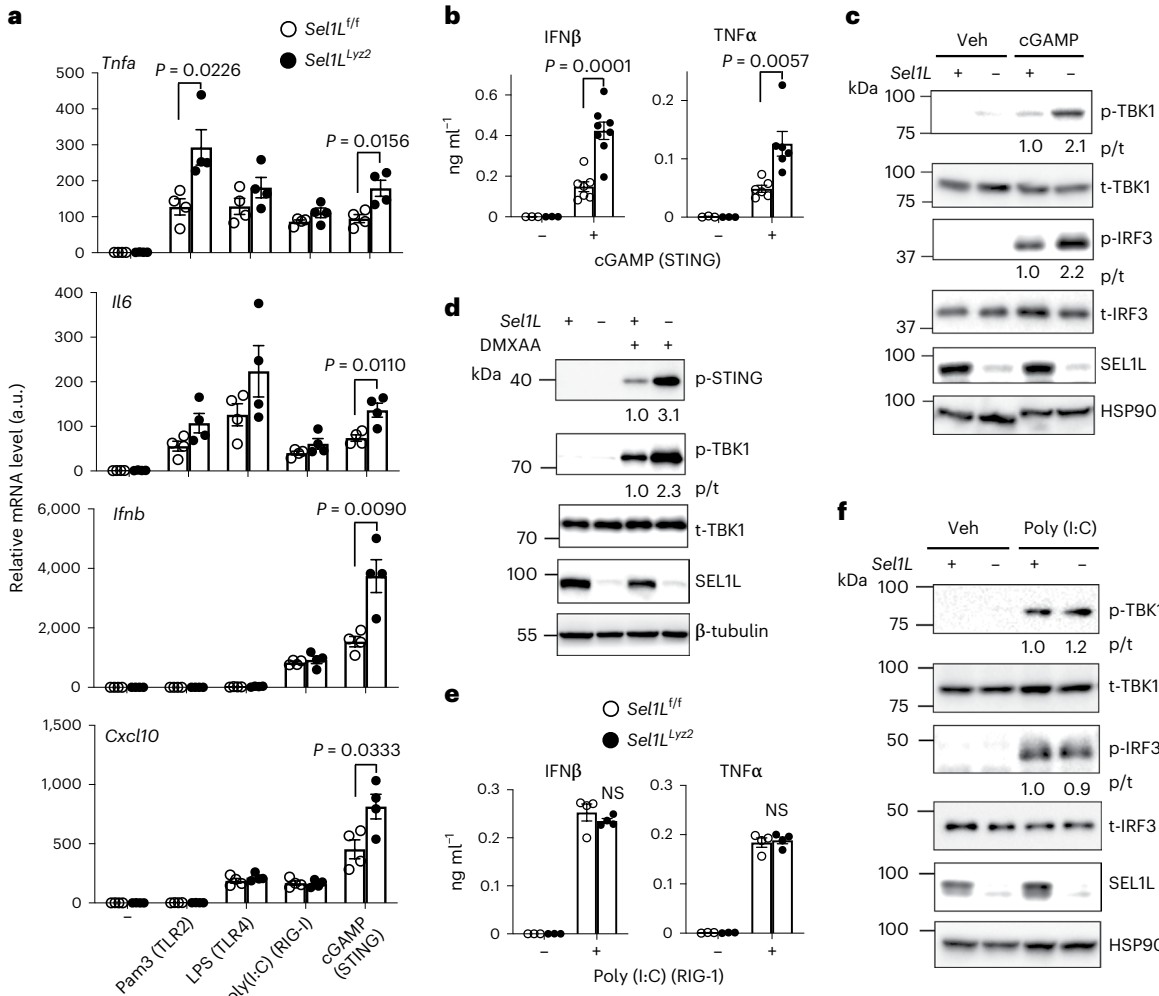

**Fig. 2 | Loss of SEL1L specifically enhances the STING signalling cascade.**
**a**, qPCR analysis in primary macrophages treated with Pam3 (TLR2) or LPS (TLR4) or transfected with double-stranded RNA (RIG-I) or cGAMP (STING) ($n = 4$ mice each pooled from two independent repeats). mRNA, messenger RNA. **b**, ELISA analysis of secreted IFNβ and TNFα in primary macrophages treated with cGAMP for 6 h ($n = 3, 3, 7$ and 8 mice (left to right), pooled from three independent repeats. **c**, Immunoblot analysis in primary macrophages transfected with vehicle (Veh) or cGAMP for 3 h, representative of four independent repeats. **d**, Immunoblot analysis in primary macrophages treated with vehicle or DMXAA

for 1.5 h, representative of four independent repeats. **e**, ELISA analysis of secreted IFNβ and TNFα in primary macrophages transfected with poly(I:C) (RIG-1) for 18 h ($n = 3, 3, 4$ and 4 mice (left to right); combined from two independent repeats). **f**, Immunoblot analysis in primary macrophages transfected with vehicle or poly(I:C) for 6 h, representative of two independent repeats. The relative intensity of p-STING (normalized to β-tubulin) or ratio of phosphorylated to total protein (p/t) are shown below the blots (**c**, **d** and **f**). All values represent means ± s.e.m. Statistical significance was determined by unpaired, two-tailed Student's $t$-test (**a**, **b** and **e**).

macrophages. Based on a STING proximity labelling-based proteomics study[6], STING protein interacted with many ER chaperones (for example, CANX and HSP90B1), oxidoreductases (for example, PDIA3 and P4HB), glycosylation regulators (for example, LMAN1, RPN1 and UGGT1) and, importantly, ERAD factors such as SEL1L and p97/VCP (Fig. 5a). Indeed, endogenous SEL1L physically interacted with endogenous ERAD factors (for example, HRD1, OS9 and CALNEXIN) and STING, but not STIM1 or TBK1, in wild-type macrophages (Fig. 5b). The interaction of STING with SEL1L–HRD1, but not phospho-TBK1, was attenuated upon DMXAA stimulation (Fig. 5c). Next, we tagged the carboxy-terminal SEL1L or STING protein with TurboID, a biotin ligase that can covalently biotinylate interacting proteins in very close proximity[70] (Extended Data Fig. 6a), and transfected them into MEF or RAW cells. SEL1L-TurboID biotinylated endogenous STING, CALNEXIN and HRD1 (Fig. 5d) in the basal state, and its interaction with STING rapidly decreased upon cGAMP stimulation whereas its interactions with HRD1 and CALNEXIN persisted (Fig. 5d). Consistently, STING-TurboID biotinylated endogenous SEL1L in the basal state, and its interaction

with SEL1L rapidly decreased upon cGAMP stimulation, whereas its interaction with phospho-TBK1 increased with time (Fig. 5e).

We then tested how STING interacts with and is ubiquitinated by SEL1L–HRD1 ERAD. Both HRD1 and STING are multi-pass transmembrane proteins with large cytosolic domains (Fig. 5f). In HEK293T cells, immunoprecipitation of transfected STING was able to pull down both endogenous SEL1L and HRD1, but not IRE1α (Extended Data Fig. 6b). Immunoprecipitation of truncated STING or HRD1 proteins showed that they interacted with each other via the cytosolic domains (Fig. 5g and Extended Data Fig. 6c). Moreover, HRD1 robustly ubiquitinated STING protein (Fig. 5h) and MG132 treatment for 5 h led to a 39% increase of STING protein in wild-type macrophages (Fig. 5h and Extended Data Fig. 6d), suggesting that SEL1L–HRD1 ERAD degrades a subset of nascent STING proteins in the ER. HRD1-mediated ubiquitination of STING required its catalytic really interesting new gene (RING) domain, as demonstrated by the decrease in polyubiquitination of STING in cells expressing the HRD1 RING ligase-dead C2A variant[71] (Fig. 5i).

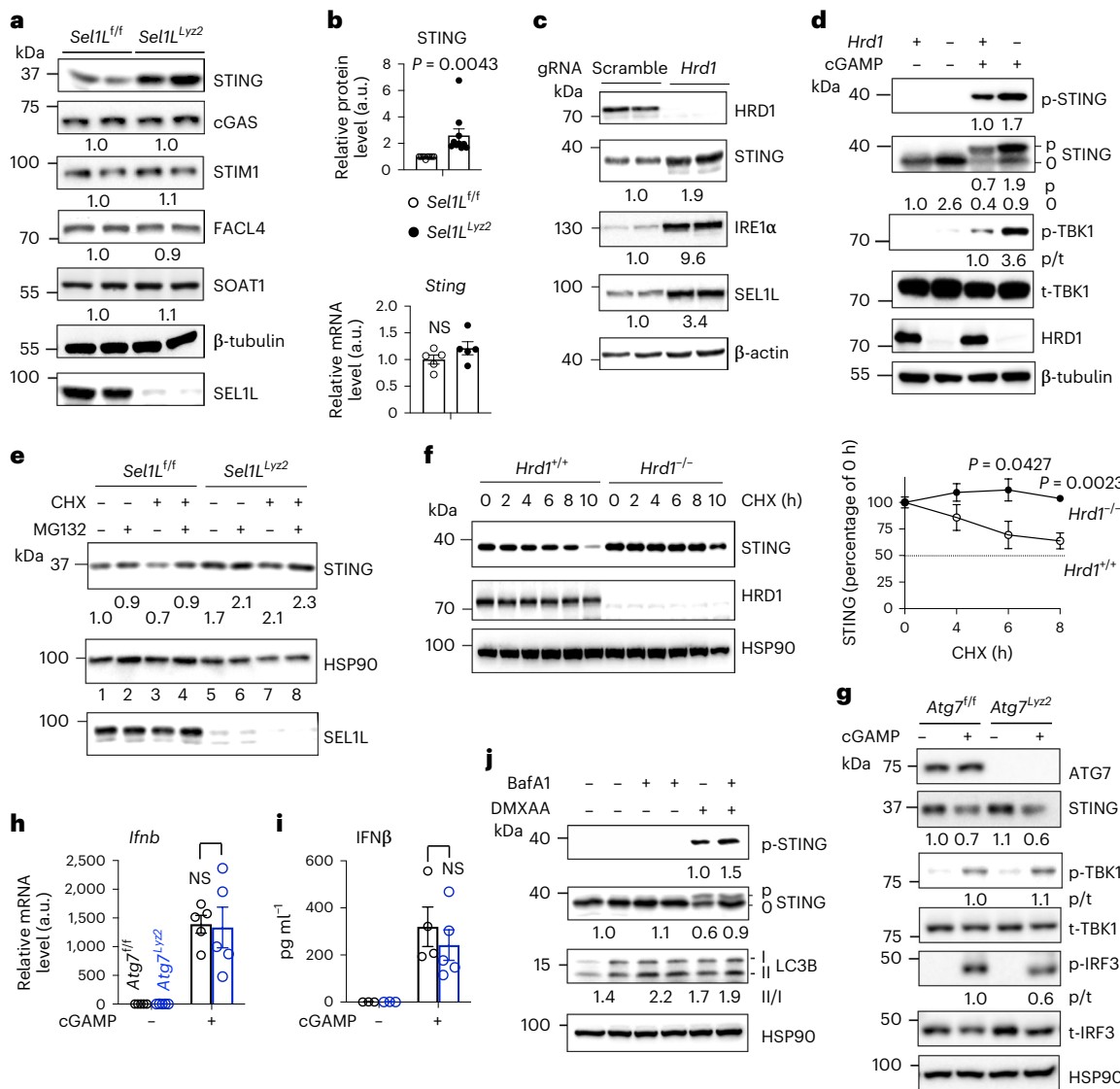

**Fig. 3 | Accumulation of STING protein in the absence of SEL1L–HRD1 ERAD in the basal state. a**, Immunoblot analysis in primary macrophages, representative of more than two independent repeats. Quantitation of STING protein levels is shown to the right ($n = 10$ each); combined from nine independent repeats. **b**, qPCR analysis of the *Sting* messenger RNA level normalized to the ribosomal gene *L32* ($n = 5$ mice each; combined from two independent repeats). **c**, Immunoblot analysis in wild-type (scramble) versus *Hrd1*⁻/⁻ RAW 264.7 cells, representative of three independent repeats. Each lane shows the results for a different *Hrd1*⁻/⁻ line. gRNA, guide RNA. **d**, Immunoblot analysis in RAW 264.7 cells treated with vehicle or cGAMP for 6 h, representative of three independent repeats. **e**, Immunoblot analysis in primary macrophages treated with vehicle, 25 μM MG132 and/or 50 μg ml⁻¹ CHX for 6 h, representative of three independent repeats. **f**, Immunoblot analysis in *Hrd1*⁺/⁺ and *Hrd1*⁻/⁻ MEFs treated with CHX for the indicated times,

with quantitation from four independent experiments shown on the right. **g**, Immunoblot analysis in primary macrophages treated with vehicle or cGAMP for 6 h, representative of two independent repeats. **h,i**, qPCR analysis of *Infb* (**h**) and ELISA analysis of secreted IFNβ (**i**) in primary macrophages treated with vehicle or cGAMP for 6 h ($n = 5$ mice each; combined from two independent repeats). **j**, Immunoblot analysis in primary macrophages treated with or without the autophagy inhibitor bafilomycin A1 (BafA1) for 6 h, DMXAA for 3 h or DMXAA and BafA1 for 3 h, representative of two independent repeats. In **a**, **c**–**e**, **g** and **j**, the quantitation of total protein levels (normalized to the loading control) or ratio of phosphorylated to total protein (p/t) is shown below the blot. All values represent means ± s.e.m. Statistical significance was determined by unpaired, two-tailed Student's *t*-test (**a**, **b**, **f**, **h** and **i**).

We then mapped the possible ubiquitination sites on STING. The E3 ligases RNF5 and TRIM30α reportedly target lysine (K) 150 or 275 of STING for ubiquitination and degradation following viral infection[13,14]. Surprisingly, STING protein with all eight cytosolic K residues mutated to arginine (R) (that is, K⁻) was still ubiquitinated by HRD1 (Fig. 5j and Extended Data Fig. 6e). As other amino acids, such as cysteine (C), serine (S) and threonine (T), are also potential ubiquitination sites by particular E3 ligases[72–74], we next replaced all nine cytosolic C residues with alanine (A) (that is, C⁻) and all 33 cytosolic S/T residues with A (that is, S/T⁻). Both STING variants were ubiquitinated by HRD1 (Fig. 5j and Extended Data Fig. 6e). However, when all cytosolic K, S, T

and C residues were replaced by A (that is, K/S/T⁻ or K/S/T/C⁻), STING ubiquitination was significantly decreased to a level similar to that in cells expressing the HRD1-dead C2A variant (Fig. 5j and Extended Data Fig. 6e). Thus, our data suggest that ubiquitination of STING by HRD1 in the basal state may occur on multiple amino acids including K, S, T and C, similar to the ubiquitination pattern of non-secreted immunoglobulin light chain by HRD1 (ref. 72). We then defined the polyubiquitin chain topology in STING ubiquitination. In contrast with K48-linked polyubiquitination of STING by the E3 ligases RNF5 and TRIM30α (refs. 13,14), HRD1-mediated ubiquitination of STING was mainly K27-linked ubiquitination (Extended Data Fig. 6f),

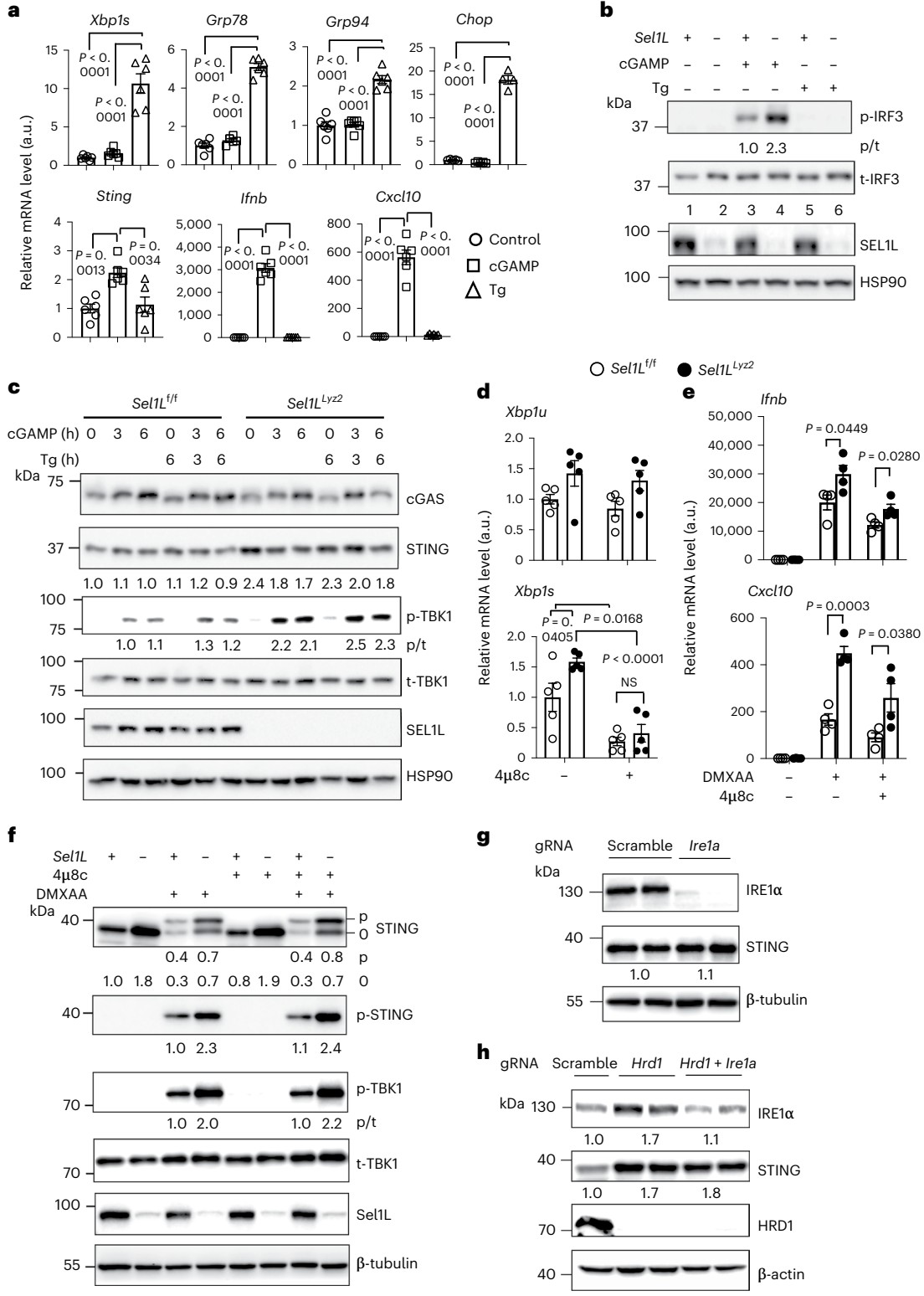

**Fig. 4 | The effect of SEL1L–HRD1 ERAD on STING is uncoupled from UPR and IRE1α. a**, qPCR analysis of ER stress (top) and inflammatory genes (bottom) in wild-type macrophages treated with vehicle (control), cGAMP or thapsigargin for 6 h (n = 6 mice each; combined from two independent repeats). **b**, Immunoblot analysis in primary macrophages treated as in **a**. **c**, Immunoblot analysis in primary macrophages treated with cGAMP and/or thapsigargin for the indicated times. The results in **b** and **c** are representative of two independent repeats. **d**, qPCR analysis of *Xbp1u* and *Xbp1s* in primary macrophages treated with vehicle or 4μ8c for 24 h (n = 5 mice each; combined from two independent repeats). **e**, qPCR analysis in macrophages treated with vehicle, IRE1α inhibitor 4μ8c (24 h)

and/or DMXAA (1 h) (n = 4 mice each; combined from two independent repeats). **f**, Immunoblot in primary macrophages treated as in **e**, representative of three independent repeats. **g,h**, Immunoblot analysis of STING expression in wild-type (scramble) versus *Ire1a*[−/−] RAW 264.7 cells (**g**) or wild-type versus *Hrd1*[−/−] and *Hrd1*[−/−]*Ire1a*[−/−] RAW 264.7 cells (**h**), representative of two independent repeats. In **b**, **c** and **f**–**h**, quantitation of total protein levels (normalized to the loading control) or the ratio of phosphorylated to total protein (p/t) is shown below each blot. All values represent means ± s.e.m. Statistical significance was determined by one-way ANOVA with Newman–Keuls post-test (**a**) or unpaired, two-tailed Student's *t*-test (**d** and **e**).

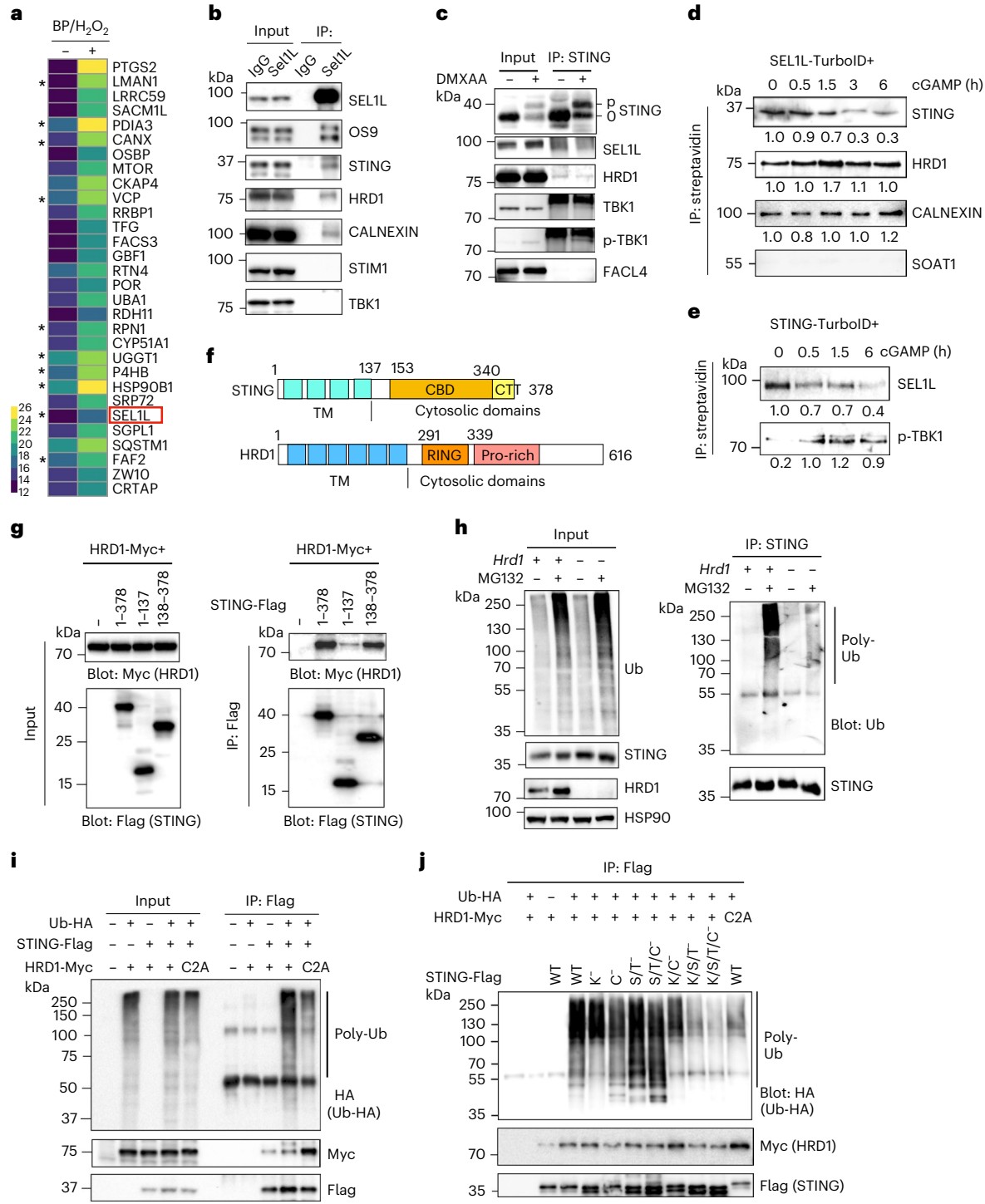

**Fig. 5 | STING directly interacts with and is ubiquitinated by SEL1L–HRD1 ERAD in the basal state. a**, Heat map showing the top 30 ER-resident STING-interacting proteins in the basal state from a published STING–APEX2 proximity labelling study[6]. In the presence of hydrogen peroxide (H₂O₂), APEX2 catalyses biotin-phenol (BP) to produce a biotin-phenoxyl intermediate with which to label proximal proteins (+BP/H₂O₂). The values shown are log₂ of the original mass spectrometry values. Asterisks highlight proteins involved in folding and degradation in the ER. **b,c**, Immunoblot analysis following immunoprecipitation of endogenous SEL1L (**b**) and STING (**c**) in primary macrophages in the basal state (**b**) or treated with or without DMXAA for 3 h (**c**), representative of three independent biological repeats. IgG, immunoglobulin G; IP, immunoprecipitation. **d,e**, Immunoblot analysis following immunoprecipitation with streptavidin beads in MEF (**d**) and RAW 264.7 cells (**e**) transfected with SEL1L-TurboID (**d**) and STING-TurboID (**e**) followed by cGAMP treatment for the indicated times, representative of three independent biological repeats.

Quantitation of total protein levels (normalized to 0 h) is shown below each blot. **f**, Diagrams of the STING and HRD1 protein domains. CBD, c-di-GMP-binding domain; CTT, carboxy-terminal tail; Pro-rich, proline-rich; TM, transmembrane. **g**, Mapping of STING and HRD1 interacting domains. Shown are the results of immunoblot analysis following Flag immunoprecipitation in HEK293T cells transfected with various plasmids encoding full-length or truncated STING proteins, as indicated. **h**, Immunoblot analysis of polyubiquitination following immunoprecipitation of endogenous STING in wild-type and *Hrd1⁻/⁻* RAW 264.7 cells treated with or without MG132 for 5 h. Ub, ubiquitin. **i,j**, Immunoblot analyses of STING ubiquitination following STING-Flag immunoprecipitation in HEK293T cells transfected with the indicated plasmids. C⁻, cytosolic C-to-A substitution; C2A, HRD1-dead variant; K⁻, cytosolic K-to-R substitution; S/T⁻, cytosolic S- and T-to-A substitution; WT, wild type. The data in **g–j** are representative of three independent biological repeats.

in a manner similar to HRD1-mediated polyubiquitination of CREBH (ref. [75]). Taken together, we conclude that STING interacts with and is ubiquitinated by HRD1.

### SEL1L–HRD1 ERAD controls the size of activable STING

Next, we investigated the importance of SEL1L–HRD1 ERAD in STING maturation and function. Unlike two previously reported ERAD substrates, pro-arginine vasopressin and proopiomelanocortin, which form detergent (NP-40)-insoluble protein aggregates in the absence of SEL1L–HRD1 ERAD[36,38], accumulated STING protein in *Sel1L*[Lyz2] macrophages remained largely soluble and undetectable in the insoluble pellets in both the basal and the active state (Fig. 6a). Sucrose density gradient fractionation showed that the distribution of STING-containing protein complexes was quite similar between the two cohorts in both the basal and the active state (Fig. 6b and Extended Data Fig. 7a). Furthermore, confocal microscopic analysis showed that in the basal state there were significantly more STING proteins in the ER of *Sel1L*[Lyz2] macrophages compared with in *Sel1L*[f/f] macrophages (Fig. 6c). Very few STING proteins were detected in the lysosomes of either cohort in the basal state (Fig. 6d). However, upon cGAMP activation, significantly more phospho-STING (p-STING) foci formed in the extra-ER compartments in *Sel1L*[Lyz2] macrophages (arrows in Fig. 6e). These data point to an expanded pool of activable STING protein in the absence of SEL1L–HRD1 ERAD.

Next, we addressed whether intracellular trafficking of STING to the extra-ER compartments is altered in the absence of SEL1L–HRD1 ERAD. We performed confocal microscopic imaging of STING at different organelles, including the *trans*-Golgi network (TGN38), endosomes (CD63) and lysosomes (lysosomal-associated membrane protein 1 (LAMP1)), in macrophages treated with or without cGAMP for the indicated times (Extended Data Fig. 7b–d). STING reached the *trans*-Golgi network, endosomes and lysosomes at 0.5, 1.5 and 6 h post-cGAMP stimulation, respectively, at comparable levels and dynamics in *Sel1L*[f/f] versus *Sel1L*[Lyz2] macrophages (Fig. 6f–h). Hence, ERAD deficiency does not affect the intracellular trafficking of STING protein in response to activation. Taken together, we conclude that SEL1L–HRD1 ERAD regulates STING protein stability while having no effect on its intracellular trafficking.

### SEL1L–HRD1 ERAD limits STING-mediated innate immunity

Next, we explored the pathophysiological significance of SEL1L–HRD1 ERAD in STING innate immunity. First, we measured the amplitude and kinetics of STING activation. Activation kinetics of STING, as measured by percentage of p-STING in total STING, were quite similar between *Sel1L*[f/f] and *Sel1L*[Lyz2] macrophages, peaking at ~1.5 h (Fig. 7a and quantitated in Fig. 7b). However, *Sel1L*[Lyz2] macrophages mounted much more robust p-STING responses than *Sel1L*[f/f] macrophages (Fig. 7c). Hence, SEL1L–HRD1 ERAD deficiency increases the size of the activable STING pool, thereby leading to augmented STING signalling.

Next, we explored the pathological significance of myeloid-specific SEL1L–HRD1 ERAD using the herpes simplex virus (HSV-1) infection model and pancreatic cancer model. First, HSV-1 infection triggered hyper-phosphorylation of TBK1 and IRF3 in *Sel1L*[f/f] macrophages and, to a much higher extent, *Sel1L*[Lyz2] macrophages (Extended Data

Fig. 8a). *Ifnb* gene expression and secreted IFNβ protein were substantially higher in *Sel1L*[Lyz2] macrophages compared with *Sel1L*[f/f] macrophages following HSV-1 infection in a dose- and STING-dependent manner (Fig. 7d,e and Extended Data Fig. 8b,c). Similar observations were obtained in *Hrd1*[-/-] macrophages upon HSV-1 infection (Extended Data Fig. 8d,e). Hence, these data suggest that SEL1L–HRD1 ERAD in macrophages limits the cellular response to HSV-1 via STING.

Second, given the importance of STING in tumour immunity, we next assessed the role of myeloid-specific SEL1L–HRD1 ERAD in tumorigenesis following subcutaneous implantation of tumour cells from spontaneous tumours derived from *Kras*[LSL-G12D];*Trp53*[LSL-R172H];*Pdx1*-Cre mice with pancreatic ductal adenocarcinoma into *Sel1L*[Lyz2] or *Sel1L*[f/f] mice. Two intraperitoneal injections of the STING agonist DMXAA (12.5 mg kg$^{-1}$ body weight) in the second week post-tumour implantation significantly but comparably decreased tumour sizes in *Sel1L*[Lyz2] versus *Sel1L*[f/f] mice (Extended Data Fig. 9a,b). This was probably due to the activation of other STING-positive cells other than macrophages (for example, dendritic cells and fibroblasts). In keeping with this model, similar levels of the circulating inflammatory cytokines IL-6 and IFNβ were present in DMXAA-treated cohorts (Extended Data Fig. 9c). We then designed a new strategy by injecting the secretome from ex vivo-activated macrophages twice into tumour-bearing wild-type mice at the second week post-tumour implantation (Fig. 7f). Strikingly, mice that received secretome from DMXAA-treated *Sel1L*[Lyz2] macrophages exhibited significantly smaller tumours than those that received secretome from DMXAA-treated *Sel1L*[f/f] macrophages (Fig. 7g,h). Secretome from naive *Sel1L*[Lyz2] macrophages (without DMXAA treatment) was insufficient to drive anti-tumour immunity (Extended Data Fig. 9d,e). The active components of the secretome were of a protein nature as heat inactivation before the injections abolished the protective effect (Fig. 7g,h). Lastly, the protective effect of the secretome from DMXAA-treated macrophages required the involvement of T cells as it was abolished when performed in nude mice (Extended Data Fig. 9f,g). Taken together, these data demonstrate that myeloid-specific SEL1L–HRD1 ERAD limits STING signalling against DNA virus and tumour growth.

## Discussion

Activated STING could be regulated by multiple mechanisms following ligand binding at the extra-ER compartments[76], but how nascent STING protein is regulated in the ER is unclear. Here we report identification of the SEL1L–HRD1 ERAD protein complex as a key suppressor of STING innate immunity. Unlike endolysosome-dependent degradation of active STING to help terminate its signalling[6,15–18], SEL1L–HRD1 ERAD degrades naive STING to limit its activation potential. Indeed, newly synthesized STING protein is prone to misfolding and directly ubiquitinated and degraded by SEL1L–HRD1 ERAD in the ER. In the absence of SEL1L–HRD1 ERAD, STING accumulates in the ER, hence forming a larger activable STING pool in the basal state, probably through additional folding processes (Extended Data Fig. 10a). Our data further show that SEL1L deficiency enhances STING-mediated innate immunity against HSV-1 infection and malignant pancreatic tumours in a transplantation model (Extended Data Fig. 10b).

**Fig. 6 | SEL1L–HRD1 ERAD controls the size of the activable STING pool in macrophages. a**, Immunoblot analysis in the NP-40 soluble (S) and pellet (P) fractions of primary macrophages treated with or without cGAMP for 3 h. HSP90 and H2A mark the S and P fractions, respectively. **b**, Sucrose gradient fractionation followed by immunoblot analysis of STING in primary macrophages with quantitation of the percentage of STING mass in each fraction shown on the right. The results in **a** and **b** are representative of two independent biological repeats. **c,d**, Representative confocal images of STING (green) co-stained with DAPI (blue) and either the ER marker KDEL (red; **c**) or the lysosomal marker LAMP1 (pink; **d**) in primary macrophages under basal conditions. **e**, Representative confocal images of p-STING and KDEL in macrophages with or

without cGAMP treatment for 6 h. The arrows point to p-STING foci outside of the ER. The results in **c–e** are representative of four independent biological repeats. **f–h**, Quantitation of the fraction of STING in the *trans*-Golgi network (TGN38; **f**), late endosomes (CD63; **g**) and lysosomes (LAMP1; **h**) in primary macrophages treated with cGAMP for the indicated times. From left to right, *n* = 13, 25, 10, 13, 12, 11, 13 and 13 cells (**f**), 14, 15, 22, 10, 15, 15, 22 and 26 cells (**g**) and 17, 13, 17, 25, 23, 20, 19 and 30 cells (**h**) pooled from two independent experiments. Mander's overlap coefficient was used for the measurement of colocalization. Original images are shown in Extended Data Fig. 7b–d. All values represent means ± s.e.m. Statistical significance was determined by one-way ANOVA with the Newman–Keuls post-test (**f–h**).

In the absence of SEL1L–HRD1 ERAD, some substrates form high-molecular-weight protein aggregates, causing a loss-of-function effect (for example, pro-arginine vasopressin[36] and proopiomelanocortin[38]), while others may exhibit an elevated abundance of functional proteins, causing a gain-of-function effect (for example, pre-B cell receptor[33,77],

IRE1α[32], CREBH[37,75] and STING in this study). The fate of the substrates in the absence of ERAD is probably determined by the biophysical and biochemical nature of the protein[25]. Hence, we cannot generalize the effect of ERAD to the fate of substrates as it will probably be substrate specific. Moreover, the accumulation of these misfolded proteins

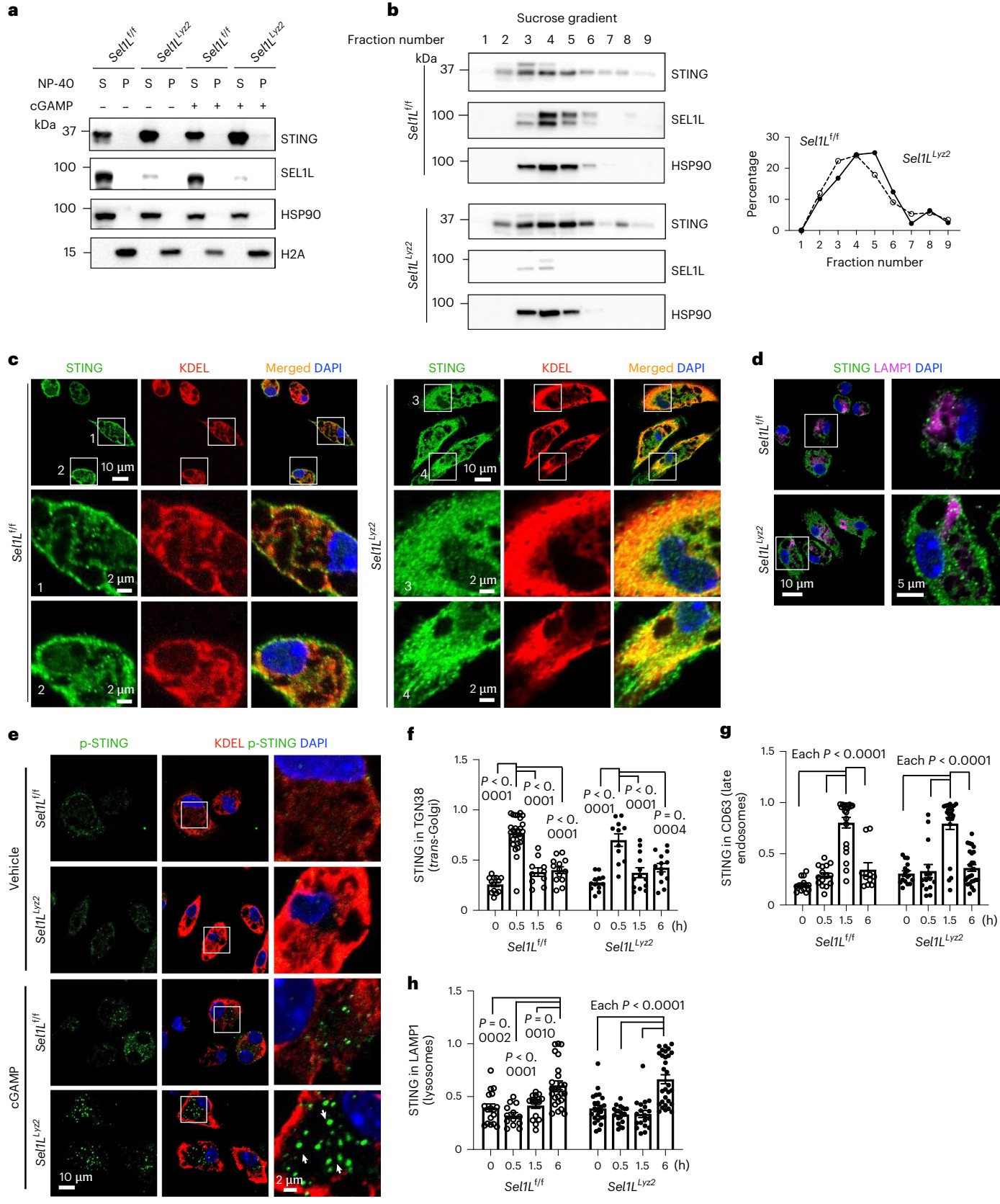

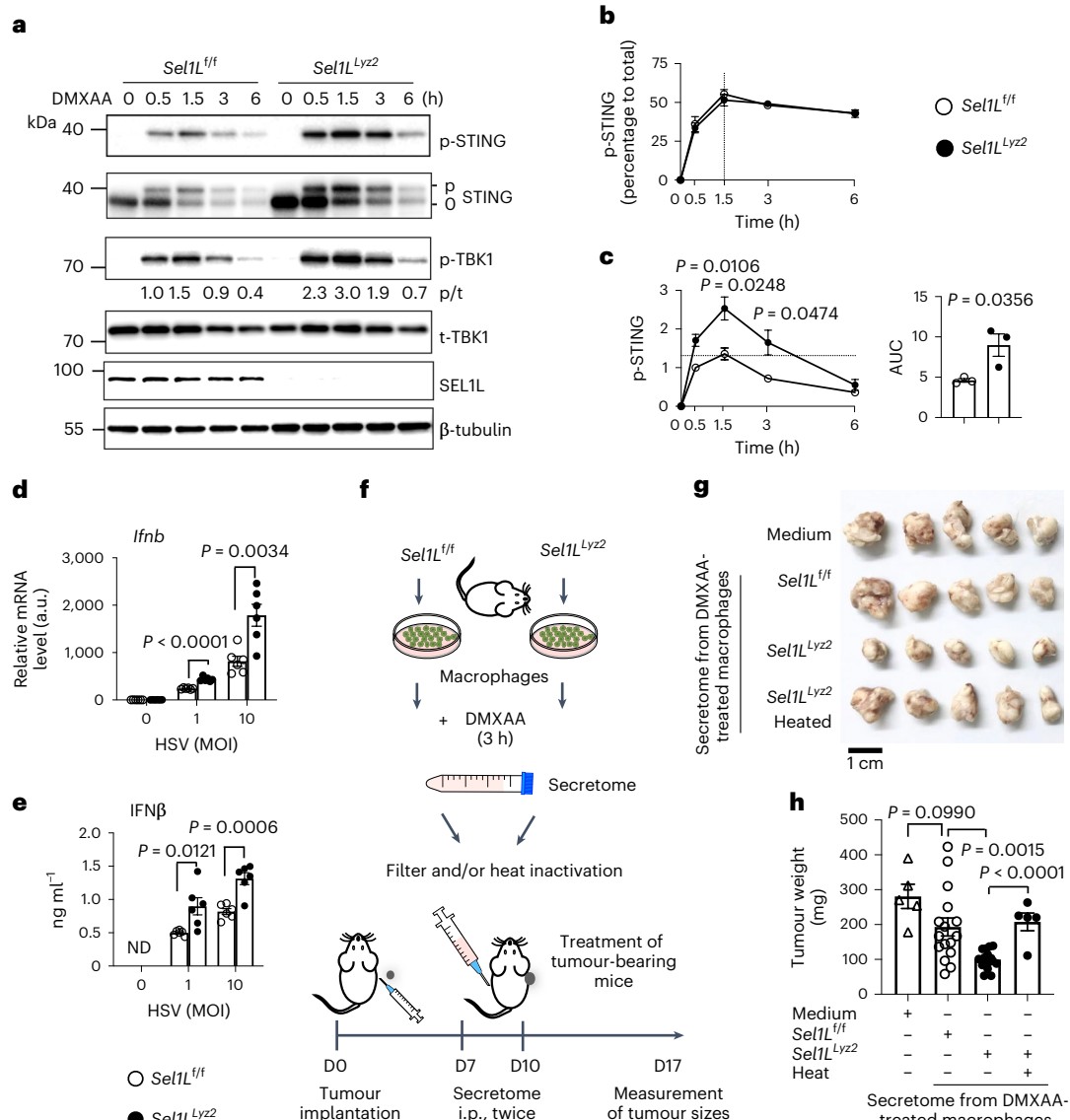

**Fig. 7 | Myeloid-specific SEL1L–HRD1 ERAD limits STING-mediated innate immunity in vitro and in vivo. a**, Immunoblots in primary macrophages treated with DMXAA at 20 µg ml⁻¹ for the indicated times. **b,c**, Quantitation of the percentages of p-STING in total STING (**b**) and p-STING signal intensity (**c**) from **a** (relative to the wild-type 0.5-h time point). The results are representative of three independent repeats. AUC, area under the curve. **d,e**, qPCR (**d**) and ELISA (**e**) analyses of *Ifnb* gene (**d**) and secreted IFNβ (**e**) in primary macrophages infected with HSV-1 at a multiplicity of infection (MOI) of 1 or 10 for 6 (**d**) and 12 h (**e**) (*n* = 6 mice each combined from two independent repeats). ND, not done. **f**, Schematic

for the cancer model in which tumour-transplanted wild-type mice received two intraperitoneal (i.p.) injections of the secretome of DMXAA-treated macrophages. D, day. **g**, Representative images of the pancreatic tumours of four groups of tumour-transplanted wild-type mice that received DMXAA-containing medium, secretomes from DMXAA-treated macrophages from *Sel1L*^f/f and *Sel1L*^Lyz2 mice and heat-inactivated *Sel1L*^Lyz2 secretomes. **h**, Quantitation of tumour weights at the end of experiment (D17) (*n* = 5, 16, 16 and 5 mice (left to right); combined from two independent repeats). All values represent means ± s.e.m. Statistical significance was determined by unpaired, two-tailed Student's *t*-test (**c**–**e** and **h**).

in the ER may not necessarily cause ER stress or overt UPR because of a number of cellular adaptive mechanisms including, but not limited to, the upregulation of ER chaperones to increase folding efficiency, the expansion of ER volume to dilute the concentration of misfolded proteins and/or enhanced protein aggregation to sequester misfolded proteins and hence attenuate their proteo-toxicity[23–25,47]. Hence, we propose that the pathophysiological effect of SEL1L–HRD1 ERAD is probably uncoupled from that of the UPR under physiological settings.

In addition to ERAD, the UPR is another branch of quality con-trol machinery that is critical for maintaining ER homeostasis. Indeed, activation of the UPR has been implicated in innate immunity and inflammatory responses. Upon TLR2/TLR4 ligand stimulation,

IRE1α becomes activated to generate the spliced form of X-box binding protein 1 (XBP1s), which is required for optimal and sustained production of proinflammatory cytokines by macrophages[53]. However, our data show that, despite the elevated protein level of IRE1α, TLR4 innate immunity is unchanged in the absence of SEL1L. Moreover, recent studies showed that persistent activation of STING leads to T cell apoptosis in a UPR-dependent manner[67] and that Toll-interacting protein deficiency decreases STING protein levels via IRE1α activation[20]. In contrast, using both pharmacological and genetic approaches, our data demonstrate that inhibition or acti-vation of the UPR and/or IRE1α pathway does not alter STING protein levels or activation. Future studies will be required to carefully examine these discrepancies.

While the ER is a well-established centre for cellular functions, including protein folding and maturation, antigen presentation and loading, mitochondrial dynamics and innate immunity[23,24,51,58], SEL1L–HRD1 ERAD has recently emerged as a key player in many physiological processes[23–25,47]. Our study shows that one of the key mechanisms underlying the regulation of STING innate immunity indeed occurs at the ER via SEL1L–HRD1 ERAD, which controls the turnover and abundance of STING in the basal state and hence its maximum activation potential. Although more factors remain to be discovered, this study demonstrates a potential therapeutic value of targeting SEL1L–HRD1 ERAD in various STING-associated diseases.

## Online content

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

[1]Department of Hepatobiliary and Pancreatic Surgery, The First Affiliated Hospital, Zhejiang University School of Medicine, Hangzhou, China. [2]Department of Molecular and Integrative Physiology, University of Michigan Medical School, Ann Arbor, MI, USA. [3]Zhejiang Provincial Key Laboratory of Pancreatic Disease, The First Affiliated Hospital, Zhejiang University School of Medicine, Hangzhou, China. [4]Graduate Program in Nutrition, Cornell University, Ithaca, NY, USA. [5]Department of Medical Microbiology and Parasitology, Zhejiang University School of Medicine, Hangzhou, China. [6]Department of Immunology, School of Medicine, Jiangsu University, Zhenjiang, China. [7]Division of Metabolism, Endocrinology and Diabetes, Department of Internal Medicine, University of Michigan Medical School, Ann Arbor, MI, USA. [8]Shanghai Institute of Nutrition and Health, Chinese Academy of Sciences, Shanghai, China. [9]Center for Translational Research in Hematologic Malignancies, Houston Methodist Cancer Center, Houston Methodist Research Institute, Houston, TX, USA. [10]Department of Biological Chemistry, University of Michigan Medical School, Ann Arbor, MI, USA. [11]These authors contributed equally: Yewei Ji, Yuan Luo. ✉e-mail: ywji@zju.edu.cn; lingq@med.umich.edu

## Methods

### Mice

*Sel1L*[fl/fl][31] or *Atg7*[fl/fl] mice[78] were crossed with myeloid-specific *Lyz2*-Cre mice (B6.129P2-*Lyz2*[tm1(Cre)Ifo]/J; strain 004781; The Jackson Laboratory) to generate myeloid cell-specific *Sel1L*[Lyz2] or *Atg7*[Lyz2] mice, with *Sel1L*[fl/fl] or *Atg7*[fl/fl] littermates as wild-type controls. *Atg7*[fl/fl] mice were provided by R. Singh (Albert Einstein College of Medicine) with permission from M. Komatsu and K. Tanaka (Tokyo Metropolitan Institute of Medical Science). OT-1 mice (C57BL/6-Tg(TcraTcrb)1100Mjb/J; strain 003831; The Jackson Laboratory) carrying a transgenic T cell receptor specifically for ovalbumin peptide 257–264 in the context of H-2K[b] BALB/c nude mice (BALB/cNj-*Foxn1*[nu]/Gpt; D000521) were purchased from GemPharmatech Co. Nude mice were on the BALB/c background, whereas all other mice were on the C57BL/6J background. All mice on a normal chow diet (13% fat, 57% carbohydrate and 30% protein; LabDiet 5LOD) were housed in a room at 20 °C and 40–60% humidity with a 12 h light/12 h dark cycle. All animal procedures were approved by and performed in accordance with the Institutional Animal Care and Use Committee at the University of Michigan Medical School (PRO00008989) and Cornell University (2007-0051) and the Animal Experimentation Ethics Committee of the First Affiliated Hospital of the Zhejiang University School of Medicine (2021-0135).

### Power analysis of animal size

Based on the sample size formula of the power analysis, $n = 8(CV)^2[1 + (1 - PC)^2]/(PC)^2$, to reach an error of 0.05, a power of 0.80, a percentage change in means (PC) of 20% and a coefficient of variation (CV) of ~10–15% (varies between the experiments), the minimum number of mice required to obtain statistical significance and ensure adequate power is four to six per group. Mice in each group were randomly chosen based on age, genotype and gender.

### Preparation of primary macrophages

Peritoneal macrophages were obtained 4 d after intraperitoneal injection of 2 ml aged 4% brewed thioglycollate broth (VWR 90000-294). Mice were euthanized and macrophages were collected by injection of phosphate-buffered saline (PBS) into the peritoneal cavity. The peritoneal exudate cells were centrifuged at ~1,000 r.p.m. for 10 min, treated with red blood cell lysis buffer and resuspended in culture medium in six-well plates for further analyses.

### HFD feeding

Eight-week-old male mice were placed on a 60% HFD composed of 60% fat, 20% carbohydrate and 20% protein (D12492; Research Diets) for up to 20 weeks. For the glucose tolerance test, mice were fasted for 16–18 h followed by injection of glucose (Sigma–Aldrich) at 1 g kg⁻¹ body weight. For the insulin tolerance test, mice were fasted for 4 h followed by an intraperitoneal injection of insulin (Sigma–Aldrich) at 40 μg kg⁻¹ body weight. Blood glucose was monitored using a OneTouch Ultra Glucose Meter at the indicated time points post-injection. Fasting glucose levels were measured following a 16 h fast.

### In vivo tumour study

The pancreatic ductal adenocarcinoma cell line derived from spontaneous tumours in a *Kras*[LSL-G12D];*Trp53*[LSL-R172H];*Pdx1*-Cre mouse model was a kind gift from R. Kalluri (MD Anderson Cancer Center)[79]. Either 5.0 × 10⁵ or 3.5 × 10⁵ *Kras*[LSL-G12D];*Trp53*[LSL-R172H];*Pdx1*-Cre cells in 100 μl PBS were injected subcutaneously into the right flank of 6- to 8-week-old C57BL/6J or nude mice, respectively. Within 1 week following tumour cell implantation, mice with similar tumour volumes and body weights were randomized into different groups of anti-tumour treatment, DMXAA (12.5 mg kg⁻¹ body weight diluted in 100 μl PBS; MedChemExpress) or DMXAA- or mock-stimulated macrophage secretomes (0.5 ml; 1 million macrophages) intraperitoneally twice at days 7 and 10. For heat inactivation, secretomes were heated at 80 °C for

20 min before injections. Tumour growth was monitored and tumour volumes were calculated as (length × width²)/2. A tumour burden of <10% of body weight or a tumour size of <20 mm in any dimension was permitted by the Ethics Committee of the First Affiliated Hospital of the Zhejiang University School of Medicine. The maximal tumour size/burden was not exceeded in this study.

### Flow cytometric analysis

The following fluorochrome- or biotin-conjugated antibodies were used: CD4 (GK1.5; 100408; BioLegend), CD8 (YTS169.4; MA5-17605, MA5-17607; Thermo Fisher Scientific), F4/80 (BM8; 123116 and 123114; BioLegend), CD11b (M1/70; 101206; BioLegend), Gr-1 (RB6-8C5; 108408; BioLegend), TCRβ (H57-597; 109206; BioLegend), B220 (RA3-6B2; 103206 and 103208; BioLegend), CD45 (30-F11; 103130; BioLegend), I-A/I-E (M5/114.15.2; 107645 and 107608; BioLegend), H-2K[b]/H-2D[b] (28-8-6 and AF6-88.5; 114606 and 116506; BioLegend), TLR2 (CB225; 148604; BioLegend), TLR4 (SA15-21; 145406; BioLegend), PD-L1 (10F.9G2; 124308; BioLegend) and isotype control antibodies. Following incubation with anti-CD16/CD32 (93; 101302; BioLegend) antibody to block Fc receptors, $1 \times 10^6$ cells were incubated with 20 μl of antibodies diluted at optimal concentrations (at 1:100 or 200) for 20 min at 4 °C. Cells were washed three times with PBS and then resuspended in 200 μl PBS for analysis. For intracellular staining, cells were fixed after surface staining and permeabilized with a BD Cytofix/Cytoperm fixation/permeabilization kit, according to the manufacturer's protocol, before analysis using the BD LSR cell analyzer. Data were analysed using CellQuest software (BD Biosciences) and FlowJo (Flowjo.com). Purification and characterization of stromal vascular cells from epididymal fat pads using flow cytometric analysis were performed as previously described[80,81].

### TEM

Peritoneal macrophages were seeded in six-well plates at $15 \times 10^6$ cells per well, followed by fixation, staining, dehydration, processing and imaging acquisition using a JEOL JEM-1400 TEM performed on a fee-for-service basis at the Electron Microscopy and Histology Core Facility at the Weill Cornell Medical College. For quantitation, regions of the ER in the TEM images were selected using the multiple AOI menu and analysed under the count and measure objects menu in Image-Pro Plus 6.0 software.

### Western blot and image quantitation

The preparation of cell lysates, phosphatase/EndoH treatment and (Phos-tag-based) western blots were performed as previously described[49,50]. The antibodies used in this study were: HSP90 (1:6,000; Abcam; ab13492), β-tubulin (1:3,000; 10068-1-AP; Proteintech), caspase-3 (1:1,000; 8G10; Cell Signaling Technology), β-actin (1:3,000; 20536-1-AP; Proteintech), IκBα (1:2,000; 9242; Cell Signaling Technology), SEL1L (1:1,000; ab78298; Abcam), BiP (1:5,000; ab21685; Abcam), HRD1 (1:300 (R. Wojcikiewicz) or 1:1,000 (13473-1-AP; Proteintech)), STING (1:1,500 (19851-1AP; Proteintech) or 1:2,000 (D2P2F; Cell Signaling Technology)), p-Ser365 STING (1:2,000; D8F4W; Cell Signaling Technology), cGAS (1:2,000; D3080; Cell Signaling Technology), p-Ser172 TBK1 (1:1,000; D52C2; Cell Signaling Technology), TBK1 (1:2,000; E9H5S; Cell Signaling Technology), p-Ser396 IRF-3 (1:2,000; D601M; Cell Signaling Technology), IRF-3 (1:2,000; D83B9; Cell Signaling Technology), ATG7 (1:1,000; D12B11; Cell Signaling Technology), OS9 (1:3,000; ab109510; Abcam), eIF2α (1:2,000; 9722; Cell Signaling Technology), p-eIF2α (1:2,000; 3597S; Cell Signaling Technology), IRE1α (1:3,000; 3294; Cell Signaling Technology), ERP44 (1:3,000; 2886; Cell Signaling Technology), STIM1 (1:2,000; 4916; Cell Signaling Technology), HA (1:2,000; H3663; Sigma–Aldrich), c-Myc (1:2,000; C3956; Sigma–Aldrich), Flag (1:2,000; F1804; Sigma–Aldrich), H2A (1:5,000; 2578; Cell Signaling Technology), LC3B (1:2,000; 2775; Cell Signaling Technology), PDI (1:2,000; ADI-SPA-890; Enzo Life Sciences),

ubiquitin (1:200; P4D1; Santa Cruz Biotechnology), SOAT1 (1:1,000; GTX32890; GeneTex), FACL4 (1:1,000; ab155282; Abcam) and calnexin (1:20,000; 10427-2-AP; Proteintech). The secondary antibodies were: goat anti-rabbit IgG-HRP (1:5,000; 1721019; Bio-Rad) and goat anti-mouse IgG-HRP (1:5,000; 1721011; Bio-Rad). The band density was quantitated using the Image Lab Software on the ChemiDoc XRS+ System (Bio-Rad). Protein levels were normalized to HSP90, β-tubulin or actin. Phosphorylated forms of proteins were normalized to the levels of total proteins. The data are presented as means ± s.e.m. unless otherwise specified.

## Immunoprecipitation

Cells were incubated with 20 mM N-ethylmaleimide in PBS for 10 min on ice, snap frozen and lysed in lysis buffer (150 mM NaCl, 1 mM ethylene-diaminetetraacetic acid, 50 mM Tris-HCl (pH 7.5), protease inhibitor cocktail (Sigma–Aldrich) and 10 mM N-ethylmaleimide) supplemented with either 1% Triton X-100 or Nonidet P-40 (NP-40) on ice for 15 min. Cells were centrifuged at 12,000g at 4 °C for 10 min. Supernatants were collected and the protein concentration was measured using the Bradford assay. The protein lysates were incubated with antibodies specific for STING (19851-1AP; Proteintech) or SEL1L (ab78298; Abcam) followed by Protein A Agarose beads (20334; Invitrogen), or directly with agarose-conjugated anti-FLAG (A4596; Sigma–Aldrich), anti-Myc (16-219; Sigma–Aldrich) or streptavidin agarose (20353; Thermo Fisher Scientific) for 16 h at 4 °C with gentle rocking (1 µg antibody or 30 µl agarose beads for 1 ml sample lysis), followed by five washes with the lysis buffer. Immunocomplexes were eluted by boiling for 5 min in sodium dodecyl sulfate (SDS) sample buffer followed by SDS polyacrylamide gel electrophoresis (SDS-PAGE) and western blot analysis.

## Sucrose gradient sedimentation analysis

Confluent primary macrophages in two 10 cm plates were harvested and lysed in 0.5 ml 1% NP-40 lysis buffer, as described above in the section 'Immunoprecipitation'. Lysates were loaded onto 4 ml sucrose gradients at ~20–50% in 150 mM NaCl, 1 mM ethylenediaminetetraacetic acid, 50 mM Tris-HCl (pH 7.5) and protease inhibitors, which were freshly prepared by layering higher- to lower-density sucrose fractions in 5% increments in polyallomer tubes of 11 mm × 3 mm × 60 mm (Beckman Coulter). Following centrifugation at 58,000 r.p.m. for 14.5 h at 4 °C using an SW 60 Ti rotor (Beckman Coulter), nine fractions were collected from the top fraction (fraction 1; the lowest density) to the bottom fraction (fraction 9; the highest density) and subsequently subjected to western blot analysis under denaturing conditions. The band intensity of each fraction was quantitated and the percentage of protein in each fraction was calculated by dividing the protein intensity in individual fractions by the total protein intensity in all fractions.

## SDS-PAGE

Protein samples were prepared in 1× denaturing SDS sample buffer (50 mM Tris-HCl (pH 6.8), 2% SDS, 10% glycerol, 0.28 M β-mercaptoethanol and 0.01% bromophenyl blue) and boiled at 95 °C for 5 min before SDS-PAGE.

## NP-40 solubility assay

Primary macrophages were harvested and lysed in NP-40 lysis buffer (50 mM Tris-HCl (pH 8.0), 0.5% NP-40, 150 mM NaCl and 5 mM MgCl$_2$) supplemented with protease inhibitors. The lysates were centrifuged at 12,000g for 10 min and the supernatant was collected as the soluble NP-40S fraction. The pellet was then resuspended in 1× SDS sample buffer with the volume normalized to the initial cell weight, heated at 95 °C for 30 min and collected as the insoluble NP-40P fraction. The NP-40S and NP-40P fractions were subsequently analysed by western blot.

## RNA extraction, reverse transcription and quantitative PCR

RNA from cells and tissues was extracted using TRIzol and a Qiagen RNA miniprep kit. Reverse transcription and quantitative PCR (qPCR) analyses were performed as previously described[30]. qPCR data were collected using a Roche LightCycler 480 instrument and the gene expression was normalized to that of the ribosomal L32 gene for each sample. The qPCR primers used for the mouse genes were as follows: GAGCAACAAGAAAACCAAGCA and TGCACACAAGCCATCTACTCA for L32; TGGGTTTTCTCTCTCTCCTCTG and CCTTTGTTCCGGTTACTTCTTG for Sel1L; AGCTACTTCAGTGAACCCCACT and CTCCTCTACAATGCCCACTGAC for Hrd1; ACTATGTGCACCTCTGCAGC and GTCCAGAATGCCCAAAAGG for Xbp1u; CTGAGTCCGAATCAGGTGCAG and GTCCATGGGAAGATGTTCTGG for Xbp1s; TGTGGTACCCACCAAGAAGTC and TTCAGCTGTCACTCGGAGAAT for Grp78; TCAGCCGATTTGCTATCTCATA and AGTACTTGGGCAGATTGACCTC for Tnfa; AGACAAAGCCAGAGTCCTTCAG and TGCCGAGTAGATCTCAAAGTGA for Il6; AGATCAACCTCACCTACAGG and TCAGAAACACTGTCTGCTGG for Ifnb; CCTGCCCACGTGTTGAGAT and TGATGGTCTTAGATTCCGGATTC for Cxcl10; and AAATAACTGCCGCCTCATTG and ACAGTACGGAGGGAGGAGGT for Sting.

Primers for M1/M2 markers were used as previously described[80,81]. The qPCR conditions were: 94 °C for 5 min, then 40 cycles of 94 °C for 15 s, 58 °C for 15 s and 72 °C for 30 s, followed by dissociation curve analysis. The reverse transcription PCR conditions were: 94 °C for 5 min, then 30–40 cycles of 94 °C for 15 s, 58 °C for 15 s and 72 °C for 30 s, followed by 70 °C for 10 min.

## Drug treatment

Cells were maintained in Dulbecco's Modified Eagle Medium supplemented with 10% foetal bovine serum (HyClone) and 1% penicillin/streptomycin. Thapsigargin (EMD Calbiochem) was dissolved to 0.6 mM in dimethyl sulfoxide and used at 300 nM. 1 mM tauroursodeoxycholate sodium (MedChemExpress), 20 nM bafilomycin A1 (Selleck Chemicals) and 0.1 mM 4µ8c (MedChemExpress) were used in cell culture. Inflammatory stimuli included the TLR2 ligand Pam3Cy (InvivoGen) at 1 µg ml⁻¹, the TLR4 ligand LPS (InvivoGen) at 500 ng ml⁻¹, the STING ligands 2′3′-cGAMP (InvivoGen), c-di-AMP (InvivoGen) and DMXAA (MedChemExpress) at 3.5 µg ml⁻¹, 3.5 µg ml⁻¹ and 20 µg ml⁻¹, respectively, and the RIG-1 ligand poly(I:C) (InvivoGen) at 2 µg ml⁻¹. 2′3′-cGAMP, c-di-AMP and poly(I:C) were delivered by transfection with lipofectamine 2000 (Thermo Fisher Scientific). The STING inhibitor H151 (InvivoGen) was dissolved in dimethyl sulfoxide at 10 mg ml⁻¹ and used at 4 µg ml⁻¹.

## In vitro T cell activation

Macrophages were cultured in 96-well plates (4 × 10⁵ cells per well) together with 4 × 10⁵ CD8⁺ T cells isolated from OT-1 mouse splenocytes and 5 µM OVA257-264 (SIINFEKL; Biomatik) at 37 °C for 48 h. For CD1d-restricted NKT cell line DN32.D3 activation, macrophages were pre-incubated with 100 ng ml⁻¹ α-galactoceramide (Toronto Research Chemicals) for 1 h and, following two washes with culture medium, incubated with 2 × 10⁵ DN32 overnight. The supernatant was collected and analysed by enzyme-linked immunosorbent assay (ELISA) for IL-2 levels.

## HSV-1 infection

HSV-1 and Vero cells were kindly provided by M. Raghavan at the University of Michigan Medical School[82]. HSV-1 was propagated and titered by plague assays on Vero cells. Macrophages were treated with the indicated multiplicity of infection and for the indicated times before ELISA analysis of the supernatant or western blot and gene expression analyses of frozen cells.

## LPS challenge in vivo

Eight-week-old female mice were injected intraperitoneally with LPS at 40 mg kg⁻¹ body weight and observed for survival every 4 h.

Serum was collected at the 0, 3 and 6 h time points post-injection for cytokine analysis.

## TurboID proximity labelling

RAW 264.7 macrophages or MEF cells were transfected with STING-TurboID-V5 or SEL1L-TurboID-V5 plasmid using DNA transfection reagent (Invigentech) in serum-free medium in the presence of H151. After 18 h, cells were stimulated with 2′3′-cGAMP for the indicated times followed by the addition of 50 µM biotin at 37 °C for 10 min. The reaction was stopped by transferring the cells to ice and washing them five times with ice-cold PBS. Cells were then lysed in lysis buffer and immunoprecipitated for biotinylated proteins using streptavidin agarose beads (Sigma–Aldrich) followed by western blot.

## CRISPR-mediated gene knockout cells

CRISPR-based knockout cell lines were generated as previously described[33] using the lentiCRISPRv2 vector from the Zhang laboratory at the Massachusetts Institute of Technology, which expresses the single guide RNA, Cas9 protein and puromycin resistance gene. The *Hrd1* and *Ern1* single guide RNAs were designed, synthesized and cloned into the lentiCRISPRv2 vector (for *Hrd1*, 5′-CACC GATCCATGCGGCATGTCGGGC-3′ (forward) and 5′-AAACGCCCGA CATGCCGCATGGATC-3′ (reverse); for *Ern1*, 5′-CACCGTGCCATCAT TGGGATCTGGG-3′ (forward) and 5′-AAACCCCAGATCCCAATGATG GCAC-3′ (reverse) and 5′-CACCGCTTGGAGGCAAGAACAACGA-3′ (forward) and 5′-AAACTCGTTGTTCTTGCCTCCAAGC-3′ (reverse)).

## ELISA

TNFα, IL-6, IL-1β, IL-2 and IFNβ ELISA kits were purchased from eBioscience or BioLegend. An Insulin ELISA Kit was purchased from Crystal Chem. All ELISAs were performed per the suppliers' protocols.

## Haematoxylin and eosin staining

Liver and adipose tissue from mice fed a HFD were collected and fixed in 4% formaldehyde. Samples were sent to the Histology Core Laboratory at Cornell University for the performance of haematoxylin and eosin staining on a fee-for-service basis. Haematoxylin and eosin images of liver and adipose tissue were collected using Aperio ImageScope software.

## Immunofluorescence staining

Cells were plated on slides, fixed overnight at 4 °C, incubated overnight in cold PBS and then incubated overnight in cold PBS with 20% sucrose. Slides were washed three times in PBS followed by blocking buffer (5% bovine serum albumin and 0.1% Tween in Tris-buffered saline (TBST)) for 30 min. The following primary antibodies were diluted in the blocking buffer and applied at 4 °C overnight: STING (1:200; 19851-1AP; Proteintech), KDEL (1:200; MAC 256; Abcam), p-Ser365 STING (1:200; D1C4T; Cell Signaling Technology), TGN38 (1:200; sc-166594; Santa Cruz Biotechnology), CD63 (1:200; sc-5275; Santa Cruz Biotechnology) and LAMP1 (1:50; 1D4B; Developmental Studies Hybridoma Bank). Slides were washed three times with TBST for 10 min each and then incubated with conjugated secondary antibodies for 2 h at room temperature. Following extensive washes with TBST, slides were covered with ProLong Gold Antifade/DAPI (Thermo Fisher Scientific). Fluorescence Images were captured under a Nikon A1 confocal microscope at the Brehm Diabetes Research Center Imaging Facility at the University of Michigan Medical School or by STEDYCON super-resolution microscopy (using secondary antibodies conjugated with Abberior STAR RED and Abberior STAR ORANGE) at the Imaging Core Facility of the First Affiliated Hospital at the Zhejiang University School of Medicine.

## Statistical analysis

The results are expressed as means ± s.e.m. Comparisons between groups were made using an unpaired two-tailed Student's *t*-test (two groups) or one-way analysis of variance (ANOVA) with Newman–Keuls post-test (multiple groups). Individual data values were provided in the source data. The data distribution was assumed to be normal, but this was not formally tested. No sample size calculation was performed for the in vitro experiments. No animals or samples were excluded from the analysis. For the LPS, DMXAA injection and HFD experiments, mice were sex and age matched and randomly assigned to experimental groups according to genotype. For the secretome treatment experiments, mice with similar tumour volumes and body weights following tumour cell implantation were randomized into different treatment groups. For the ligand and chemical treatment experiments, cell culture samples were randomly assigned to control and experimental groups. In most cases, data collection and analysis were not performed blind to the conditions of the experiments; however, most studies were repeated independently by at least two different individuals. All of the experiments were repeated at least twice or performed with independent samples.

## Reporting summary

Further information on research design is available in the Nature Portfolio Reporting Summary linked to this article.

## Data availability

Previously published STING proximity-based proteomics data are available[6]. Other data supporting the findings of this study are available from the corresponding author upon reasonable request. Source data are provided with this paper.

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

## Acknowledgements

We thank R. Singh (Albert Einstein College of Medicine), M. Komatsu and K. Tanaka (Tokyo Metropolitan Institute of Medical Science) for the *Atg7*f/f mice; M. Raghavan (University of Michigan Medical School) for the HSV-1 virus; J. Moon (University of Michigan Medical School) for the TLR2 and RIG-1 agonists; M. Kronenberg (La Jolla Institute for Immunology) for the DN32.D3 cell line; and other members of the L.Q. laboratory for comments and technical assistance. This work was supported by R01CA163910 to C.-C.A.H., the National Natural Science Foundation of China (82171731) and Investigator Start-up Fund of the First Affiliated Hospital of Zhejiang University (B20735) to Y.J., the National Natural Science Foundation of China (81871234) to S.X., the National Natural Science Foundation of China (82188102) to T.L. and 1R01DK120047, 1R01DK120330, 1R35GM130292 and the Michigan Protein Folding Diseases Initiative to L.Q. Y.J. was supported in part by American Heart Association Scientist Development Grant 17SDG33670192 and Michigan Nutrition Obesity Research Center Pilot/Feasibility Grant P30DK089503. S.A.W. is supported by American

Heart Association Predoctoral Fellowship 828841. L.Q. is the recipient of Junior Faculty, Career Development and Innovative Basic Science awards from the American Diabetes Association.

## Author contributions

Y.J. and Y. Luo. designed and performed most of the experiments. Y.W., Y.S., M.S., L.Z., X.W., Z.H., S.A.W., L.L.L., Y. Lu., L.C., F.C., Shengnuo Chen, W.Q., X.X., Siyu Chen and Z.Z. assisted with some of the experiments. Z.X. performed the initial characterization of *Sel1L*<sup>Lyz2</sup> mice. Z.Z., D.P., S.X., C.-C.A.H. and T.L. provided some of the reagents and discussions. L.Q. and Y.J. conceived of/supervised the projects and wrote the manuscript. All authors edited and approved the manuscript.

## Competing interests

The authors declare no competing interests.

## Additional information

**Extended data** is available for this paper at https://doi.org/10.1038/s41556-023-01138-4.

**Correspondence and requests for materials** should be addressed to Yewei Ji or Ling Qi.

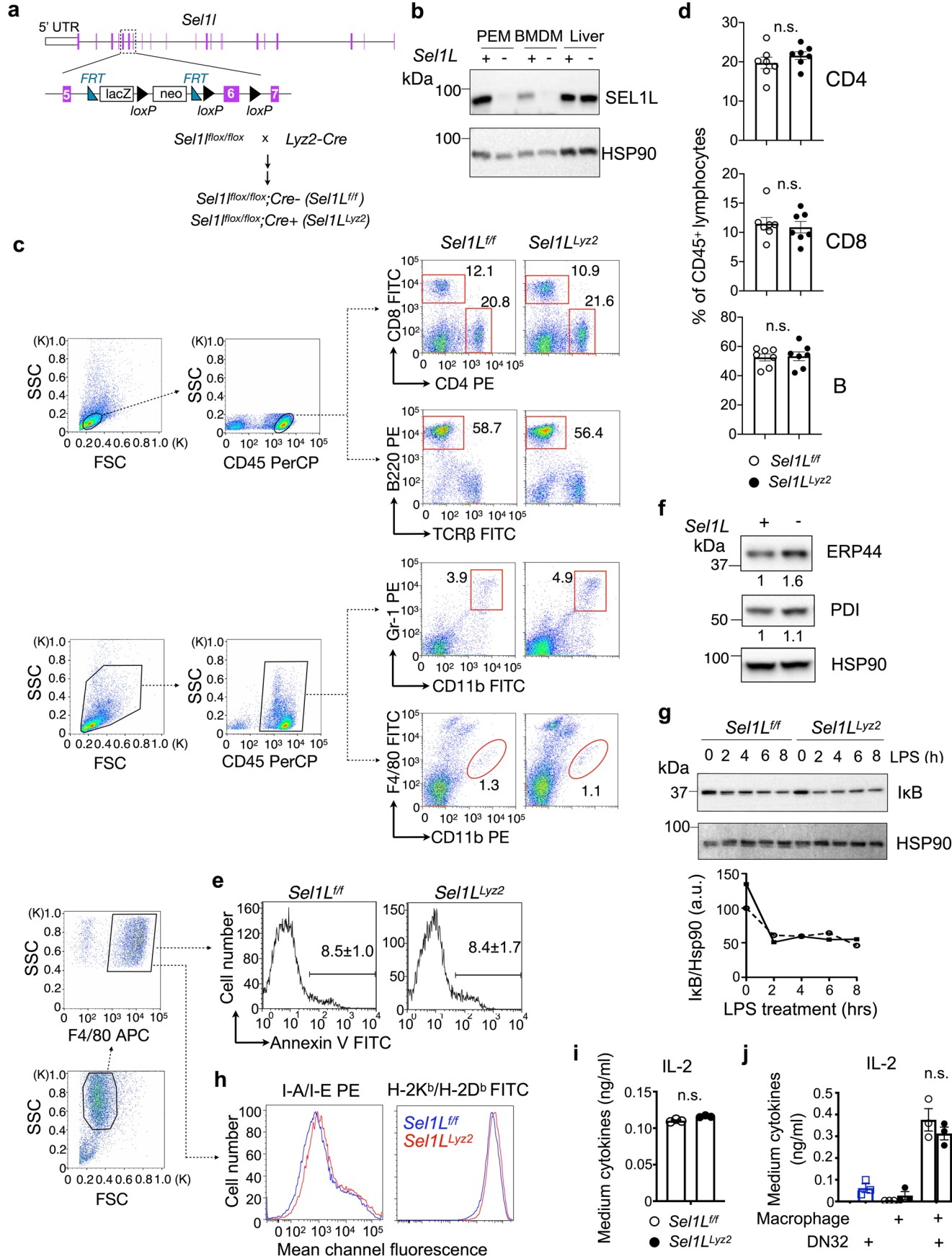

**Extended Data Fig. 1 | See next page for caption.**

**Extended Data Fig. 1 | Generation of *Sel1L^Lyz2* mice and characterization of immune cell composition, inflammation and cell death.** (**a**) Schematic diagram of the *Sel1L* floxed allele and generation of *Sel1L^Lyz2* mice. Exon 6 of the *Sel1L* gene was flanked by two loxP sites. (**b**) Immunoblot of SEL1L showing cell type-specific deletion of SEL1L in *Sel1L^Lyz2* mice, representative of three independent repeats. PEM, peritoneal exudate macrophages; BMDM, bone marrow-derived macrophages. (**c**, **d**) Flow cytometric analysis showing CD4⁺ T, CD8⁺ T, B220⁺ B cells, Gr-1⁺ CD11b⁺ neutrophils and F4/80⁺CD11b⁺ macrophages in the spleens (c), with quantitation of myeloid cells and lymphocytes shown in Fig. 1b and Extended Data Fig. 1d, respectively. n = 7 mice each, combined from 2 independent repeats. (**e**) Flow cytometric analysis of Annexin V⁺ primary macrophages. (**f**) Immunoblot showing expression of ER proteins in macrophages, with quantitation (normalized to HSP90) shown below the gel. Each lane, pooled macrophages from three mice. Data are representative of three independent biological repeats. (**g**) Immunoblot showing IκB protein levels post-LPS treatment in macrophages, with quantitation shown below. (**h**) Flow cytometric analysis of surface H-2K^b/H-2D^b (MHC I) and I-A/I-E (MHC II) levels on macrophages. Gating strategies shown on the left (c, e, h). (**i**, **j**) ELISA showing secreted IL-2 levels in the culture supernatants of macrophages with either OT1 CD8⁺ T (i) or DN32.D3 NKT cells (j). N = 3 each, from 2 independent repeats (i, j). Values, mean ± s.e.m. n.s., not significant by unpaired, two-tailed, Student's *t*-test (d, i, j).

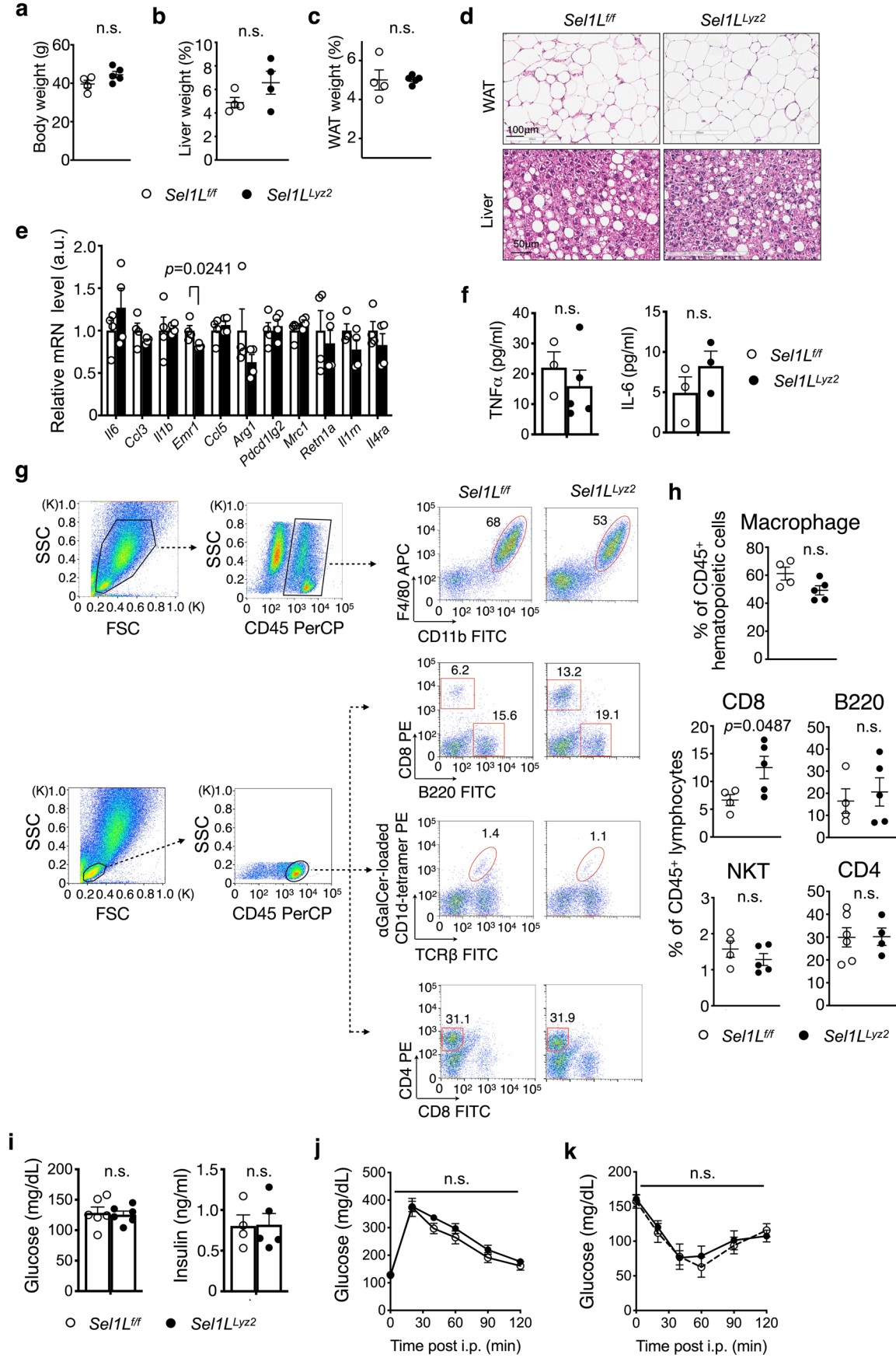

**Extended Data Fig. 2 | See next page for caption.**

**Extended Data Fig. 2 | Myeloid-specific SEL1L is dispensable for inflammatory responses and insulin sensitivity in diet-induced obesity.** (a–c) Body (a) and tissue (b-c) weights after 20 weeks on 60% high-fat-diet (HFD). From left to right, n = 4, 5 mice (a,c) and n = 4 mice each (b), combined from 2 independent repeats (a-c). (**d**) H&E images of WAT and liver of mice on HFD for 20 weeks, representative of 2 independent biological repeats. (**e**) Q-PCR analysis of M1/M2 macrophage markers in WAT. n = 4 mice each. (**f**) ELISA analysis of serum TNFα and IL-6 levels in mice on HFD for 20 weeks. n = 3 *Sel1L^{f/f}*, 5 *Sel1L^{Lyz2}* mice for TNFα and n = 3 mice each for IL-6, combined from 2 independent repeats. (**g**, **h**) Flow cytometric analysis of various immune cells in WAT following 20-week HFD, with quantitation shown in h. n = 4 *Sel1L^{f/f}*, 5 *Sel1L^{Lyz2}* mice for macrophages, CD8⁺,

B220⁺ and NKT cells; and n = 6, 4 mice for CD4⁺ cells. Gating strategies shown on the left. Data are combined from 2 independent repeats. (**i**) Serum glucose and insulin levels in HFD mice following a 6 hr fast for insulin and 16 hr fast for glucose. n = 6 mice each for glucose, and n = 4 *Sel1L^{f/f}*, 5 *Sel1L^{Lyz2}* mice for insulin, combined from 2 independent repeats. (**j**) Glucose tolerance test (GTT) of mice on HFD for 20 weeks. n = 6 mice each. (**k**) Insulin tolerance test (ITT) of mice on HFD for 14 weeks. n = 4 *Sel1L^{f/f}* and 6 *Sel1L^{Lyz2}* mice. Data are representative of two independent repeats, and source data for all repeats are provided (j, k). Values, mean ± s.e.m. *P* values were determined by unpaired, two-tailed, Student's *t*-test (a-c, e, f, h-k); n.s., not significant.

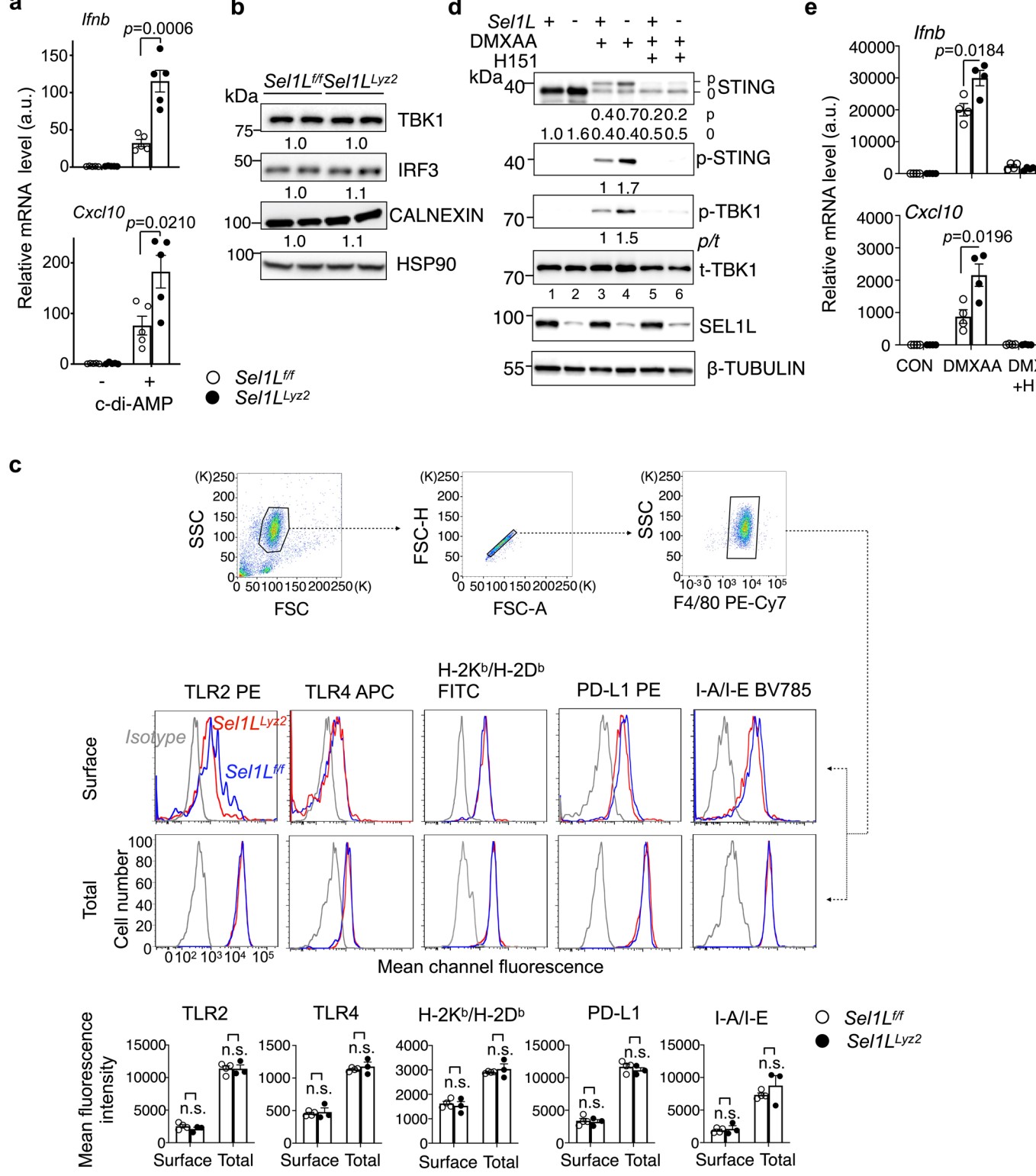

**Extended Data Fig. 3 | Loss of SEL1L specifically enhances STING signaling and STING protein level.** (**a**) q-PCR analysis of *Infb* and *Cxcl10* in primary macrophages treated with vehicle or c-di-AMP for 3 hr. n = 5 mice for each, combined from two independent repeats. (**b**) Immunoblot analysis of indicated proteins in primary macrophages, with quantitation of the average of two samples (normalized to HSP90) shown below the gel, representative of at least two independent repeats. (**c**) Flow cytometric analysis of surface (upper) and total levels of TLR2, TLR4, PD-L1, H-2K^b/H-2D^b (MHC I) and I-A/I-E (MHC II) in macrophages, with quantitation of mean fluorescence intensity shown below.

Gating strategies shown on top. n = 4 *Sel1L^{f/f}* and 3 *Sel1L^{Lyz2}* mice each, combined from two independent repeats. (**d**) Immunoblot analysis of the STING pathway in primary macrophages pretreated with or without the STING inhibitor H151 for 2 hr followed by DMXAA treatment for another 1 hr. Quantitation shown below the gel. *p-/0-/t-*, phosphorylated-/non-phosphorylated/total proteins. Data are representative of two biologically independent repeats. (**e**) q-PCR analysis of *Infb* and *Cxcl10* in macrophages treated as in (d). n = 4 mice each, combined from two independent repeats. Values, mean ± s.e.m. P values were determined by unpaired, two-tailed, Student's *t*-test (a, c, e); n.s., not significant.

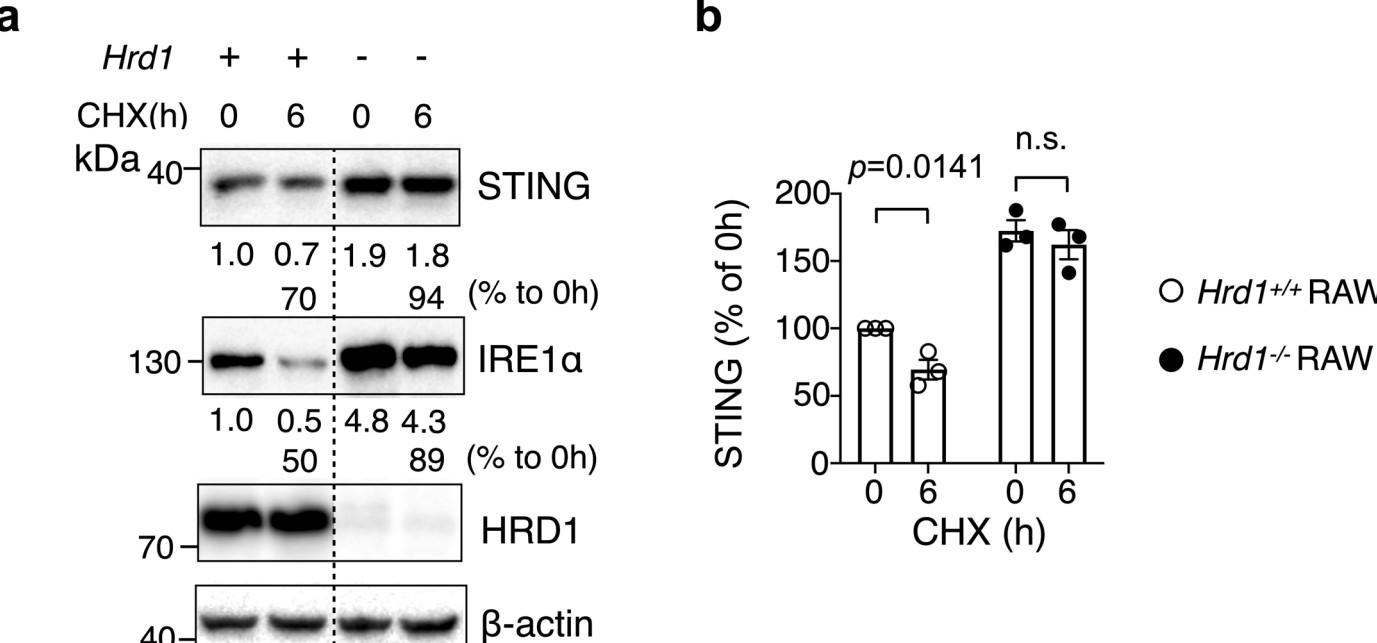

**Extended Data Fig. 4 | STING protein stabilization in *Hrd1*⁻/⁻ macrophages.** (**a**, **b**) Immunoblot analysis in WT or *Hrd1*⁻/⁻ RAW 264.7 cells treated with cycloheximide (CHX) for indicated time points with quantitation of relative band intensity (normalized to β-actin) shown below the gel (**a**), and quantitation of

STING from 3 independent repeats shown in (**b**). Values represent mean ± s.e.m. *P* values were determined by unpaired, two-tailed, Student's *t*-test; n.s., not significant.

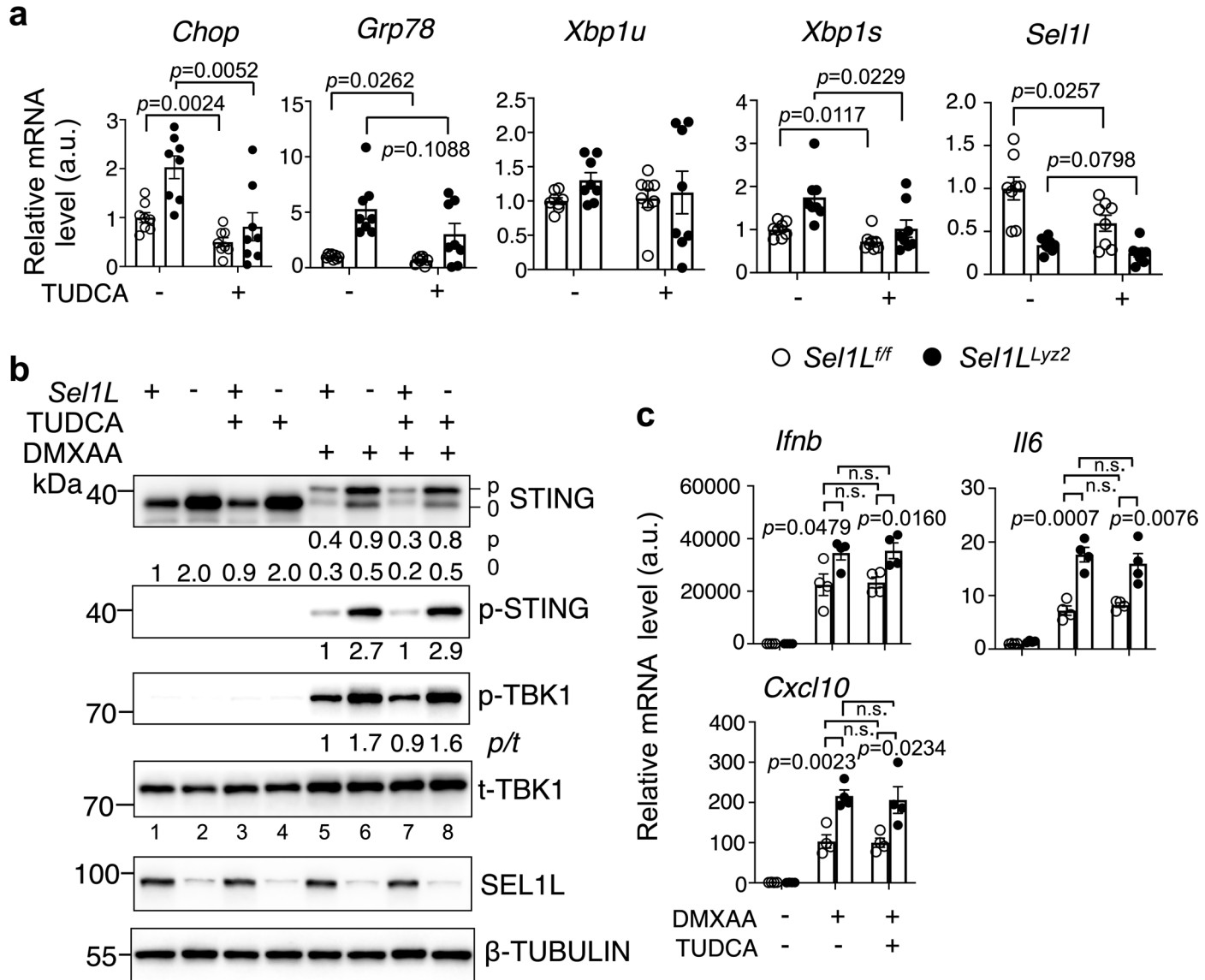

**Extended Data Fig. 5 | The effect of SEL1L-HRD1 ERAD on STING signaling is uncoupled from ER stress. (a)** q-PCR analysis of ER stress markers in macrophages treated with vehicle or TUDCA for 24 hr. n = 8 mice each, combined from 3 independent repeats. (**b**) Immunoblot of the STING pathway in macrophages pretreated with TUDCA for 24 hr followed by DMXAA for another 1 hr. The numbers below the blot indicate relative band intensity of STING, p-STING (normalized to β-tubulin), or ratio of phosphorylated to total protein (*p/t*), representative of 2 independent repeats. (**c**) Q-PCR analysis of inflammatory genes in primary macrophages treated as in (b). n = 4 mice each, combined from 2 independent repeats. Values, mean ± s.e.m. *P* values were determined by unpaired, two-tailed, Student's *t*-test (a, c); n.s., not significant.

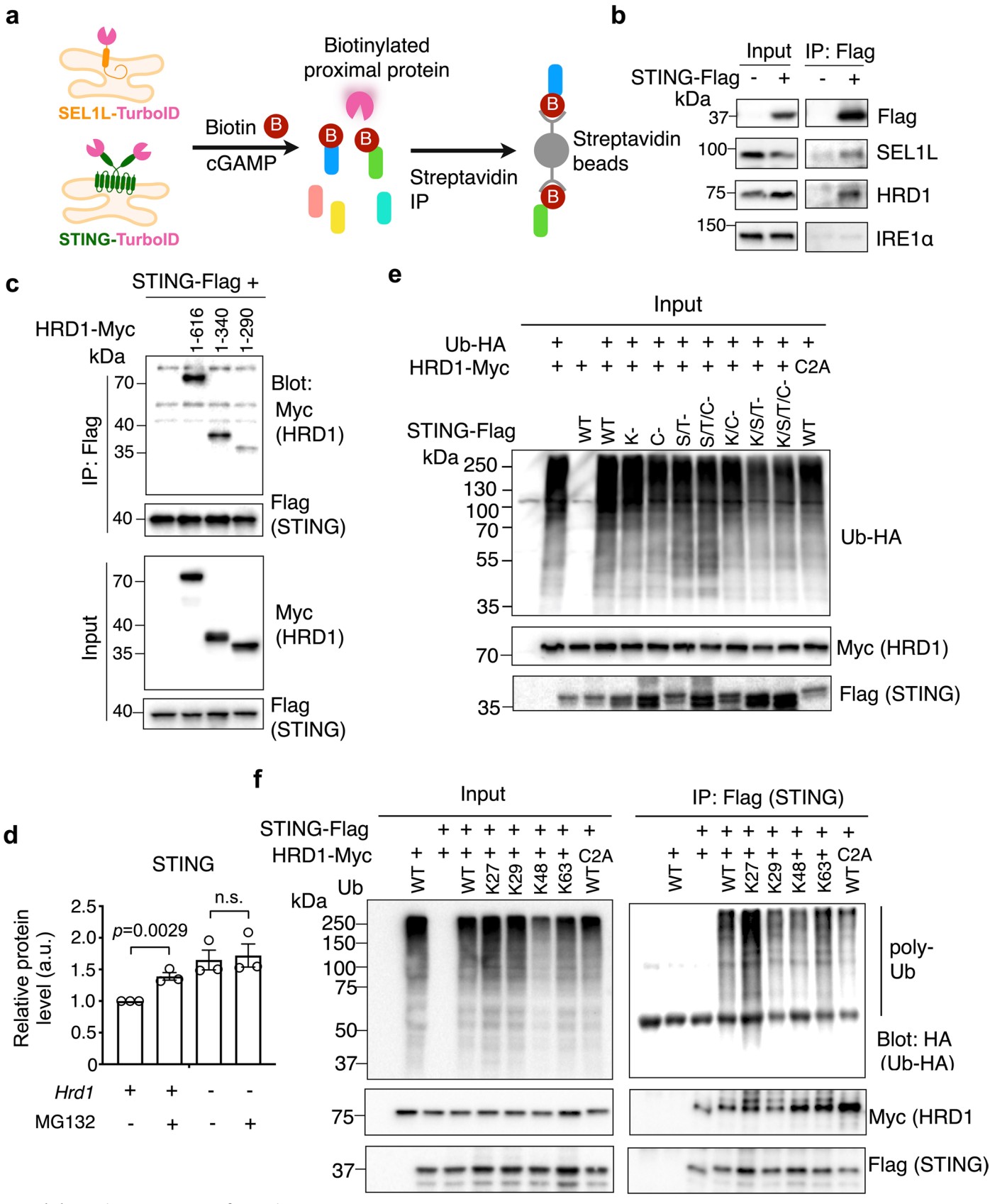

**Extended Data Fig. 6 | See next page for caption.**

**Extended Data Fig. 6 | STING interacts with and is ubiquitinated by SEL1L-HRD1 ERAD. (a)** Diagram for SEL1L- and STING-TurboID proximity labeling experiment. B, biotin. (**b**) Immunoblot analysis following immunoprecipitation of Flag in lysates of HEK293T cells transfected with STING-Flag. (**c**) Immunoblot analysis following immunoprecipitation of Flag from lysates of HEK293T cells transfected with STING-Flag and full length (1-616) or truncated HRD1-Myc plasmids. (**d**) Quantitation of STING levels in macrophages with or without MG132 treatment for 5 hr shown in Fig. 5h, n = 3, combined from three independent repeats. Values represent mean ± s.e.m. *P* values were determined by unpaired, two-tailed, Student's *t*-test; n.s., not significant. (**e**) Immunoblot analysis in lysates of HEK293T cells transfected with Ub-HA, WT- or C2A ligase-dead mutant HRD1-myc, and different STING mutant-Flag. STING K⁻, 8 cytosolic Lys mutated to Arg; C⁻, 9 cytosolic Cys to Ala; S/T⁻, 33 cytosolic Ser/Thr to Ala. This was the input for the immunoprecipitation experiment shown in Fig. 5j. (**f**) Immunoblot analysis following immunoprecipitation of STING-Flag in the lysates of HEK293T cells transfected with STING-Flag, HRD1-myc (WT and RING-dead C2A), and Ub-HA (WT and K-only mutants). Data are representative of at least 2 independent biological repeats (b, c, e, f).

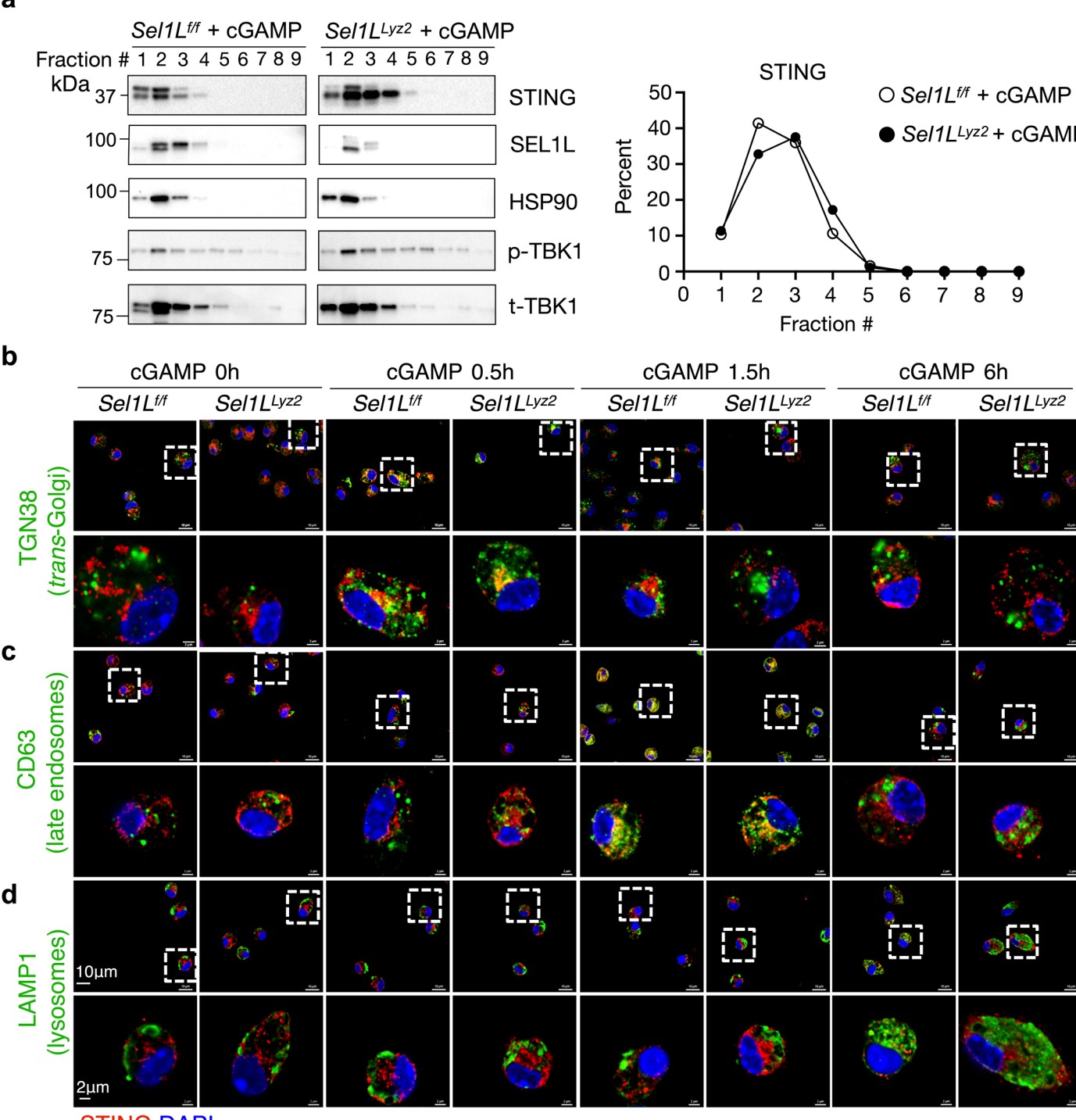

**Extended Data Fig. 7 | Distribution and trafficking of STING upon stimulation are unaffected by SEL1L-HRD1 ERAD.** (**a**) Sucrose gradient fractionation followed by immunoblot analysis in primary macrophages treated with cGAMP for 3 hr, with quantitation of percent of STING protein in each fraction on the right, representative of 2 independent repeats. (**b**–**d**) Representative confocal microscopic images of STING, co-stained with organellar markers such as TGN38 (trans-Golgi, b), CD63 (late endosome, c), LAMP1 (lysosome, d) in primary macrophages treated with or without cGAMP for indicated time points. Quantitation shown in Fig. 6f–h.

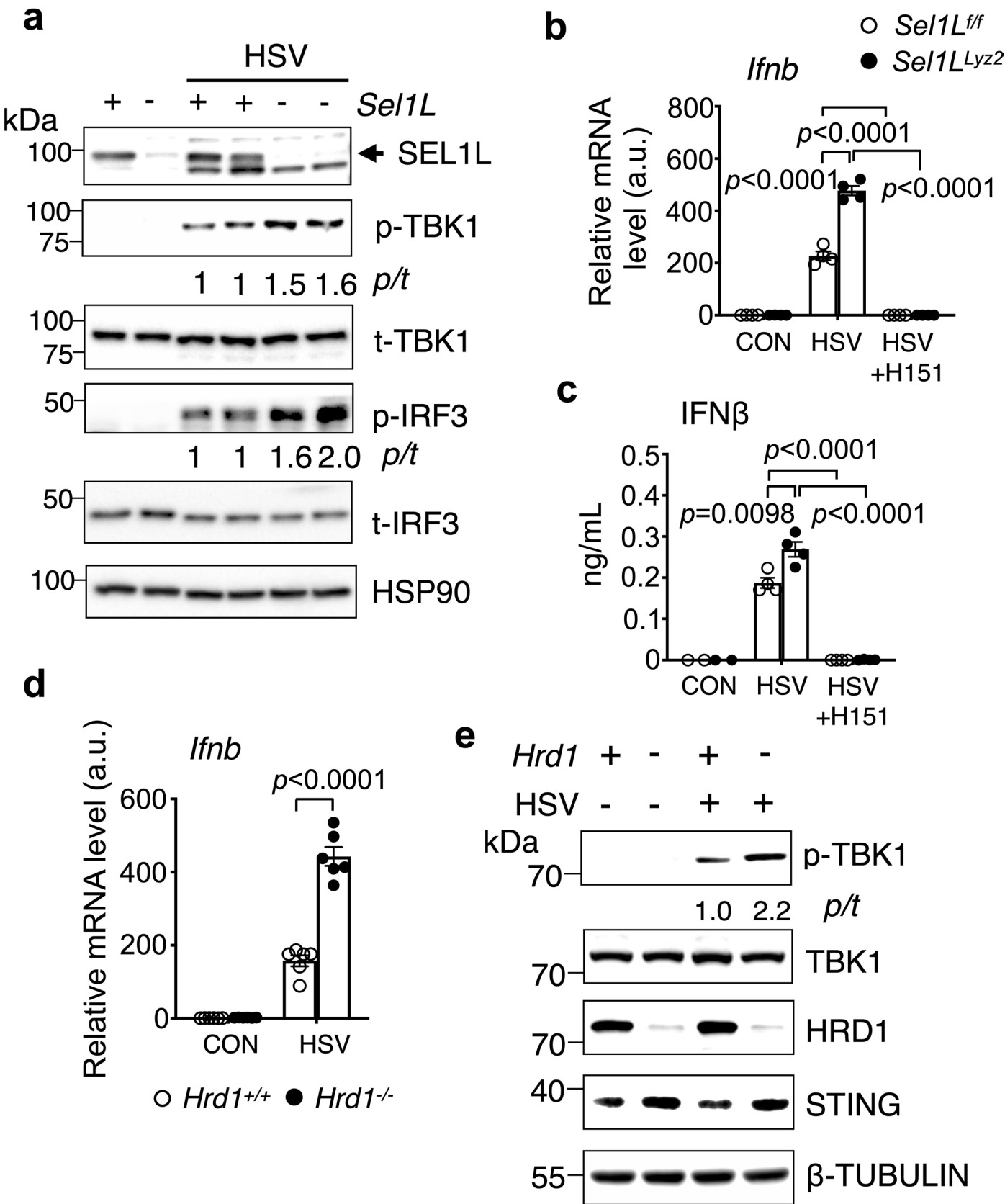

**Extended Data Fig. 8 | See next page for caption.**

**Extended Data Fig. 8 | SEL1L-HRD1 ERAD limits STING-mediated immunity against HSV-1 infection.** (**a**) Immunoblot of the STING pathway in primary macrophages treated with HSV-1 (MOI = 1) for 6 hr, with quantitation of the ratio of phosphorylated to total proteins (p/t) shown below the gels. (b-c) q-PCR (**b**) and ELISA (**c**) analyses of gene expression and secreted protein levels of IFN-β, respectively, in macrophages infected with HSV-1 (MOI = 5) for 6 hr, with or without the STING inhibitor H151 pretreatment. *n* = 4 mice each, combined from 2 independent repeats. (**d**) q-PCR analysis of *Ifnb* gene in *Hrd1*⁺/⁺ and *Hrd1*⁻/⁻ RAW 264.7 cells infected HSV-1 (MOI = 5) for 6 hr. *n* = 6 each, combined from 2 independent repeats. (**e**) Immunoblot showing TBK1 activation in *Hrd1*⁺/⁺ and *Hrd1*⁻/⁻ RAW 264.7 cells treated with HSV-1 (MOI = 5) for 6 hr, with the quantitation of the ratio of phosphorylated to total proteins (p/t) shown below the gel. Data are representative of two independent biological repeats (a, e). Values represent mean ± s.e.m. *P* values were determined by unpaired, two-tailed, Student's *t*-test (b-d).

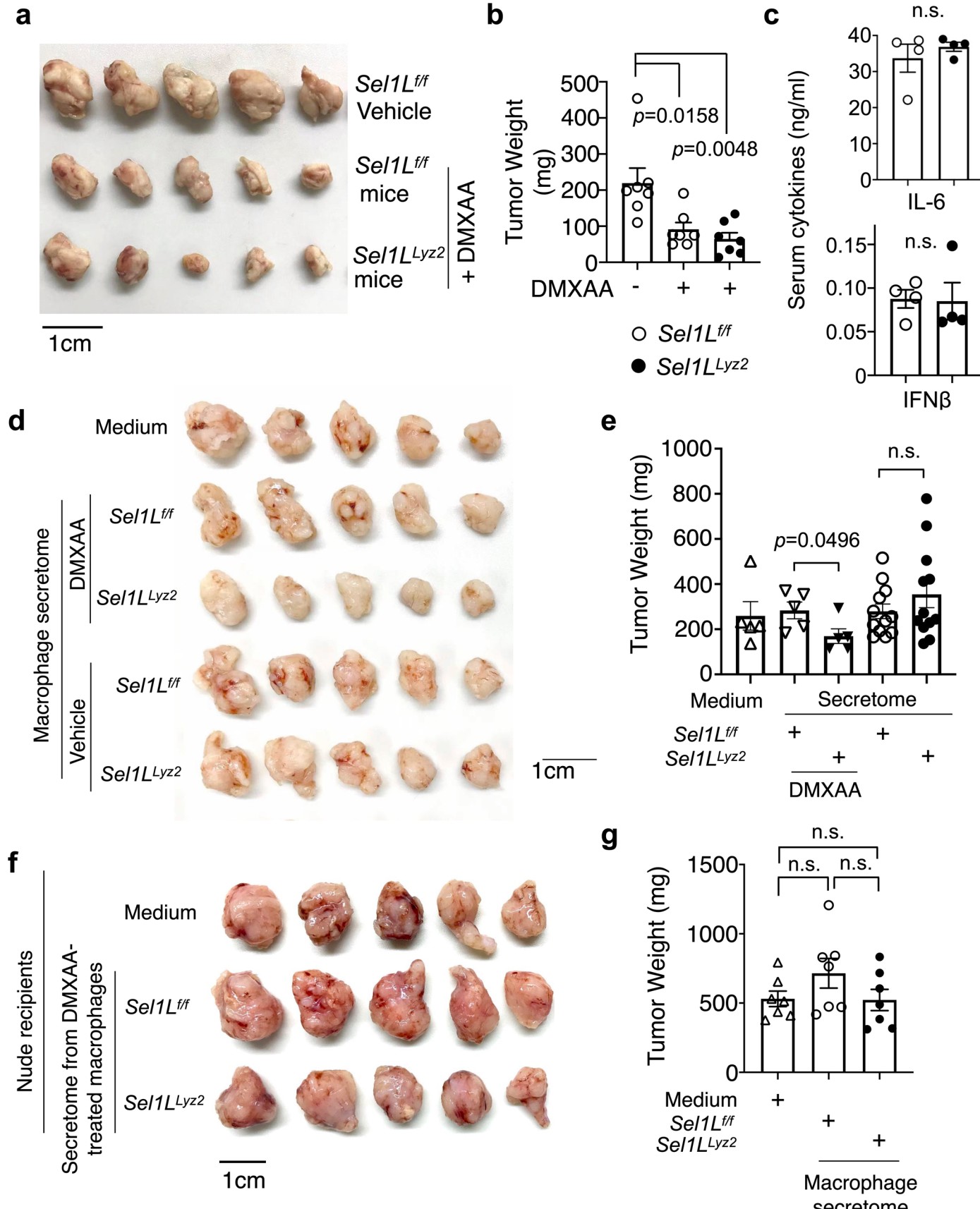

**Extended Data Fig. 9 | See next page for caption.**

**Extended Data Fig. 9 | SEL1L-HRD1 ERAD limits STING-mediated anti-tumor immunity. (a, b)** Representative images of tumors from tumor cell-transplanted *Sel1L^{f/f}* and *Sel1L^{Lyz2}* mice administered with vehicle or DMXAA (12.5 mg/kg body weight, *i.p.*) twice. Quantitation of tumor weights at the end of experiment shown in (b). n = 7 mice for each group, combined from 2 independent repeats. **(c)** ELISA analysis of serum cytokines in *Sel1L^{f/f}* and *Sel1L^{Lyz2}* mice 1.5 hr after DMXAA injection. n = 4 mice each, combined from 2 independent repeats. **(d, e)** Representative images of tumors of 5 groups of WT mice received medium or secretome from vehicle- or DMXAA-treated *Sel1L^{f/f}* and *Sel1L^{Lyz2}* macrophages.

Experiments were performed as described in Fig. 7f. Quantitation of tumor weights at the end of experiment shown in (e). n = 5, 5, 5, 12, 12 mice (left to right), combined from 2 independent repeats. **(f, g)** Representative images of tumors of 3 groups of tumor cell-transplanted nude mice received DMXAA-medium, secretomes from DMXAA-treated macrophages from *Sel1L^{f/f}* or *Sel1L^{Lyz2}* mice (f). Quantitation of tumor weights at the end of experiment shown in (g). n = 7 mice for each group, combined from 2 independent repeats. Values represent mean ± s.e.m. *P* values were determined by unpaired, two-tailed, Student's *t*-test (b, c, e, g); n.s., no significance.

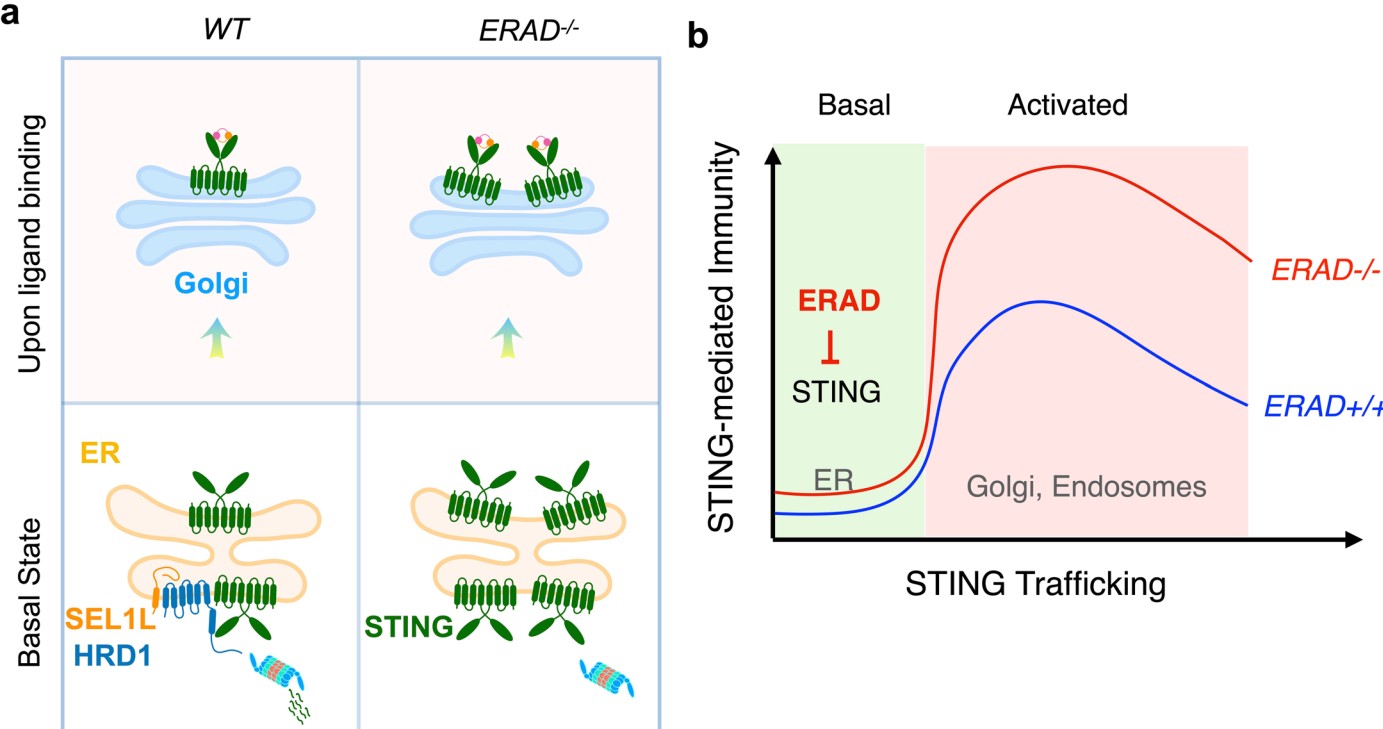

**Extended Data Fig. 10 | Proposed model for the regulation of STING-mediated immunity by SEL1L-HRD1 ERAD at the ER.** (**a**) Newly synthesized STING protein in the ER is subjected to SEL1L-HRD1 ERAD-mediated proteasomal degradation. In the absence of SEL1L-HRD1 ERAD, a larger pool of activable STING is present at the ER under basal state. (**b**) SEL1L-HRD1 ERAD limits the activation potential of STING signaling and immunity via controlling the abundance of STING protein at the ER under basal state.

# Reporting Summary

## Statistics

For all statistical analyses, confirm that the following items are present in the figure legend, table legend, main text, or Methods section.

| n/a | Confirmed | |
|---|---|---|
| ☐ | ☒ | The exact sample size (*n*) for each experimental group/condition, given as a discrete number and unit of measurement |
| ☐ | ☒ | A statement on whether measurements were taken from distinct samples or whether the same sample was measured repeatedly |
| ☐ | ☒ | The statistical test(s) used AND whether they are one- or two-sided *Only common tests should be described solely by name; describe more complex techniques in the Methods section.* |
| ☐ | ☒ | A description of all covariates tested |
| ☐ | ☒ | A description of any assumptions or corrections, such as tests of normality and adjustment for multiple comparisons |
| ☐ | ☒ | A full description of the statistical parameters including central tendency (e.g. means) or other basic estimates (e.g. regression coefficient) AND variation (e.g. standard deviation) or associated estimates of uncertainty (e.g. confidence intervals) |
| ☐ | ☒ | For null hypothesis testing, the test statistic (e.g. $F$, $t$, $r$) with confidence intervals, effect sizes, degrees of freedom and $P$ value noted *Give P values as exact values whenever suitable.* |
| ☒ | ☐ | For Bayesian analysis, information on the choice of priors and Markov chain Monte Carlo settings |
| ☒ | ☐ | For hierarchical and complex designs, identification of the appropriate level for tests and full reporting of outcomes |
| ☒ | ☐ | Estimates of effect sizes (e.g. Cohen's *d*, Pearson's *r*), indicating how they were calculated |

*Our web collection on statistics for biologists contains articles on many of the points above.*

## Software and code

Policy information about availability of computer code

| | |
|---|---|
| Data collection | Confocal data of STING localization were collected using NIS-elements C 4.60.00 (Nikon) and stedycon smart control (STEDYCON super-resolution microscopy); H&E images of liver and adipose tissue were collected using Aperio Imagescope software v102.0.4.6; Flow data were collected using the FACSDiva v6.2 software (BD Biosciences); Western Blot data were collected by Image Lab software 4.1 (Bio-rad); TEM images were collected with the use of JEM-1400 TEM. |
| Data analysis | Prism v6 and v8 was used for statistics analysis; Imaging data of STING colocalization were analyzed using the Fiji 2.0.0 software (ImageJ); Flow data were analyzed using the FACSDiva v6.2 software (BD Biosciences) and Flowjo 8.7 (Flowjo.com); Western Blot data were analyzed by Image Lab software 4.1 (Bio-rad). TEM images were analyzed with Image-Pro Plus 6.0 software. |

For manuscripts utilizing custom algorithms or software that are central to the research but not yet described in published literature, software must be made available to editors and reviewers. We strongly encourage code deposition in a community repository (e.g. GitHub). See the Nature Portfolio guidelines for submitting code & software for further information.

## Data

Policy information about availability of data

All manuscripts must include a data availability statement. This statement should provide the following information, where applicable:
- Accession codes, unique identifiers, or web links for publicly available datasets
- A description of any restrictions on data availability
- For clinical datasets or third party data, please ensure that the statement adheres to our policy

Previously published STING proximity-based proteomics data are available from [Chu, TT., Tu, X., Yang, K. et al. Nature (2021). https://doi.org/10.1038/s41586-021-03762-2]. All source data for all graphical and unprocessed blot images are provided with this study. All other data supporting the findings of this study are available from the corresponding author on reasonable request.

## Human research participants

Policy information about studies involving human research participants and Sex and Gender in Research.

| | |
|---|---|
| Reporting on sex and gender | N/A |
| Population characteristics | N/A |
| Recruitment | N/A |
| Ethics oversight | N/A |

Note that full information on the approval of the study protocol must also be provided in the manuscript.

# Field-specific reporting

Please select the one below that is the best fit for your research. If you are not sure, read the appropriate sections before making your selection.

☒ Life sciences ☐ Behavioural & social sciences ☐ Ecological, evolutionary & environmental sciences

For a reference copy of the document with all sections, see nature.com/documents/nr-reporting-summary-flat.pdf

# Life sciences study design

All studies must disclose on these points even when the disclosure is negative.

| | |
|---|---|
| Sample size | Based on sample size formula of the power analysis, $N=8(CV)2[1+(1-PC)2]/(PC)2$, to reach the error = 0.05, Power = 0.80, percentage change in means (PC) = 20%, co-efficient of variation (CV) = 10 ~ 15% (varies between the experiments), 4-6 mice per group are the minimal number of mice to obtain statistical significance and to ensure adequate power. The sample size for each animal experiment was described in the figure legend. The way of predetermining sample size is described in Supplementary materials page 1, "Power analysis of the animal size". No sample size calculation was performed for in vitro experiments, sample size of these experiments was determined based on previously published studies in the field of cell biology and immunology where differences were observed. Sample sizes were provided throughout manuscript. |
| Data exclusions | No animals or samples were excluded from the analysis. |
| Replication | All experiments were repeated at least twice or performed with independent samples. All attempts for replication were successful. This is described in text methods part, section "Statistical Analysis". The exact repeat times of experiments are indicated in the figure legends. |
| Randomization | For LPS injection, DMXAA injection, HFD feeding experiments, mice were sex- and age-matched and randomly assigned to experimental groups according to the genotype. For secretome treatment experiments, mice with similar tumor volume and body weights following tumor cell implantation were randomized into different groups of treatment. For ligands and chemical molecules treatment experiments, cell culture samples were randomly allocated to control and experimental groups. |
| Blinding | When experiments were done by one investigator, blindness is not applied or possible. When experiments are done sequentially by different investigators, investigators were blinded to allocation during experiments and outcome assessment. The experiments were repeated by different investigators independently. |

# Reporting for specific materials, systems and methods

We require information from authors about some types of materials, experimental systems and methods used in many studies. Here, indicate whether each material, system or method listed is relevant to your study. If you are not sure if a list item applies to your research, read the appropriate section before selecting a response.

## Materials & experimental systems

| n/a | Involved in the study |
|-----|----------------------|
| ☐ | ☒ Antibodies |
| ☐ | ☒ Eukaryotic cell lines |
| ☒ | ☐ Palaeontology and archaeology |
| ☐ | ☒ Animals and other organisms |
| ☒ | ☐ Clinical data |
| ☒ | ☐ Dual use research of concern |

## Methods

| n/a | Involved in the study |
|-----|----------------------|
| ☒ | ☐ ChIP-seq |
| ☐ | ☒ Flow cytometry |
| ☒ | ☐ MRI-based neuroimaging |

## Antibodies

**Antibodies used**

Flow cytometric antibodies for cell markers: CD4 (GK1.5, BioLegend 100408), CD8 (YTS169.4, Thermo Fisher MA5-17605, MA5-17607), F4/80 (BM8, BioLegend 123116, 123114), CD11b (M1/70, BioLegend 101206), Gr-1 (RB6-8C5, BioLegend 108408), TCR β (H57-597, BioLegend 109206), B220 (RA3-6B2, BioLegend 103206, 103208), CD45 (30-F11, BioLegend 103130), I-A/I-E (M5/114.15.2, BioLegend 107645, 107608), H-2Kb/H-2Db (28-8-6, AF6-88.5, BioLegend 114606, 116506), TLR2 (CB225, BioLegend 148604), TLR4 (SA15-21, BioLegend 145406), PD-L1 (10F.9G2, BioLegend 124308) and anti-CD16/CD32 (93, BioLegend 101302). All used at 1:100 or 200 dilution.

Western blot antibodies: HSP90 (Abcam ab13492, 1:6,000), β-Tubulin (Proteintech 10068-1-AP, 1:3000), Caspase-3 (Cell Signaling 8G10, 1:1,000), β-Actin (Proteintech 20536-1-AP, 1:3000), IκBα (Cell Signaling 9242, 1:2,000), SEL1L (Abcam ab78298, 1:1000), BiP (Abcam ab21685; 1:5000), HRD1 (Dr. Richard Wojcikiewicz, 1:300; Proteintech 13473-1-AP, 1:1000), STING (Proteintech 19851-1AP, 1:1500; Cell Signaling D2P2F; 1:2000), p-STING (Ser365) (Cell Signaling D8F4W; 1:2000), cGAS (Cell Signaling D3080; 1:2000), p-TBK1 (Ser172) (Cell Signaling D52C2; 1:1000), TBK1 (Cell Signaling E9H5S; 1:2000), p-IRF-3 (Ser396) (Cell Signaling D601M; 1:2000), IRF-3 (Cell Signaling D83B9; 1:2000), ATG7 (Cell Signaling D12B11; 1:1000), OS9 (Abcam ab109510; 1:3000), eIF2α (Cell Signaling 9722; 1:2000), p-eIF2α (Cell Signaling 3597S; 1:2000), IRE1α (Cell Signaling 3294; 1:3000), ERP44 (Cell Signaling 2886; 1:3000), STIM1 (Cell Signaling 4916; 1:2000), HA (SIGMA H3663; 1:2000), c-Myc (SIGMA C3956; 1:2000), Flag (SIGMA F1804; 1:2000), H2A (Cell Signaling 2578; 1:5000), LC3B (Cell Signaling 2775; 1:2000), PDI (ENZO ADI-SPA-890, 1:2,000), Ubiquitin (Santa Cruz P4D1, 1:200), SOAT1 (GeneTex GTX32890, 1:1,000), FACL4 (Abcam ab155282, 1:1,000), Calnexin (Proteintech 10427-2-AP, 1:20,000), goat anti-rabbit IgG-HRP (BioRad 1721019, 1:5,000) and goat anti-mouse IgG-HRP (BioRad 1721011, 1:5,000).

Antibodies for immunofluorescent staining: STING (Proteintech 19851-1AP, 1:200), KDEL (Abcam MAC 256, 1:200), Phospho-STING (Ser365) (Cell Signaling D1C4T, 1:200), TGN38 (Santa Cruz sc-166594,1:200), CD63 (Santa Cruz sc-5275,1:200), LAMP1 (DSHB 1D4B, 1:50).

Antibodies for immunoprecipitation: STING (Proteintech 19851-1AP), SEL1L (Abcam ab78298), protein A-agarose beads (Invitrogen 20334), agarose-conjugated anti-FLAG (Sigma A4596), agarose-conjugated anti-Myc (Sigma 16-219) , streptavidin agarose (Thermo Fisher 20353). 1μg antibody or 30 μl agarose beads for 1mL sample lysis.

**Validation**

Antibodies used in this study were from best available vendors with good citation and validated by the vendors. We further verified specificity busing Western blot, IP and confocal imaging, compared with isotype controls as well as positive and negative control samples. The information of validation and citation are available on the manufacturer's websites.

1) Mouse anti-HSP90 (ab13492/ Abcam, 1:6,000): The manufacture states that the specificity of the antibody was tested by western blot on various mammalian cell lysates. https://www.abcam.com/hsp90-antibody-ac88-ab13492.html#lb

2) Rabbit β-Tubulin (Proteintech 10068-1-AP, 1:3000): The manufacture states that the specificity of the antibody was tested by western blot on various mammalian cell lysates. https://www.ptglab.com/products/TUBB3-Antibody-10068-1-AP.htm

3) Rabbit anti- Caspase-3 (Cell Signaling 9665, 1:1,000): The manufacture states that the specificity of the antibody was tested by western blot on HeLa cell lysates. https://www.cellsignal.com/product/productDetail.jsp?productId=9665

4) Rabbit anti- β-Actin (Proteintech 20536-1-AP, 1:3000): The manufacture states that the specificity of the antibody was tested by western blot on various mammalian cell lysates and tissues. https://www.ptglab.com/products/ACTB-Antibody-20536-1-AP.htm

5) Rabbit anti-IκBα (Cell Signaling 9242, 1:2,000): The manufacture states that the specificity of the antibody was tested by western blot on HeLa cell lysates. https://www.cellsignal.com/products/primary-antibodies/ikba-antibody/9242?_=1673953782135&Ntt=9242&tahead=true

6) Rabbit anti- SEL1L (Abcam ab78298, 1:1000; Abclonal A12073, 1:2000): The manufacture states that the specificity of the antibody was tested by western blot on various mammalian cell lysates and tissues. https://www.abcam.com/sel1l-antibody-ab78298.html#lb, https://abclonal.com/search/index?keyword=SEL1L+Rabbit +pAb&catid=56&__hash__=d54593da28f3677abb7adc59204265a1_e4843318fa4fda248241fe6564807c09&Searchbar=

7) Rabbit anti- BiP (Abcam ab21685; 1:5000): The manufacture states that the specificity of the antibody was tested by western blot on various mammalian cell lysates. https://www.abcam.com/grp78-bip-antibody-ab21685.html#lb

8) Rabbit anti-HRD1 (Proteintech 13473-1-AP, 1:1000): The manufacture states that the specificity of the antibody was tested by western blot on various mammalian cell lysates and tissues. https://www.ptglab.com/products/SYVN1-Antibody-13473-1-AP.htm

9) Rabbit anti- STING (Proteintech 19851-1AP, 1:1500 for WB): The manufacture states that the specificity of the antibody was tested by western blot on various mammalian cell lysates and tissues. https://www.ptglab.com/products/TMEM173-Antibody-19851-1-AP.htm

10) Rabbit anti-p-STING (Ser365) (Cell Signaling 72971 clone D8F4W, 1:2000): The manufacture states that the specificity of the antibody was tested by western blot on Raw 264.7 cells. https://www.cellsignal.com/products/primary-antibodies/phospho-sting-

ser365-d8f4w-rabbit-mab/72971?site-search-type=Products&N=4294956287&Ntt=d8f4w&fromPage=plp&_requestid=10805356

11) Rabbit anti-cGAS (Cell Signaling 31659 clone D3080; 1:2000): The manufacture states that the specificity of the antibody was tested by western blot on various mammalian cell lysates. https://www.cellsignal.com/products/primary-antibodies/cgas-d3o8o-rabbit-mab-mouse-specific/31659?site-search-type=Products&N=4294956287&Ntt=cgas&fromPage=plp

12) Rabbit anti- p-TBK1 (Ser172) (Cell Signaling 5483 clone D52C2; 1:1000): The manufacture states that the specificity of the antibody was tested by western blot on THP-1 cell lysates. https://www.cellsignal.com/products/primary-antibodies/phospho-tbk1-nak-ser172-d52c2-xp-rabbit-mab/5483?site-search-type=Products&N=4294956287&Ntt=d52c2&fromPage=plp

13) Validation of Rabbit anti- TBK1 (Cell Signaling 51872 clone E9H5S; 1:2000): The manufacture states that the specificity of the antibody was tested by western blot on various mammalian cell lysates. https://www.cellsignal.com/products/primary-antibodies/tbk1-nak-e9h5s-mouse-mab/51872?site-search-type=Products&N=4294956287&Ntt=e9h5s&fromPage=plp&_requestid=10806496

14) Validation of Rabbit anti- p-IRF-3 (Ser396) (Cell Signaling 29047 clone D601M; 1:2000): The manufacture states that the specificity of the antibody was tested by western blot on Human cell lysates and mice. https://www.cellsignal.com/products/primary-antibodies/phospho-irf-3-ser396-d6o1m-rabbit-mab/29047

15) Validation of Rabbit anti- IRF-3 (Cell Signaling 4302 clone D83B9; 1:2000): The manufacture states that the specificity of the antibody was tested by western blot on various mammalian cell lysates. https://www.cellsignal.com/products/primary-antibodies/irf-3-d83b9-rabbit-mab/4302?site-search-type=Products&N=4294956287&Ntt=anti-+irf-3+&fromPage=plp

16) Validation of Rabbit anti- ATG7 (Cell Signaling 8558 clone D12B11; 1:1000): The manufacture states that the specificity of the antibody was tested by western blot on various mammalian cell lysates. https://www.cellsignal.com/products/primary-antibodies/atg7-d12b11-rabbit-mab/8558?site-search-type=Products&N=4294956287&Ntt=atg7&fromPage=plp

17) Validation of Rabbit anti- OS9 (Abcam ab109510; 1:3000): The manufacture states that the specificity of the antibody was tested by western blot on various mammalian cell lysates. https://www.abcam.com/os9-antibody-epr42722-ab109510.html

18) Validation of Rabbit anti- eIF2α (Cell Signaling 9722; 1:2000): The manufacture states that the specificity of the antibody was tested by western blot on PC12 cell lysates. https://www.cellsignal.com/products/primary-antibodies/eif2a-antibody/9722

19) Validation of Rabbit anti- p-eIF2α (Cell Signaling 3597; 1:2000): The manufacture states that the specificity of the antibody was tested by western blot on various mammalian cell lysates. https://www.cellsignal.com/products/primary-antibodies/phospho-eif2a-ser51-119a11-rabbit-mab/3597?_=1673956082958&Ntt=3597S&tahead=true

20) Validation of Rabbit anti- IRE1α (Cell Signaling 3294 clone 14C10; 1:3000): The manufacture states that the specificity of the antibody was tested by western blot on various mammalian cell lysates. https://www.cellsignal.com/products/primary-antibodies/ire1a-14c10-rabbit-mab/3294

21) Validation of Rabbit anti- ERP44 (Cell Signaling 2886; 1:3000): The manufacture states that the specificity of the antibody was tested by western blot on various mammalian cell lysates. https://www.cellsignal.com/products/primary-antibodies/erp44-antibody/2886

22) Validation of Rabbit anti- STIM1 (Cell Signaling 4916; 1:2000): The manufacture states that the specificity of the antibody was tested by western blot on various mammalian cell lysates. https://www.cellsignal.com/products/primary-antibodies/stim1-antibody/4916

23) Validation of Mouse anti- HA (Sigma H3663; 1:2000): The manufacture states that the specificity of the antibody was tested by western blot on HEK-293T cell lysates. https://www.sigmaaldrich.com/US/en/product/sigma/h3663

24) Validation of Rabbit anti- c-Myc (Sigma C3956; 1:2000): The manufacture states that the specificity of the antibody was tested by western blot on HEK-293T cell lysates. https://www.sigmaaldrich.com/US/en/product/sigma/c3956

25) Validation of Mouse anti- Flag (Sigma F1804 clone M2; 1:2000): The manufacture states that the antibody has been optimized for signal band detection of the FLAG antigen (DYKDDDK) fused proteins in mammalian, plant and bacterial systems. https://www.sigmaaldrich.com/US/en/product/sigma/f1804

26) Validation of Rabbit anti- H2A (Cell Signaling 2578; 1:5000): The manufacture states that the specificity of the antibody was tested by western blot on various mammalian cell lysates. https://www.cellsignal.com/products/primary-antibodies/histone-h2a-antibody-ii/2578

27) Validation of Rabbit anti- LC3B (Cell Signaling 2775; 1:2000): The manufacture states that the specificity of the antibody was tested by western blot on various mammalian cell lysates. https://www.cellsignal.com/products/primary-antibodies/lc3b-antibody/2775

28) Validation of Rabbit anti- PDI (ENZO ADI-SPA-890, 1:2,000): The manufacture states that the specificity of the antibody was tested by western blot on various mammalian cell lysates and tissues. https://www.enzolifesciences.com/ADI-SPA-890/pdi-polyclonal-antibody/

29) Validation of Mouse anti- Ubiquitin (Santa Cruz sc8017 clone P4D1, 1:200): The manufacture states that the specificity of the antibody was tested by western blot on various human cell lysates. https://www.scbt.com/p/ubiquitin-antibody-p4d1?requestFrom=search

30) Validation of Rabbit anti- SOAT1 (GeneTex GTX32890, 1:1,000): The manufacture states that the specificity of the antibody was tested by western blot on various mammalian cell lysates and tissues. https://www.genetex.com/Product/Detail/SOAT1-antibody/GTX32890#datasheet

31) Validation of Rabbit anti- FACL4 (Abcam ab155282, 1:1,000): The manufacture states that the specificity of the antibody was tested by western blot on various mammalian cell lysates and tissues. https://www.abcam.com/facl4-antibody-epr8640-ab155282.html#lb

32) Validation of Rabbit anti- Calnexin (Proteintech 10427-2-AP, 1:20,000): The manufacture states that the specificity of the antibody was tested by western blot on various mammalian cell lysates. https://www.ptglab.com/products/CANX-Antibody-10427-2-AP.htm

33) Validation of Rabbit anti- Phospho-STING (Ser365) (Cell Signaling 62912 clone D1C4T, 1:200): The manufacture states that the specificity of the antibody was tested by Confocal immunofluorescent analysis of Raw 264.7 cells, transfected with poly(dA:dT) or mock transfected. https://www.cellsignal.com/products/primary-antibodies/phospho-sting-ser365-d1c4t-rabbit-mab/62912?site-search-type=Products&N=4294956287&Ntt=phospho-sting+&fromPage=plp

34) Validation of Mouse anti- TGN38 (Santa Cruz sc-166594,1:200): The manufacture states that the specificity of the antibody was tested by Confocal immunofluorescent analysis and western blot on various mammalian cell. https://www.scbt.com/p/tgn38-antibody-b-6?requestFrom=search

35) Validation of Mouse anti- CD63 (Santa Cruz sc-5275,1:200): The manufacture states that the specificity of the antibody was tested by western blot on various mammalian cell. https://www.scbt.com/p/cd63-antibody-mx-49-129-5?requestFrom=search

36) Validation of Mouse anti- LAMP1 (DSHB 1D4B, 1:50): The manufacture states that the specificity of the antibody was tested by western blot on various mammalian cell. https://dshb.biology.uiowa.edu/1D4B

37) Validation of Mouse agarose-conjugated anti-FLAG (Sigma A4596): The manufacture states that the antibody specifically detected free N-terminal of FALG sequence(N-Asp-Tyr-Lys-Asp-Asp-Asp-Asp-Lys-C). https://www.sigmaaldrich.com/US/en/product/

sigma/a4596

38) Validation of Mouse agarose-conjugated anti-Myc (Sigma 16-219 clone 4A6): The manufacture states that the antibody specifically detected Myc antigen (MEQKLISEEDL). https://www.sigmaaldrich.com/US/en/product/mm/16219

38) Validation of PE anti- Mouse CD4 (BioLegend 100408 clone GK1.5): The manufacture states that the antibody was tested by Flow cytometric analysis of C57BL/6 mouse splenocytes. https://www.biolegend.com/en-us/products/fitc-anti-mouse-cd4-antibody-248

39) Validation of anti- Rat CD8 (Thermo Fisher MA5-17605 clone YTS169.4): The manufacture states that the antibody was tested by Flow cytometric analysis of Mouse Thymus cells. https://www.thermofisher.cn/cn/zh/antibody/product/CD8-alpha-Antibody-clone-YTS169-4-Monoclonal/MA5-17605

40) Validation of anti-Rat F4/80 Antibody (BioLegend 123116, 123114 clone BM8): The manufacture states that the antibody was tested by Flow cytometric analysis of thioglycolate-elicited BALB/c mouse peritoneal macrophages. https://www.biolegend.com/en-us/products/pe-cyanine7-anti-mouse-f4-80-antibody-4070

41) Validation of anti-Rat CD11b (BioLegend 101206 clone M1/70): The manufacture states that the antibody was tested by Flow cytometric analysis of C57BL/6 mouse bone marrow cells. https://www.biolegend.com/en-us/products/fitc-anti-mouse-human-cd11b-antibody-347

42) Validation of anti-Rat Gr-1 (BioLegend 108408 clone RB6-8C5): The manufacture states that the antibody was tested by Flow cytometric analysis of C57BL/6 mouse bone marrow cells. https://www.biolegend.com/en-us/products/pe-anti-mouse-ly-6g-ly-6c-gr-1-antibody-460

43) Validation of anti-Armenian Hamster TCR β (BioLegend 109206 clone H57-597): The manufacture states that the antibody was tested by Flow cytometric analysis of C57BL/6 mouse splenocytes. https://www.biolegend.com/en-us/products/fitc-anti-mouse-tcr-beta-chain-antibody-270

44) Validation of anti-Rat B220 (BioLegend 103206, 103208 clone RA3-6B2): The manufacture states that the antibody was tested by Flow cytometric analysis of C57BL/6 mouse splenocytes. https://www.biolegend.com/en-us/products/fitc-anti-mouse-human-cd45r-b220-antibody-445, https://www.biolegend.com/en-us/products/pe-anti-mouse-human-cd45r-b220-antibody-447

45) Validation of anti-Rat CD45 (BioLegend 103130 clone 30-F11): The manufacture states that the antibody was tested by Flow cytometric analysis of C57BL/6 mouse splenocytes. https://www.biolegend.com/en-us/products/percp-anti-mouse-cd45-antibody-4265

46) Validation of anti-Rat I-A/I-E (BioLegend 107645, 107608 clone M5/114.15.2): The manufacture states that the antibody was tested by Flow cytometric analysis of C57BL/6 mouse splenocytes. https://www.biolegend.com/en-us/products/brilliant-violet-785-anti-mouse-i-a-i-e-antibody-12087, https://www.biolegend.com/en-us/products/pe-anti-mouse-i-a-i-e-antibody-367

47) Validation of anti-Mouse H-2Kb/H-2Db (BioLegend 114606 clone 28-8-6; 116506 clone AF6-88.5): The manufacture states that the antibody was tested by Flow cytometric analysis of C57BL/6 mouse splenocytes. https://www.biolegend.com/en-us/products/fitc-anti-mouse-h-2kb-h-2db-antibody-1683; https://www.biolegend.com/en-us/products/fitc-anti-mouse-h-2kb-antibody-1748

48) Validation of anti-Rat TLR2 (BioLegend 148604 clone CB225): The manufacture states that the antibody was tested by Flow cytometric analysis of thioglycolate-elicited BALB/c mouse peritoneal macrophages. https://www.biolegend.com/en-us/products/pe-anti-mouse-cd282-tlr2-antibody-10230

49) Validation of anti-Rat TLR4 (BioLegend 145406 clone SA15-21): The manufacture states that the antibody was tested by Flow cytometric analysis of thioglycolate-elicited BALB/c mouse peritoneal macrophages. https://www.biolegend.com/en-us/products/apc-anti-mouse-cd284-tlr4-antibody-8871

50) Validation of anti-Rat PD-L1 (BioLegend 124308 clone 10F.9G2): The manufacture states that the antibody was tested by Flow cytometric analysis of C57BL/6 mouse splenocytes. https://www.biolegend.com/en-us/products/pe-anti-mouse-cd274-b7-h1-pd-l1-antibody-4497

51) Validation of anti-Rat CD16/CD32 (BioLegend 101302 clone 93): The manufacture states that the antibody was tested by Flow cytometric analysis of C57BL/6 mouse splenocytes. https://www.biolegend.com/en-us/products/purified-anti-mouse-cd16-32-antibody-190

# Eukaryotic cell lines

Policy information about cell lines and Sex and Gender in Research

| | |
|---|---|
| Cell line source(s) | MEF and RAW 264.7 cell lines were originally obtained from ATCC; DN32.D3 cell line was a gift from Dr. Mitchell Kronenberg (La Jolla Institute for Immunology); PDAC cell line was a gift from Dr. Raghu Kalluri (MD Anderson Cancer Center, Houston, TX, USA). Vero Cells were provided by Dr. Malini Raghavan at University of Michigan Medical School. |
| Authentication | The cells have been authenticated by morphology. Expression and secretion of inflammatory cytokines upon ligand stimulation was confirmed for MEF, RAW 264.7 and DN32.D3. PDAC cells form tumors in mice with expected tumor morphology and characteristics. High HSV-1 susceptibility of Vero Cells were confirmed, consistent with absence of type I IFN in this cell line. Authentication was done before the experiments during this study. |
| Mycoplasma contamination | All cell lines used for experiments had no mycoplasma contamination after testing. |
| Commonly misidentified lines (See ICLAC register) | These cell lines are not listed in that database. |

# Animals and other research organisms

Policy information about studies involving animals; ARRIVE guidelines recommended for reporting animal research, and Sex and Gender in Research

| | |
|---|---|
| Laboratory animals | All genetically engineered mice were in C57BL/6J background. Nude mice are on the BALB/c background. 8-week-old male Sel1LLyz2 and Sel1Lf/f mice were used for high-fat diet feeding for up to 20 weeks. For in vivo tumor study, 6-8 week-old male mice were used for tumor cell implantation. For LPS Challenge in vivo, 8 w-old female Sel1LLyz2 and Sel1Lf/f mice were used. For collection of primary macrophage and T cells for in vitro study, 2-4 months-old, male and female Sel1LLyz2, Sel1Lf/f, Atg7Lyz2, Atg7f/f and OT1 |

mice were used. The background, gender and age of mice are specifically indicated in the text and legends.

Wild animals
N/A

Reporting on sex
Both males and females were used in this study, and results are applicable for both genders.

Field-collected samples
No field-collected samples used in this study.

Ethics oversight
All animal procedures were approved by and done in accordance with the Institutional Animal Care and Use Committee (IACUC) at the University of Michigan Medical School (PRO00008989) and Cornell University (#2007-0051), and Animal Experimentation Ethics Committee of the First Affiliated Hospital, Zhejiang University School of Medicine (#2021-0135).

Note that full information on the approval of the study protocol must also be provided in the manuscript.

# Flow Cytometry

## Plots

Confirm that:

☒ The axis labels state the marker and fluorochrome used (e.g. CD4-FITC).

☒ The axis scales are clearly visible. Include numbers along axes only for bottom left plot of group (a 'group' is an analysis of identical markers).

☒ All plots are contour plots with outliers or pseudocolor plots.

☒ A numerical value for number of cells or percentage (with statistics) is provided.

## Methodology

Sample preparation
Following incubation with anti-CD16/CD32 antibody to block Fc receptors, 1 million cells were incubated with 20 µl of antibodies diluted at optimal concentrations (at 1:100 or 200) for 20 min at 4°C. Cells were washed three times with PBS and then resuspended in 200 µl PBS for analysis. For intracellular staining, cells were fixed after surface staining and permeabilized with a BD Cytofix/Cytoperm fixation/permeabilization kit according to the manufacturer's protocol. Purification of stromal vascular cells (SVC) from epididymal fat pads followed by flow cytometric analysis were performed as previously described (80).
The information is detailed in the methods under the section of 'Flow cytomytric analysis of cell surface markers''.

Instrument
Samples were collected using BD LSR cell analyzer.

Software
Sample are collected by using the FACSDiva v6.2 software (BD Biosciences) and data are analyzed with using FACSDiva v6.2 and Flowjo 8.7 (Flowjo.com).

Cell population abundance
At least 10,000  live splenocytes and 100,000 stromal vascular cells were collected for FACS analysis. Relevant cells sorted from mouse  were re-run on Flow cytometry to ensure purity.

Gating strategy
Gates and boundaries were clearly visible in the figures.

☒ Tick this box to confirm that a figure exemplifying the gating strategy is provided in the Supplementary Information.

