## [Peer Review File · Nature Cell Biology]

Peer Review Information

Journal: Nature Cell Biology

Manuscript Title: SEL1L-HRD1 endoplasmic reticulum-associated degradation controls STING-mediated innate immunity by limiting the size of activable STING pool

Corresponding author name(s): Dr Ling Q

Reviewer Comments & Decisions:

Decision Letter, initial version:

*Please delete the link to your author homepage if you wish to forward this email to co-authors.

Dear Dr Qi,

I apologize for the delay. Your manuscript, "Endoplasmic reticulum-associated degradation controls STING-mediated innate immunity by limiting the size of activable STING pool", has now been seen by 3 referees, who are experts in ERAD (referee 1); UPR (referee 2); and cGAS/STING (referee 3). As you will see from their comments (attached below) they find this work of potential interest, but have raised substantial concerns, which in our view would need to be addressed with considerable revisions before we can consider publication in Nature Cell Biology.

Nature Cell Biology editors discuss the referee reports in detail within the editorial team, including the chief editor, to identify key referee points that should be addressed with priority, and requests that are overruled as being beyond the scope of the current study. To guide the scope of the revisions, I have listed these points below. We are committed to providing a fair and constructive peer-review process, so please feel free to contact me if you would like to discuss any of the referee comments further.

In particular, it would be essential to:

- A) Clarify the regulatory relationship between protein levels of STING and SEL1L (all Reviewers)
- B) Assess the involvement of, and potential effects on, other ER protein interactors that may be affected by SEL1L/HRD1 perturbation (Reviewers #1 and #2)
- C) Test the role if any, of autophagy in controlling STING levels (Reviewer #3)
- D) Examine the physiological impact of the Hrd1-STING axis which includes effects on the antiviral (Reviewers #2 and #3) and anti-tumor (Reviewer #3) responses. We will note that although addition of further mouse data along these lines would add to the study, a new mouse model would not be strictly required.
- E) All other referee concerns pertaining to strengthening existing data, providing controls, methodological details, clarifications and textual changes, should also be addressed.
- F) Finally please pay close attention to our guidelines on statistical and methodological reporting (listed below) as failure to do so may delay the reconsideration of the revised manuscript. In particular please provide:

We would be happy to consider a revised manuscript that would satisfactorily address these points, unless a similar paper is published elsewhere, or is accepted for publication in Nature Cell Biology in the meantime.

- ensure that it conforms to our format instructions and publication policies (see below and <https://www.nature.com/nature/for-authors>).

- provide a point-by-point rebuttal to the full referee reports verbatim, as provided at the end of this letter.

- provide the completed Reporting Summary (found here <https://www.nature.com/documents/nr-reporting-summary.pdf>). This is essential for reconsideration of the manuscript will be available to editors and referees in the event of peer review. For more information see <http://www.nature.com/authors/policies/availability.html> or contact me.

When submitting the revised version of your manuscript, please pay close attention to our [href="https://www.nature.com/nature-portfolio/editorial-policies/image-integrity">Digital Image Integrity Guidelines](https://www.nature.com/nature-portfolio/editorial-policies/image-integrity). and to the following points below:

Nature Cell Biology is committed to improving transparency in authorship. As part of our efforts in this direction, we are now requesting that all authors identified as 'corresponding author' on published papers create and link their Open Researcher and Contributor Identifier (ORCID) with their account on the Manuscript Tracking System (MTS), prior to acceptance. ORCID helps the scientific community achieve unambiguous attribution of all scholarly contributions. You can create and link your ORCID from the home page of the MTS by clicking on 'Modify my Springer Nature account'. For more information please visit www.springernature.com/orcid.

This journal strongly supports public availability of data. Please place the data used in your paper into a public data repository, or alternatively, present the data as Supplementary Information. If data can only be shared on request, please explain why in your Data Availability Statement, and also in the correspondence with your editor. Please note that for some data types, deposition in a public repository is mandatory - more information on our data deposition policies and available repositories appears below.

[Redacted]

We would like to receive a revised submission within six months.

We hope that you will find our referees' comments, and editorial guidance helpful. Please do not hesitate to contact me if there is anything you would like to discuss.

Best wishes,

Daryl Jason David

Daryl J.V. David, PhD

Senior Editor, Nature Cell Biology
Consulting Editor, Nature Communications
Nature Portfolio

Heidelberger Platz 3, 14197 Berlin, Germany
Email: daryl.david@nature.com
ORCID: <https://orcid.org/0000-0002-9253-4805>

Reviewers' Comments:

Reviewer #1:

Remarks to the Author:

Ji and colleagues investigate a potential role of the HRD1 ERAD complex in innate immunity by generating macrophage-specific KO mice for SEL1L (SEL1LLyz2), a key subunit of the Hrd1 complex.

It is shown that SEL1LLyz2 mice appeared normal and displayed normal repertoires of macrophages as well as other immune cells. SEL1LLyz2 macrophages appeared to display normal cell surface levels of MHC I and II and to respond normally to Toll-like receptor and RIG-I stimulation. However, STING stimulation with cGAMP resulted in increased production of various cytokines in SEL1LLyz2 macrophages. Consistent with the enhanced response, STING steady state protein levels were higher both in SEL1LLyz2 macrophages and in a macrophage cell line deficient for HRD1. Biochemical and proximity proteomics approaches, indicated that HRD1 complex interacts and ubiquitylates STING. The interaction appears to be mediated by the cytosolic regions of both STING and HRD1. Finally, the physiological relevance of increased STING basal levels was tested in two distinct paradigms, following a herpes simplex virus infection and in anti-tumor response in to KRAS-driven tumor implants.

Based on these findings, it is suggested that the basal pool of ER STING available for activation is regulated by the HRD1 ERAD complex. A model in which "ligand binding triggers the dissociation between SEL1L-HRD1 ERAD and STING leading to the ER-exit and activation of STING protein" is put forward. This model suggests that loss of HRD1 increases STING basal levels and results in enhanced STING signaling upon stimulation.

The manuscript focuses on an interesting topic, it is clearly written and most of the data appears of good quality, even if it is often overinterpreted. It is also convincingly shown that the levels of STING are post-translationally regulated by the HRD/SEL1L complex. On the negative side, the evidence for the model proposed is weak (see below). In addition, the physiological importance of STING regulation by the HRD1 complex is unclear. The differences observed in the HSV-1 model are subtle while the phenotypes in the tumor restriction model are detected only if a strong STING activator (DMXAA) is applied exogenously.

Major points

1- The increase in STING levels in SEL1L and HRD1 deficient cells is clear and convincing. However, it is less clear how this occurs and some of the observations are confusing and paradoxical. Based on cycloheximide chase experiments, the turnover of endogenous STING is slow and hardly affected by

HRD1 depletion (Fig 3F). Perhaps pulse-labeling experiments would be useful to capture small differences that are difficult to see by cycloheximide chase (IRE1a would serve as a good control in these experiments). On the other hand, STING is robustly ubiquitinated in a Hrd1-dependent fashion (Fig 5H-J). Surprisingly, under these conditions it is stated that ubiquitinated STING is degraded so quickly that the modified species can only be captured upon proteasome inhibition (lines 314-316). However, it is puzzling that such quick degradation does not appear to impact on total STING levels (Fig 5H, anti-STING blot). It would be important to reconcile these observations.

2- Immunoprecipitation experiments suggest an interaction between SEL1L and STING (Fig 5B-C). These data would be strengthened by the inclusion of appropriate specificity controls, ideally an abundant ER membrane protein rather than TBK1, a soluble protein localized to a different cellular compartment, or cGAMP, which triggers STING to traffic out of the ER. Interactions between STING and HRD1/SEL1L were also detected using proximity biotinylation. These were also lost upon STING stimulation by cGAMP. Like the immunoprecipitations, these experiments also lack specificity controls remaining unclear if all/most ER membrane proteins are labelled under these conditions. It seems that the Turbo-ID experiments are reporting on STING ER localization and not so much on its association with the HRD1 complex. Figure 5G shows an interesting experiment suggesting that HRD1/STING interaction requires their cytosolic domains, in particular HRD1 RING domain which facilitates substrate ubiquitination. These data led to the proposal that STING associates with the Hrd1 complex under basal conditions, and that the association is lost upon STING activation. However, the data presented is insufficient to support this model (proximity biotinylation data precludes any conclusion about direct protein-protein interactions as stated for example in line 305). What distinguishes this pool of STING from the one that is ubiquitinated and degraded in a HRD1/SEL1L dependent manner as suggested by experiments in Figs 3E,F and 5H-J?

3- It is well documented that mutations in HRD1/SEL1L lead to UPR activation in a variety of cell types and settings. Therefore, it is surprising that SEL1Llyz2 macrophages do not show UPR activation. Do these cells have low secretory capacity? The authors mention that SEL1Llyz2 mutations "resets the ER homeostasis". What does that mean? How does it relate with the pronounced changes in ER morphology (Figure 1D)? These appear very dramatic although quantification would be appropriate. In the absence of ER stress, what may explain these changes?

Other points:

- Imaging in macrophages is likely challenging but the quality of the imaging data in Figure 6 is poor. Any chance that this can be done in a cell type with better cytology?
- There is a high number of misspelled words and undefined terms (for ex S.C. and I.P.). It is worth reviewing the text carefully.
- Lines 168-169: "excluding the possibility that endogenous MHC protein complexes are bona fide SEL1L-HRD1 ERAD substrates.". This is not completely accurate since the degradation of endogenous MHC I by HRD only becomes only apparent upon B2m depletion.

Reviewer #2:

Remarks to the Author:

ER Associated Degradation (ERAD) is a mechanism by which misfolded proteins in the endoplasmic reticulum (ER) undergo degradation through the ubiquitin-proteasome system. In this manuscript, the authors report that the core ERAD factors, SEL1L and Hrd1, promote the degradation of STING in the basal state, thereby affecting the innate immune response. The authors specifically show that STING levels increase in cells devoid of SEL1L or Hrd1. They further show that STING binds to these ERAD factors and becomes ubiquitylated. When STING becomes bound to the ligand, STING dissociates from the ERAD complex to traffic to other subcellular compartments. The authors further show the

pathological significance of ERAD in the context of herpes simplex virus (HSV-1) infection and tumor growth.

Regarding the significance, the study is potentially very interesting as it reports a previously unrecognized role of ERAD in the innate immune response that affects viral infection and tumor progression. At the same time, some key conclusions could benefit from additional supporting evidence. Among those is the question as to whether STING is really a specific target of ERAD. Alternatively, one could argue that most ER transmembrane proteins become subject to ERAD while they undergo folding and maturation. According to this alternative view, STING degradation by ERAD may reflect that a fraction of STING may be slow in reaching the native conformation during the folding process. Also, the authors' experiments regarding the pathological significance (Fig. 7) specifically examine the role of SEL1L. There are no experiments here to support the specific role of the SEL1L-STING axis. Addressing these questions could significantly strengthen the manuscript. Below are some specific suggestions for the authors' consideration:

1. What if Sel1L-Hrd1 affects most ER transmembrane proteins, and STING is just one of them? In this manuscript, the authors have examined many proteins in the SEL1L knockouts. I couldn't help but to notice that those stabilized in the SEL1L deficient cells are all ER membrane proteins (e.g., Hrd1, IRE1a, and STING). Those not stabilized in SEL1L knockouts are either cytoplasmic (Hsp90, Tubulin, IRF3, Tbk1) or ER luminal proteins (BiP). In addition, the degree of STING increase in SEL1L deficient cells is modest (approximately 2 fold). Based on this, one could argue for an alternative model that most (or all) ER transmembrane proteins are subject to ERAD while they undergo folding and maturation. If the authors want to establish a more specific relationship between SEL1L and STING, they should show examples of a few control ER transmembrane proteins that are NOT stabilized in Sel1L knockouts.
2. In Figure 5d, the authors use SEL1L-TurboID to show SEL1L's interaction with STING. If the authors subject this sample for mass spectrometry analysis, they could find out whether STING is one of the very few specific SEL1L interactors, or alternatively, whether SEL1L broadly interacts with most ER transmembrane proteins.
3. Figure 7d-h: Since Sel1L-Hrd1 regulates many other proteins (as cited by the authors), it is unclear whether the observed effects of Sel1L loss in Figure 7d are specifically due to STING. It appears possible that the effect could be due to other SEL1L substrates. The authors need to present additional evidence to support the pathological significance of the SEL1L-STING axis. Perhaps one way would be to suppress the effect of Sel1L loss (Fig. 6d) by reducing or knocking down STING.
4. Also, the authors could repeat the Sel1L knockout Macrophage secretome experiments (Fig. 7f-h) without DMXAA treatment.

Minor comments:

1. Figure 7 shows pathological implications of SEL1L loss. To corroborate, can the authors perform the HSV infection experiments with the Hrd1 knockout cell line?
2. There are many grammatical and spelling errors throughout the manuscript.

Reviewer #3:

Remarks to the Author:

ER-associated degradation is responsible for the recruitment and retrotranslocation of misfolded ER proteins for proteasomal degradation in the cytosol. In this manuscript, Ji et al demonstrate a role for the SEL1L-HRD ERAD pathway in regulating basal levels of STING and thereby limiting the availability of STING for ligand binding. Overall, the data suggest that at basal state, SEL1L-HRD1 ERAD negatively regulates the protein level of STING in the ER.

The authors performed their study using mice genetically engineered to knock out SEL1L under the

control of Cre-Recombinase expression and examined the effects of SEL1L deficiency in a subset of myeloid cells including macrophages using Lyz2-Cre targeting. At basal level, SEL1L deficiency does not affect immune cells, and Lyz2-Cre targeted SEL1L deficiency did not lead to changes in LPS induced or obesity induced inflammation.

However, the authors did identify that SEL1L deficient macrophages show augmented activation by STING agonists, suggesting regulation of early STING signaling events. Their subsequent investigation reveals that SEL1L degrades basal through ERAD mediated proteosomal degradation and not through either autophagy or UPR mediated degradation. Using confocal microscopy, the authors show that at basal state, more STING protein was found in the ER but not the lysosomes of Sel1L^{Lyz2} macrophages but activation of STING using cGAMP resulted in increased p-STING foci formation within the extra-ER compartments, suggesting increased activable STING pool when SEL1L-HRD1 ERAD is not present.

Mechanistically, the authors demonstrate using co-immunoprecipitation and proximity ligation assays that SEL1L and STING directly interact with one another through their cytosolic domains. This interaction leads to ubiquitination of STING by HRD1 at multiple sites and STING's subsequent degradation by the proteasome. In the absence of SEL1L, hypersensitivity to STING agonists results from an increased size of activatable STING in the ER, leading to enhanced responsiveness of macrophages to HSV infection. The authors also show that secreted factors from SEL1L deficiency macrophages treated with STING agonist DMXAA provided protection against tumor growth. Overall, the findings within this manuscript are intriguing and novel. Basal regulation of STING protein at the ER is not well understood, and this work provides important findings that extend our understanding of how regulation of basal levels STING at the ER by ERAD limits the activation potential of STING in cells. This work has clear implications for therapeutic design to control STING activation in the setting of cancer, inflammation, and infection.

However, while the strengths of this work are clear and elegant ex vivo analysis of macrophages, the extension of these authors findings to a relevant in vivo setting was not as convincing. I would like to see the authors address/discuss the following concerns:

Major Points

- Line 249-250 – The authors dismiss autophagy as a regulator of STING protein stability under basal state. However, STING is known to induce autophagy and its degradation is regulated by autophagy. Since Atg7 is associated with canonical autophagy, it remains possible that a non-canonical autophagy pathway may contribute to regulation of basal STING protein levels. To address this, the authors could consider use of the autophagy inhibitor bafilomycin to provide pharmacologic evidence that basal STING protein levels are not broadly regulated by autophagy.
- In line 258, the authors discuss not finding upregulation of UPR markers in their investigation of macrophages. The original paper they cite (citation 53) as observing upregulation of ER stress/UPR following STING activation was focused on T lymphocytes, not macrophages as used in this study. While the authors are fair in their conclusions that STING does not induce ER stress in macrophages, can the authors comment on whether they find SEL1L-HRD1 mediated regulation of STING levels also holds true in lymphocytes? Or can they provide an alternative citation that examines ER stress induction following STING activation in macrophages?
- In Figure 7, the authors provide clear evidence that SEL1L deficiency enhances macrophage response to HSV1 infection. However, the authors do not demonstrate or discuss if this observation is relevant to HSV1 clearance either in vitro or in vivo. To examine if SEL1L deficient macrophages have enhanced viral clearance capacity ex vivo, authors could consider measuring HSV titers following infection. Moreover, it is well known that macrophages play a role in restricting HSV1 infection. It would strengthen their evidence if the authors could demonstrate that SEL1L-Lyz2 conditional knock-out mice show differences in survival and viral clearance in response to HSV1 infection.
- Additionally, the authors argue in Figure 7 that SEL1L-Lyz2 macrophages provide protection against tumor growth but demonstrated this effect through delivery of macrophage secreted factors generated ex vivo by STING agonism. While the findings of this experiment are clear, this result is somewhat expected as SEL1L KO macrophages should produce elevated inflammatory cytokines which explain

the benefit of delivery of macrophages secreted factors against tumor growth. To enhance the physiologic relevance and impact of the authors findings, we suggest that the authors consider targeting myeloid populations well described as having a STING dependent role in tumor immunity in vivo.

- Minor Points

- In their discussion (line 76), authors should also discuss the well described autophagic mechanisms connected to STING degradation in addition to proteosomal mechanisms they currently discuss.
- In Figure 2a, the authors also find that TNF α is significantly increased following TLR2 agonism by Pam3, but fail to follow-up or discuss this observation. What is the significance of this finding? Is this the subject of a subsequent investigation?
- Extended Figure 2F – The authors provide evidence that CD8 T cells are increased, but did the authors look at CD4+ T cells to see if there was a difference between Sel1L f/f and Sel1L $Ly2$ mice?
- Figure 5a – Authors show that ERAD factors such as SEL1L interacts with STING. Should consider moving this to the beginning of the manuscript to highlight the significance of studying SEL1L.
- Figure 6b – in reference to the graph of fraction # and percent, comparing Sel1L f/f and Sel1L $Ly2$, can the authors explain how STING mass was quantitated? This is not clearly explained in the methods.

ABSTRACT AND MAIN TEXT – please follow the guidelines that are specific to the format of your manuscript, as listed in our Guide to Authors (http://www.nature.com/ncb/pdf/ncb_gta.pdf) Briefly,

Nature Cell Biology Articles, Resources and Technical Reports have 3500 words, including a 150 word abstract, and the main text is subdivided in Introduction, Results, and Discussion sections. Nature Cell Biology Letters have up to 2500 words, including a 180 word introductory paragraph (abstract), and the text is not subdivided in sections.

Methods should be written concisely, but should contain all elements necessary to allow interpretation and replication of the results. As a guideline, Methods sections typically do not exceed 3,000 words. The Methods should be divided into subsections listing reagents and techniques. When citing previous methods, accurate references should be provided and any alterations should be noted. Information must be provided about: antibody dilutions, company names, catalogue numbers and clone numbers for monoclonal antibodies; sequences of RNAi and cDNA probes/primers or company names and catalogue numbers if reagents are commercial; cell line names, sources and information on cell line identity and authentication. Animal studies and experiments involving human subjects must be reported in detail, identifying the committees approving the protocols. For studies involving human subjects/samples, a statement must be included confirming that informed consent was obtained. Statistical analyses and information on the reproducibility of experimental results should be provided in a section titled "Statistics and Reproducibility".

All Nature Cell Biology manuscripts submitted on or after March 21 2016 must include a Data availability statement as a separate section after Methods but before references, under the heading "Data Availability". For Springer Nature policies on data availability see <http://www.nature.com/authors/policies/availability.html>; for more information on this particular policy see <http://www.nature.com/authors/policies/data/data-availability-statements-data->

citations.pdf. The Data availability statement should include:

- Accession codes for primary datasets (generated during the study under consideration and designated as "primary accessions") and secondary datasets (published datasets reanalysed during the study under consideration, designated as "referenced accessions"). For primary accessions data should be made public to coincide with publication of the manuscript. A list of data types for which submission to community-endorsed public repositories is mandated (including sequence, structure, microarray, deep sequencing data) can be found here <http://www.nature.com/authors/policies/availability.html#data>.
- Unique identifiers (accession codes, DOIs or other unique persistent identifier) and hyperlinks for datasets deposited in an approved repository, but for which data deposition is not mandated (see here for details <http://www.nature.com/sdata/data-policies/repositories>).
- At a minimum, please include a statement confirming that all relevant data are available from the authors, and/or are included with the manuscript (e.g. as source data or supplementary information), listing which data are included (e.g. by figure panels and data types) and mentioning any restrictions on availability.
- If a dataset has a Digital Object Identifier (DOI) as its unique identifier, we strongly encourage including this in the Reference list and citing the dataset in the Methods.

We recommend that you upload the step-by-step protocols used in this manuscript to the Protocol Exchange. More details can be found at www.nature.com/protocolexchange/about.

All imaging data should be accompanied by scale bars, which should be defined in the legend. Cropped images of gels/blots are acceptable, but need to be accompanied by size markers, and to retain visible background signal within the linear range (i.e. should not be saturated). The boundaries of panels with low background have to be demarked with black lines. Splicing of panels should only be considered if unavoidable, and must be clearly marked on the figure, and noted in the legend with a statement on whether the samples were obtained and processed simultaneously. Quantitative comparisons between samples on different gels/blots are discouraged; if this is unavoidable, it should only be performed for samples derived from the same experiment with gels/blots were processed in parallel, which needs to be stated in the legend.

Figures should be provided at approximately the size that they are to be printed at (single column is 86 mm, double column is 170 mm) and should not exceed an A4 page (8.5 x 11"). Reduction to the scale that will be used on the page is not necessary, but multi-panel figures should be sized so that the whole figure can be reduced by the same amount at the smallest size at which essential details in each panel are visible. In the interest of our colour-blind readers we ask that you avoid using red and green for contrast in figures. Replacing red with magenta and green with turquoise are two possible colour-safe alternatives. Lines with widths of less than 1 point should be avoided. Sans serif typefaces,

such as Helvetica (preferred) or Arial should be used. All text that forms part of a figure should be rewritable and removable.

The total number of Supplementary Figures (not including the “unprocessed scans” Supplementary Figure) should not exceed the number of main display items (figures and/or tables (see our Guide to Authors and March 2012 editorial <http://www.nature.com/ncb/authors/submit/index.html#suppinfo>; <http://www.nature.com/ncb/journal/v14/n3/index.html#ed>). No restrictions apply to Supplementary Tables or Videos, but we advise authors to be selective in including supplemental data.

GUIDELINES FOR EXPERIMENTAL AND STATISTICAL REPORTING

REPORTING REQUIREMENTS – We are trying to improve the quality of methods and statistics reporting in our papers. To that end, we are now asking authors to complete a reporting summary that collects information on experimental design and reagents. The Reporting Summary can be found here <https://www.nature.com/documents/nr-reporting-summary.pdf>. If you would like to reference the guidance text as you complete the template, please access these flattened versions at <http://www.nature.com/authors/policies/availability.html>.

Author Rebuttal to Initial comments

We thank all three reviewers for their insightful and constructive comments – very helpful indeed! In the past 3 months, we now have carefully addressed all the comments from the reviewers, which have been instrumental in our effort to further improve and strengthen our manuscript.

Reviewers' Comments:

Reviewer #1:

Remarks to the Author:

Ji and colleagues investigate a potential role of the HRD1 ERAD complex in innate immunity by generating macrophage-specific KO mice for SEL1L (SEL1LLyz2), a key subunit of the Hrd1 complex. It is shown that SEL1LLyz2 mice appeared normal and displayed normal repertoires of macrophages as well as other immune cells. SEL1LLyz2 macrophages appeared to display normal cell surface levels of MHC I and II and to respond normally to Toll-like receptor and RIG-I stimulation. However, STING stimulation with cGAMP resulted in increased production of various cytokines in SEL1LLyz2 macrophages. Consistent with the enhanced response, STING steady state protein levels were higher both in SEL1LLyz2 macrophages and in a macrophage cell line deficient for HRD1. Biochemical and proximity proteomics approaches, indicated that HRD1 complex interacts and ubiquitylates STING. The interaction appears to be mediated by the cytosolic regions of both STING and HRD1. Finally, the physiological relevance of increase STING basal levels was tested in two distinct paradigms, following a herpes simplex virus infection and in anti-tumor response in to KRAS-driven tumor implants.

Based on these findings, it is suggested that the basal pool of ER STING available for activation is regulated by the HRD1 ERAD complex. A model in which “ligand binding triggers the dissociation between SEL1L-HRD1 ERAD and STING leading to the ER-exit and activation of STING protein” is put forward. This model suggests that loss of HRD1 increases STING basal levels and results in enhanced STING signaling upon stimulation.

The manuscript focuses on an interesting topic, it is clearly written and most of the data appears of good quality, even if it is often overinterpreted. It is also convincingly shown that the levels of STING are post-translationally regulated by the HRD/SEL1L complex. On the negative side, the evidence for the model proposed is weak (see below). In addition, the physiological importance of STING regulation by the HRD1 complex is unclear. The differences observed in the HSV-1 model are subtle while the phenotypes in the tumor restriction model are detected only if a strong STING activator (DMXAA) is applied exogenously.

We thank this reviewer for his/her insightful comments. We now have addressed all the issues raised here, which have significantly improved our work and solidified our model that STING is misfolding prone and is degraded by SEL1L-HRD1 ERAD under basal condition, which limits the amplitude of STING activation following ligand binding. In the *in vivo* tumor model, the use of secretome from agonist-treated primary macrophages is a better approach to investigate the role of myeloid-specific ERAD in STING innate immunity than the *in vivo* injection of agonist into WT and KO mice, as many other cell types such as T cells may compensate the loss of ERAD in myeloid cells. Many additional experiments, *in vivo* and *in vitro*, have been performed as detailed below in Response Figures 7, 8, 9, and 12 to further strengthen the data for the tumor model.

Major points

1- The increase in STING levels in SEL1L and HRD1 deficient cells is clear and convincing. However, it is less clear how this occurs and some of the observations are confusing and paradoxical. Based on cycloheximide chase experiments, the turnover of endogenous STING is slow and hardly affected by HRD1 depletion (Fig 3F). Perhaps pulse-labeling experiments would be useful to capture small differences that are difficult to see by cycloheximide chase (IRE1a would serve as a good control in these experiments). On the other hand, STING is robustly ubiquitinated in a Hrd1-dependent fashion (Fig 5H-J). Surprisingly, under these conditions it is stated that ubiquitinated STING is degraded so quickly that the modified species can only be captured upon proteasome inhibition (lines 314-316). However, it is puzzling that such quick degradation does not appear to impact on total STING levels (Fig 5H, anti-STING blot). It would be

important to reconcile these observations.

We thank the reviewer for this great comment. We now have performed the CHX-chase experiments to examine STING protein turnover in various types of macrophages and MEFs, including primary macrophages and MEFs in Fig. 3e-f and Raw 264.7 macrophage cell line in Response Figure 1a. All of these data showed that in WT cells, STING protein half-life was about 8-9 hr. Multiple attempts to perform pulse-chase experiment have failed due to the lack of specific STING Ab for pulse chase: as shown in Response Figure 1b, STING Ab and non-specific isotype IgG both pulled down a band around 38 kDa (arrow). In light of this, we feel that translation shut-off assay using cycloheximide is the best alternative.

As to the question of ubiquitination and protein levels, our data shown in Fig. 5h suggested that a small fraction of total STING protein is ubiquitinated at any given time (in this experiment, for 5 hr window), which are degraded by the proteasomes and can only be visualized with the addition of MG132. This would explain why “quick (proteasomal) degradation does not appear to impact on total STING levels” as stated by the reviewer. We now have quantitated the input of STING protein levels in macrophages with or without MG132 treatment (original data in Fig. 5h), which showed a 39% increase in WT, but not *HRD1*^{-/-}, macrophages upon MG132 treatment for 5 hr (Response Figure 1C). We also clarified this point in the Results (Line 324-29): “Moreover, *HRD1* robustly ubiquitinated STING protein, which was quickly degraded

by the proteasomes as MG132 treatment was required to visualize the poly-ubiquitin chain of STING (Fig. 5h). The observation that MG132 treatment for 5 hr led to 39% increase of non-ubiquitinated STING protein in *Hrd1*^{+/+} macrophages (Fig. 5h and Extended Data Fig. 6d) suggested that ERAD degrades a subset of nascent STING protein in the ER”. Response Figure 1a and 1c are now shown in Extended Data Figure 4 and 6d in the revised manuscript.

± s.e.m. n.s., not significant; ***P* < 0.01, by unpaired, two-tailed, Student's *t*-test.

2- Immunoprecipitation experiments suggest an interaction between SEL1L and STING (Fig 5B-C). These data would be strengthened by the inclusion of appropriate specificity controls, ideally an abundant ER membrane protein rather than TBK1, a soluble protein localized to a different cellular compartment, or cGAMP, which triggers STING to traffic out of the ER. Interactions between STING and HRD1/SEL1L were also detected using proximity biotinylation. These were also lost upon STING stimulation by cGAMP. Like the immunoprecipitations, these experiments also lack specificity controls remaining unclear if all/most ER membrane proteins are labelled under these conditions. It seems that the Turbo-ID experiments are reporting on STING ER localization and not so much on its association with the HRD1 complex. Figure 5G shows an interesting experiment suggesting that HRD1/STING interaction requires their cytosolic domains, in particular HRD1 RING domain which facilitates substrate ubiquitination. These data led to the proposal that STING associates with the Hrd1 complex under basal conditions, and that the association is lost upon STING activation. However, the data presented is insufficient to support this model (proximity biotinylation data precludes any conclusion about direct protein-protein interactions as stated for example in line 305).

What distinguishes this pool of STING from the one that is ubiquitinated and degraded in a HRD1/SEL1L dependent manner as suggested by experiments in Figs 3E,F and 5H-J?

We thank the reviewer for the great comments. As requested, we now included additional ER membrane proteins as controls for immunoprecipitation in macrophages with or without cGAMP stimulation experiments shown in Response Figure 2a-c. These new data showed interactions under basal conditions between SEL1L and STING, as well as components of the SEL1L-HRD1 ERAD machinery including HRD1, OS9 and Calnexin, but not with other ER membrane proteins such as calcium sensor STIM1 and STING downstream effector TBK1 (Response Figure 2a-b). Moreover, DMXAA treatment decreased the interaction between STING and SEL1L-HRD1 ERAD complex, but had an opposite effect on the STING-TBK1 interaction. As a negative control, ER-resident lipid metabolism gene FA4L4 had no interaction with STING under either condition (Response Figure 2b). In addition, SEL1L-TurboID biotinylated STING, HRD1 and Calnexin, but not ER-resident Sterol O-acyltransferase (SOAT1) under basal state; and cGAMP stimulation decreased SEL1L interaction with STING, but the interaction of SEL1L with HRD1 and Calnexin persisted (Response Figure 2c). Hence, we concluded that SEL1L-HRD1 ERAD interacts selectively with a subset of ER membrane proteins and that its effect on STING is specific under the basal state. Fig. 5b-d is now replaced with Response Figure 2a-c in the revised manuscript.

We thank the reviewer for the great point about the Turbo-ID experiment, which we missed. Indeed, this experiment reflected the direct interaction between SEL1L-HRD1 ERAD and STING “in the ER”, and failed to support our original claim that “ligand binding triggers the dissociation between SEL1L-HRD1 ERAD and STING” - an overstatement. Hence, we have revised the manuscript and the model in Extended Data Fig. 10 (Response Figure 2d) to reflect that ligand binding and ERAD-STING interaction may represent two independent events. Whether and how they may be coupled are interesting open questions. The original statement “Ligand binding triggers the dissociation between SEL1L-HRD1 ERAD and STING, leading to the ER-exit and activation of STING protein (Extended Data Fig. 10a).” is now replaced with a new statement throughout the manuscript, e.g. “Indeed, newly synthesized STING protein is misfolding prone and directly ubiquitinated and degraded by SEL1L-HRD1 ERAD in the ER. In the absence of SEL1L-HRD1 ERAD, STING accumulates in the ER, hence forming a larger activable STING pool under the basal state (Extended Data Fig. 10a).” at Line 432-5.

Response Figure 2. The interaction between SEL1L-HRD1 ERAD and STING at the ER. (a-b) Immunoblot analysis following immunoprecipitation (IP) of endogenous SEL1L (a) and STING (b) from lysates of primary macrophage cells treated with or without DMXAA for 3 hr. **(c)** Immunoblot analysis following IP with streptavidin beads of lysates from cells transfected with SEL1L-TurboID plasmids and treated with cGAMP for the indicated time points. **(d)** A revised model: a fraction of STING in the ER is subject to SEL1L-HRD1 ERAD. In the absence of SEL1L-HRD1 ERAD, more STING proteins accumulate at the ER, and upon ligand binding, exit the ER and become activated in the extra-ER compartments.

3- It is well documented that mutations in HRD1/SEL1L lead to UPR activation in a variety of cell types and settings. Therefore, it is surprising that SEL1Llyz2 macrophages do not show UPR activation. Do these cells have low secretory capacity? The authors mention that SEL1Llyz2 mutations “resets the ER homeostasis”. What does that mean? How does it relate with the pronounced changes in ER morphology (Figure 1D)? These appear very dramatic although quantification would be appropriate. In the absence of ER stress, what may explain these changes?

We thank the reviewer for the insightful comments. Macrophages are considered as a secretory cell type with high secretory capacity upon stimulation¹. *Sel1L*^{-/-} and WT macrophages exhibit comparable secretory capacity of various inflammatory cytokines upon ligand stimulation (Fig. 1g and 2e). We and others have recently shown that, in most cell types examined to date, SEL1L-HRD1 ERAD deficiency is coupled with very mild ER stress, and without overt cell death before or at disease initiation *in vivo*²⁻⁴. These findings point to the existence of one or more compensatory mechanism(s) in cells with SEL1L-HRD1 ERAD dysfunction, including but not limited to, elevation of ER folding chaperones and ER volume (Fig. 1), sequestration of soluble toxic proteins and/or the activation of selective ER-phagy⁵. Quantitation of the ER perimeter and area is shown in Response Figure 3a-b (now Fig. 1d in the revised manuscript). These compensatory mechanisms, in our opinion, “reset ER homeostasis” in response to ERAD deficiency, which are likely resulted from a combined action of protein accumulation and aggregation and low-level UPR. In providing further support for the altered ER homeostasis, we now showed that treatment of ER stressor thapsigargin acutely induced slightly stronger ER stress in *Sel1L*-deficient macrophages as measured by *Xbp1* mRNA splicing (Response Figure 3c). We have clarified this point in the Results (Line 146-49): “Taken together, these data indicate that SEL1L deficiency is well tolerated by macrophages *in vivo* under basal condition, as demonstrated by a subtle UPR and the lack of any detectable changes in immune cell composition and survival.”

Response Figure 3. Altered ER homeostasis in *Sel1L*^{-/-} macrophage.

(a-b) Quantitation of ER perimeter length (a) and area (b) in TEM images. (c) RT-PCR analysis showing *Xbp1* splicing in macrophages treated without or with 100 nM Tg for 1 and 1.5 hr. Values, mean ± s.e.m. n.s., not significant; ****P* < 0.001, by unpaired, two-tailed, Student's *t*-test.

Other points:

- Imaging in macrophages is likely challenging but the quality of the imaging data in Figure 6 is poor. Any chance that this can be done in a cell type with better cytology?

We thank the reviewer for this great comment. We now included new data with much improved quality, showing the localization of (p)-STING in the KDEL-positive ER of macrophages (Response Figure 4a-b). Some of these images are now shown in Figure 6c in the revised manuscript.

Response Figure 4. ERAD^{-/-} macrophages have an enlarged pool of ER-resident STING under basal state. (a-b) Representative confocal microscopic images of (a) STING and (b) p-STING staining, co-stained with KDEL and DAPI in primary peritoneal macrophages. In b, cells were treated with or without cGAMP for 6 hr prior to the immunostaining.

- There is a high number of misspelled words and undefined terms (for ex S.C. and I.P.). It is worth reviewing the text carefully.

We thank the reviewer for the comment. We apologize for the errors and typos and now have gone through the text carefully and fixed them, including s.c. and i.p. (subcutaneous and intraperitoneal, respectively).

- Lines 168-169: "excluding the possibility that endogenous MHC protein complexes are bona fide SEL1L-HRD1 ERAD substrates.". This is not completely accurate since the degradation of endogenous MHC I by HRD only becomes only apparent upon B2m depletion.

We now have rephrased this sentence based on this suggestion (Line 169-71)": "..., suggesting that, unlike orphan or mutant MHC class I heavy chains, endogenous MHC protein complexes are not ERAD substrates."

Reviewer #2:

Remarks to the Author:

ER Associated Degradation (ERAD) is a mechanism by which misfolded proteins in the endoplasmic reticulum (ER) undergo degradation through the ubiquitin-proteasome system. In this manuscript, the authors report that the core ERAD factors, SEL1L and Hrd1, promote the degradation of STING in the basal state, thereby affecting the innate immune response. The authors specifically show that STING levels increase in cells devoid of SEL1L or Hrd1. They further show that STING binds to these ERAD factors and becomes ubiquitylated. When STING becomes bound to the ligand, STING dissociates from the ERAD complex to traffic to other subcellular compartments. The authors further show the pathological significance of ERAD in the context of herpes simplex virus (HSV-1) infection and tumor growth.

Regarding the significance, the study is potentially very interesting as it reports a previously unrecognized role of ERAD in the innate immune response that affects viral infection and tumor progression. At the same time, some key conclusions could benefit from additional supporting evidence. Among those is the question as to whether STING is really a specific target of ERAD. Alternatively, one could argue that most ER transmembrane proteins become subject to ERAD while they undergo folding and maturation. According to this alternative view, STING degradation by ERAD may reflect that a fraction of STING may be slow in reaching the native conformation during the folding process. Also, the authors' experiments regarding the pathological significance (Fig. 7) specifically examine the role of SEL1L. There are no experiments here to support the specific role of the SEL1L-STING axis. Addressing these questions could significantly strengthen the manuscript.

We thank this reviewer for his/her insightful comments. We now have addressed all the issues raised below, which have significantly improved our work and solidified our model. Despite decades of research, it remains unclear, for the field in general, what makes a protein a substrate of SEL1L-HRD1 ERAD. Our new data showed that SEL1L-HRD1 ERAD selectively interacts with a small fraction of ER membrane proteins and that SEL1L-HRD1 ERAD specially downregulates STING protein abundance in the ER, thereby limiting its activable pool of STING.

Below are some specific suggestions for the authors' consideration:

1. What if Sel1L-Hrd1 affects most ER transmembrane proteins, and STING is just one of them? In this manuscript, the authors have examined many proteins in the SEL1L knockouts. I couldn't help but to notice that those stabilized in the SEL1L deficient cells are all ER membrane proteins (e.g., Hrd1, IRE1a, and STING). Those not stabilized in SEL1L knockouts are either cytoplasmic (Hsp90, Tubulin, IRF3, Tbk1) or ER luminal proteins (BiP). In addition, the degree of STING increase in SEL1L deficient cells is modest (approximately 2 fold). Based on this, one could argue for an alternative model that most (or all) ER transmembrane proteins are subject to ERAD while they undergo folding and maturation. If the authors want to establish a more specific relationship between SEL1L and STING, they should show examples of a few control ER transmembrane proteins that are NOT stabilized in Sel1L knockouts.

We thank the reviewer for this great comment. We now have examined the levels of several additional ER membrane proteins, including FACL4, SOAT1, CALNEXIN, STIM1, TLR2, TLR4, PD-L1, H-2K^b/H-2D^b (MHC I) and I-A/I-E (MHC II) proteins in *Sel1L*^{-/-} macrophages, none of which were accumulated in the absence of SEL1L-HRD1 ERAD (Response Figure 5a-d). Hence, our new data showed that STING is selectively targeted by SEL1L-HRD1 ERAD. These data are now shown in Figure 3a and Extended Data Figure 3b-c in the revised manuscript.

Response Fig 5. Not all ER and membrane proteins are accumulated in *Sel1L*^{-/-} macrophages. (a) Immunoblot analysis in *Sel1L*^{fl/fl} and *Sel1L*^{Ly2/2} macrophages, with quantitation (normalized to β -tubulin) shown below the gel. (b-d) Flow cytometric analysis of surface (b) and total (c) levels of TLR2, TLR4, PD-L1, H-2K^b/H-2D^b (MHC I) and I-A/I-E (MHC II) in macrophages, with quantitation of mean fluorescence intensity shown in (d). N=4 and 3 mice each. Values represent mean \pm SEM. n.s., no significant difference, using two-tailed Student's *t* test (d).

2. In Figure 5d, the authors use SEL1L-TurboID to show SEL1L's interaction with STING. If the authors subject this sample for mass spectrometry analysis, they could find out whether STING is one of the very few specific SEL1L interactors, or alternatively, whether SEL1L broadly interacts with most ER transmembrane proteins.

We thank the reviewer for this great comment. Indeed, the proposed experiment is one of the major efforts in the Qi laboratory to take an unbiased approach to identify ERAD interactomes in cell type-specific manner. Taking advantage of the notion that SEL1L interaction with endogenous substrates are prolonged in the absence of HRD1, we have generated SEL1L specific antibody suited for immunoprecipitation and performed LC/MS on the immunoprecipitates in different cell types, including macrophages and MEF cells. As shown in Response Figure 6a-b, SEL1L interacting membrane proteins are only minor fractions of total ER membrane proteins (60 and 27 vs. 943 and 976) in MEFs and macrophages, respectively. Thus, together with those data shown in Response Figure 5, we believe that SEL1L selectively interacts with a small fraction of ER membrane proteins including STING.

Response Figure 6. SEL1L interacts with a small fraction of ER transmembrane proteins in MEFs and macrophages. Graph showing the number of ER membrane proteins in the candidate hits of SEL1L-IP MS from WT, *Sel1L*^{-/-} and *HRD1*^{-/-} MEF cells (a) and macrophages (b).

3. Figure 7d-h: Since *Sel1L-Hrd1* regulates many other proteins (as cited by the authors), it is unclear whether the observed effects of *Sel1L* loss in Figure 7d are specifically due to STING. It appears possible that the effect could be due to other SEL1L substrates. The authors need to present additional evidence to support the pathological significance of the SEL1L-STING axis. Perhaps one way would be to suppress the effect of *Sel1L* loss (Fig. 6d) by reducing or knocking down STING.

We thank the reviewer for this great comment. We now have repeated the HSV infection experiment by including the STING-specific inhibitor H151, which showed that STING indeed links ERAD to IFN- β response against HSV (Response Figure 7). These data are now shown in Extended Data Figure 8b-c in the revised manuscript.

Response Figure 7. The new “SEL1L-HRD1 ERAD-STING” axis regulates macrophage antiviral immunity. (a-b) q-PCR (a) and ELISA (b) analysis of gene expression and secretion of IFN- β , respectively, in macrophages infected with HSV-1 (MOI = 5) for 6 hr, with or without STING inhibitor H151 pretreatment. $n=4$ mice each. Values represent mean \pm s.e.m. ** $P < 0.01$; *** $P < 0.001$, by unpaired, two-tailed, Student’s t -test.

4. Also, the authors could repeat the *Sel1L* knockout Macrophage secretome experiments (Fig. 7f-h) without DMXAA treatment.

We thank the reviewer for this great comment. We now have performed the experiment as suggested, which showed that secretome from naïve *Sel1L*^{Ly2/2} macrophage (without DMXAA treatment) is insufficient to drive anti-tumor immunity (Response Figure 8). These data are now shown in Extended Data Figure 9 in the revised manuscript.

Response Figure 8. The new “SEL1L-HRD1 ERAD-STING” axis in macrophages regulates anti-tumor immunity. (a) Representative images of pancreatic tumors of 5 groups of WT mice received medium or secretome from DMXAA- or mock- treated macrophages from *Sel1L*^{fl/fl} and *Sel1L*^{Ly2/2} mice. Experiments were performed as described in Figure 7f. Quantitation of tumor weights at the end of experiment shown in (b). $n=5, 5, 5, 12, 12$ mice (left to right). Values represent mean \pm s.e.m. * $P < 0.05$; n.s., no significant difference, by unpaired, two-tailed, Student’s t -test.

Minor comments:

1. Figure 7 shows pathological implications of SEL1L loss. To corroborate, can the authors perform the HSV infection experiments with the *Hrd1* knockout cell line?

We thank the reviewer for this great comment. We now have performed HSV infection experiment using *HRD1*-deficient macrophage, which established the role of SEL1L-*HRD1* ERAD in innate immunity against HSV (Response Figure 9). These data are now shown in Extended Data Figure 8d-e in the revised manuscript.

Response Figure 9. HRD1 in macrophages regulates antiviral immunity. (a) q-PCR analysis of *Ifnb* gene in *Hrd1*^{+/+} and *Hrd1*^{-/-} RAW 264.7 cells infected HSV-1 (MOI = 5) for 6 hr. $n=6$ each. Values represent mean \pm s.e.m. *** $P < 0.001$, by unpaired, two-tailed, Student’s t -test. (b) Representative immunoblot showing TBK1 activation in *Hrd1*^{+/+} and *Hrd1*^{-/-} RAW 264.7 cells treated with HSV-1 (MOI = 5) for 6 hr, with the quantitation of the ratio of phosphorylated to total proteins (p/t) shown below the gel.

2. There are many grammatical and spelling errors throughout the manuscript.

We apologize for the errors. We now have gone through the text very carefully and fixed all.

Reviewer #3:

Remarks to the Author:

ER-associated degradation is responsible for the recruitment and retrotranslocation of misfolded ER proteins for proteasomal degradation in the cytosol. In this manuscript, Ji et al demonstrate a role for the SEL1L-HRD ERAD pathway in regulating basal levels of STING and thereby limiting the availability of STING for ligand binding. Overall, the data suggest that at basal state, SEL1L-HRD1 ERAD negatively regulates the protein level of STING in the ER. The authors performed their study using mice genetically engineered to knock out SEL1L under the control of Cre-Recombinase expression and examined the effects of SEL1L deficiency in a subset of myeloid cells including macrophages using Lyz2-Cre targeting. At basal level, SEL1L deficiency does not affect immune cells, and Lyz2-Cre targeted SEL1L deficiency did not lead to changes in LPS induced or obesity induced inflammation. However, the authors did identify that SEL1L deficient macrophages show augmented activation by STING agonists, suggesting regulation of early STING signaling events. Their subsequent investigation reveals that SEL1L degrades basal through ERAD mediated proteasomal degradation and not through either autophagy or UPR mediated degradation. Using confocal microscopy, the authors show that at basal state, more STING protein was found in the ER but not the lysosomes of Sel1LLyz2 macrophages but activation of STING using cGAMP resulted in increased p-STING foci formation within the extra-ER compartments, suggesting increased activable STING pool when SEL1L-HRD1 ERAD is not present. Mechanistically, the authors demonstrate using co-immunoprecipitation and proximity ligation assays that SEL1L and STING directly interact with one another through their cytosolic domains. This interaction leads to ubiquitination of STING by HRD1 at multiple sites and STING's subsequent degradation by the proteasome. In the absence of SEL1L, hypersensitivity to STING agonists results from an increased size of activatable STING in the ER, leading to enhanced responsiveness of macrophages to HSV infection. The authors also show that secreted factors from SEL1L deficiency macrophages treated with STING agonist DMXAA provided protection against tumor growth.

Overall, the findings within this manuscript are intriguing and novel. Basal regulation of STING protein at the ER is not well understood, and this work provides important findings that extend our understanding of how regulation of basal levels STING at the ER by ERAD limits the activation potential of STING in cells. This work has clear implications for therapeutic design to control STING activation in the setting of cancer, inflammation, and infection. However, while the strengths of this work are clear and elegant ex vivo analysis of macrophages, the extension of these authors findings to a relevant in vivo setting was not as convincing.

We thank this reviewer for his/her insightful comments. We now have addressed all the issues raised below, which have significantly improved our work and strengthened our model.

I would like to see the authors address/discuss the following concerns:

Major Points

- Line 249-250 – The authors dismiss autophagy as a regulator of STING protein stability under basal state. However, STING is known to induce autophagy and its degradation is regulated by autophagy. Since Atg7 is associated with canonical autophagy, it remains possible that a non-canonical autophagy pathway may contribute to regulation of basal STING protein levels. To address this, the authors could consider use of the autophagy inhibitor bafilomycin to provide pharmacologic evidence that basal STING protein levels are not broadly regulated by autophagy.

We thank the reviewer for this great comment. We now have performed the experiment to measure basal STING levels using a lysosome inhibitor bafilomycin A1 (BafA1), a compound that inhibits lysosomal acidification and degradation^{6,7}. In line with previous reports^{8,9}, our data showed that BafA1 increased STING protein level in DMXAA-stimulated cells, while having no effect on STING protein level under basal state (Response Figure 10) – now shown in Fig. 3j in the revised manuscript. Hence, our data, together with those shown in Figure 3g-i using primary *Atg7*^{-/-} macrophages, demonstrated that *Atg7*-mediated macroautophagy is dispensable for regulating STING protein stability under both basal and active states, while endolysosome-mediated degradation plays an important role in the degradation of active STING. We now also revised the Introduction and Discussion to clarify this point (Line 82-5): “*In addition, activated*

STING can be sorted into acidified endolysosomes for degradation. This membrane trafficking process includes adaptor protein complex 1 (AP-1)-mediated delivery from the Golgi to endolysosomes via clathrin-coated transport vesicles, lysosomal membrane Niemann–Pick type C1 (NPC1) or p62/SQSTM1-dependent autophagy⁸⁻¹²”; and (Line 429-31): “Unlike endolysosome-dependent degradation of active STING to help terminate its signaling⁸⁻¹² (Fig. 3j), SEL1L-HRD1 ERAD degrades naïve STING under the basal state to limits its activation potential.”

Response Figure 10. Autophagy regulates STING protein level following agonist activation, while having no effect under basal state. Immunoblot analysis in primary macrophages treated with or without autophagy inhibitor bafilomycin A1 (BafA1) for 6 hr, DMXAA for 3 hr, or DMXAA and BafA1 for 3 hr with quantitation shown below the gel.

- In line 258, the authors discuss not finding upregulation of UPR markers in their investigation of macrophages. The original paper they cite (citation 53) as observing upregulation of ER stress/UPR following STING activation was focused on T lymphocytes, not macrophages as used in this study. While the authors are fair in their conclusions that STING does not induce ER stress in macrophages, can the authors comment on whether they find SEL1L-HRD1 mediated regulation of STING levels also holds true in lymphocytes? Or can they provide an alternative citation that examines ER stress induction following STING activation in macrophages?

We thank the reviewer for this great comment. We now have performed the experiment as suggested using a recently generated T cell-specific *Sel1L*^{-/-} (*Sel1L*^{Lck}) mice¹³. Our pilot data showed that, while STING protein level was not elevated in *Sel1L*^{Lck} immature thymocytes (Response Figure 11), it is elevated slightly in the mature splenic T cells. This finding suggest that, while STING may not be regulated by ERAD in T lymphocytes during development, it may be a substrate of SEL1L-HRD1 ERAD in mature T cells. This may

reflect the cell type-specific role of ERAD in different immune cells, and whether SEL1L-HRD1 regulates STING in T-cell subpopulation remains to be addressed. Given that this manuscript is focused on the myeloid cells, we feel that this data is beyond the scope of the current manuscript.

Response Figure 11. STING protein level was unchanged in SEL1L-deficient thymocytes. Immunoblot analysis of thymus (a) and purified splenic T cells (b) from *Sel1L*^{fl/fl} and *Sel1L*^{Lck} mice¹³, with quantitation of STING (normalized to β-tubulin) shown below the gel. IRE1α, a known SEL1L-HRD1 ERAD substrate, was elevated in the *Sel1L*^{Lck} thymocytes.

- In Figure 7, the authors provide clear evidence that SEL1L deficiency enhances macrophage response to HSV1 infection. However, the authors do not demonstrate or discuss if this observation is relevant to HSV1 clearance either in vitro or in vivo. To examine if SEL1L deficient macrophages have enhanced viral clearance capacity ex vivo, authors could consider measuring HSV titers following infection. Moreover, it is well known that macrophages play a role in restricting HSV1 infection. It would strengthen their evidence if the authors could demonstrate that SEL1L-Lyz2 conditional knock-out mice show differences in survival and viral clearance in response to HSV1 infection.

We thank the reviewer for this great comment. We now have performed the additional experiment to examine the ERAD effect on HSV titer *in vitro* and infection *in vivo*. Unexpectedly, in spite of higher IFN-β response (Fig. 7d-e, and Response Figure 12c), ERAD deficiency in myeloid cells does not block or attenuate HSV infection progression *in vivo*. Indeed, both viral titer and survival rate were comparable between the two cohorts (Response Figure 12a-b, d). Hence, although *Sel1L*^{Lyz2} macrophages exhibit higher STING innate immunity *in vitro* and *in vivo* in response to HSV infection, *Sel1L*^{Lyz2} mice are unable to mount sufficient protection against HSV infection, pointing to additional factors and steps regulated by SEL1L-HRD1 ERAD during HSV infection. As this observation potentially opens up new directions as well

as for the sake of clarity and simplicity, we decided not to include this data in the revise manuscript. Nonetheless, we have discussed this point in the Discussion (Line 435-40): “Our data further show that *SEL1L* deficiency enhances *STING*-mediated type I IFN immune response against HSV-1 infection *ex vivo* (Fig. 7d-e). However, whether *ERAD* deficiency renders the host resistant to HSV-1 infection *in vivo* remains to be determined as the process of establishing a successful viral infection is much more complicated, which also involves viral cellular entry, viral DNA replication, and clearance¹⁴. Whether these processes may be regulated by *SEL1L*-*HRD1* *ERAD* remains open interesting questions.”

Response Figure 12. HSV-1 infection titers and survival. (a-b) q-PCR analysis of intracellular HSV-1 DNA copy (a) and plaque analysis of supernatants (b) from *Sel1L^{fl/fl}* and *Sel1L^{Lyz2}* macrophages infected with HSV-1 (MOI = 5) for 1 hr. Following washes with PBS, cells were cultured for 24 hr prior to analyses. n=6 (a) or

3 (b) each. n.s., not significant, by unpaired, two-tailed, Student’s t-test. (c) q-PCR analysis of *Ifnb* gene in the brains from *Sel1L^{fl/fl}* and *Sel1L^{Lyz2}* mice 3 days post *i.p.* injection of HSV-1, 1x10⁷ pfu per mouse. n=3 mice each. Values represent mean ± s.e.m. **P < 0.01, by unpaired, two-tailed, Student’s t-test. (d) Survival curve of *Sel1L^{fl/fl}* and *Sel1L^{Lyz2}* mice post *i.p.* injection of HSV-1, 2x10⁸ pfu per mouse. n=13 and 12 for *Sel1L^{fl/fl}* and *Sel1L^{Lyz2}* mice. p=0.3367 by Log-rank (Mantel-Cox) test.

- Additionally, the authors argue in Figure 7 that *SEL1L*-*Lyz2* macrophages provide protection against tumor growth but demonstrated this effect through delivery of macrophage secreted factors generated *ex vivo* by *STING* agonism. While the findings of this experiment are clear, this result is somewhat expected as *SEL1L* KO macrophages should produce elevated inflammatory cytokines which explain the benefit of delivery of macrophages secreted factors against tumor growth. To enhance the physiologic relevance and impact of the authors findings, we suggest that the authors consider targeting myeloid populations well described as having a *STING* dependent role in tumor immunity *in vivo*.

We thank the reviewer for this great comment. We are currently dissecting the myeloid cell population using cell type-specific mouse models. As this effort would take another couple more years to complete, we now have discussed the possible role of different myeloid cell populations in the Discussion (Line 440-6): “Lastly, our data show that administration of secretome from *STING* agonist-activated *Sel1L*-deficient macrophages, but not the agonist itself, renders the host resistant to malignant pancreatic tumors in a transplantation model (Fig. 7f-h and Extended Data Fig. 9). Although the role of *SEL1L*-*HRD1* *ERAD* in various myeloid populations in tumor immunity remains unclear in this model, this study uncovers a novel function for *SEL1L*-*HRD1* *ERAD* in suppressing *STING*-mediated innate immunity (Extended Data Fig. 10b).”

Minor Points

- In their discussion (line 76), authors should also discuss the well described autophagic mechanisms connected to *STING* degradation in addition to proteosomal mechanisms they currently discuss.

We thank the reviewer for this great comment. Indeed, the majority of literature has shown the role of endolysosomes in the degradation of *STING* following activation; and data on autophagy is relatively weak – in fact our data show Atg7-dependent autophagy is not involved in *STING* signaling under both basal and active states. We now also revised the Introduction, Results, and Discussion to clarify this point:

Introduction- Line 82-5: “In addition, activated *STING* can be sorted into acidified endolysosomes for degradation. This membrane trafficking process includes adaptor protein complex 1 (AP-1)-mediated delivery from the Golgi to endolysosomes via clathrin-coated transport vesicles, lysosomal membrane Niemann–Pick type C1 (NPC1) or p62/SQSTM1-dependent autophagy⁸⁻¹².”;

Results- Line 252-62: “To further demonstrate the importance of SEL1L-HRD1 ERAD in STING biology, we generated myeloid-specific *Atg7* knockout (*Atg7^{LYZ2}*) mice with defects in macroautophagy, another major intracellular proteolytic pathway¹⁵. Surprisingly, there was no difference in STING protein levels, its signaling pathway (Fig. 3g), and gene expression and secretion of IFN β (Fig. 3h-i) in *Atg7^{LYZ2}* vs. *Atg7^{fl/fl}* macrophages under both basal and active states, pointing to a dispensable role of ATG7-mediated macroautophagy in regulating STING activation. However, in line with previous reports^{8,9}, treatment with bafilomycin A1, a compound that inhibits lysosomal acidification and degradation^{6,7}, increased STING protein level in DMXAA-stimulated cells, while having no effect on STING protein level under basal state (Fig. 3j). Taken together, we conclude that, unlike active STING which are degraded in the endolysosomes, degradation of STING under basal state occurs in the ER and is mediated by SEL1L-HRD1 ERAD.”

Discussion- Line 429-31: “Unlike endolysosome-dependent degradation of active STING to help terminate its signaling⁸⁻¹² (Fig. 3j), SEL1L-HRD1 ERAD degrades naïve STING under the basal state to limits its activation potential.”

- In Figure 2a, the authors also find that TNF α is significantly increased following TLR2 agonism by Pam3, but fail to follow-up or discuss this observation. What is the significance of this finding? Is this the subject of a subsequent investigation?

We thank the reviewer for this great comment. We have confirmed this observation and, while interesting, its significance remains unclear. Indeed, the first and co-corresponding author Dr. Ji, now a Professor at Zhejiang University (China), is exploring this direction in his own laboratory.

- Extended Figure 2F – The authors provide evidence that CD8 T cells are increased, but did the authors look at CD4+ T cells to see if there was a difference between *Sel1L^{f/f}* and *Sel1L^{LYZ2}* mice?

We thank the reviewer for this great comment. We now have measured and quantitated the percent of CD4⁺ T cells in adipose tissue, which were not affected in HFD-fed *Sel1L*-deficient mice (Response Figure 13). These data are now shown in Extended Data Figure 2g-h in the revised manuscript.

Response Figure 13. Flow cytometric analysis of CD4⁺ T cells in the WAT stromal vascular cells after 20-week HFD, with quantitation shown in (b). n=6 or 4 mice. Values represent mean \pm s.e.m; n.s., no significant difference, by unpaired, two-tailed, Student's *t*-test.

- Figure 5a – Authors show that ERAD factors such as SEL1L interacts with STING. Should consider moving this to the beginning of the manuscript to highlight the significance of studying SEL1L.

We thank the reviewer for this great comment. We now have revised the flow of the story based on the suggestion.

- Figure 6b – in reference to the graph of fraction # and percent, comparing *Sel1L^{f/f}* and *Sel1L^{LYZ2}*, can the authors explain how STING mass was quantitated? This is not clearly explained in the methods.

We thank the reviewer for this great comment. We now have included the description of quantitation in the Methods: Following centrifugation at 58,000 rpm for 14.5 hr at 4 °C using an SW 60 Ti rotor (Beckman Coulter), 9 fractions were collected with the top fraction of the lowest density named as the fraction #1 and subsequently subjected to Western blot analysis under denaturing conditions. Band intensity of each fraction was quantitated and percent of protein in each fraction was calculated by dividing protein intensity in individual fraction by total protein intensity from all fractions.

References:

- 1 Arango Duque, G. & Descoteaux, A. Macrophage cytokines: involvement in immunity and infectious diseases. *Frontiers in immunology* **5**, 491 (2014). <https://doi.org/10.3389/fimmu.2014.00491>
- 2 Qi, L., Tsai, B. & Arvan, P. New Insights into the Physiological Role of Endoplasmic Reticulum-Associated Degradation. *Trends Cell Biol* **27**, 430-440 (2017). <https://doi.org/10.1016/j.tcb.2016.12.002>
- 3 Hwang, J. & Qi, L. Quality Control in the Endoplasmic Reticulum: Crosstalk between ERAD and UPR pathways. *Trends Biochem Sci* **43**, 593-605 (2018). <https://doi.org/10.1016/j.tibs.2018.06.005>
- 4 Bhattacharya, A. & Qi, L. ER-associated degradation in health and disease - from substrate to organism. *J Cell Sci* **132**, jcs232850 (2019). <https://doi.org/10.1242/jcs.232850>
- 5 Shrestha, N. *et al.* Integration of ER protein quality control mechanisms defines beta-cell function and ER architecture. *J Clin Invest* (2022). <https://doi.org/10.1172/JCI163584>
- 6 Tapper, H. & Sundler, R. Bafilomycin A1 inhibits lysosomal, phagosomal, and plasma membrane H(+)-ATPase and induces lysosomal enzyme secretion in macrophages. *J Cell Physiol* **163**, 137-144 (1995). <https://doi.org/10.1002/jcp.1041630116>
- 7 Mauvezin, C. & Neufeld, T. P. Bafilomycin A1 disrupts autophagic flux by inhibiting both V-ATPase-dependent acidification and Ca-P60A/SERCA-dependent autophagosome-lysosome fusion. *Autophagy* **11**, 1437-1438 (2015). <https://doi.org/10.1080/15548627.2015.1066957>
- 8 Gonugunta, V. K. *et al.* Trafficking-Mediated STING Degradation Requires Sorting to Acidified Endolysosomes and Can Be Targeted to Enhance Anti-tumor Response. *Cell reports* **21**, 3234-3242 (2017). <https://doi.org/10.1016/j.celrep.2017.11.061>
- 9 Chu, T. T. *et al.* Tonic prime-boost of STING signalling mediates Niemann-Pick disease type C. *Nature* **596**, 570-575 (2021). <https://doi.org/10.1038/s41586-021-03762-2>
- 10 Zhu, H., Zhang, R., Yi, L., Tang, Y. D. & Zheng, C. UNC93B1 attenuates the cGAS-STING signaling pathway by targeting STING for autophagy-lysosome degradation. *J Med Virol* **94**, 4490-4501 (2022). <https://doi.org/10.1002/jmv.27860>
- 11 Prabakaran, T. *et al.* Attenuation of cGAS-STING signaling is mediated by a p62/SQSTM1-dependent autophagy pathway activated by TBK1. *EMBO J* **37** (2018). <https://doi.org/10.15252/embi.201797858>
- 12 Liu, Y. *et al.* Clathrin-associated AP-1 controls termination of STING signalling. *Nature* **610**, 761-767 (2022). <https://doi.org/10.1038/s41586-022-05354-0>
- 13 Yao, X. *et al.* T-cell-specific Sel1L deletion exacerbates EAE by promoting Th1/Th17-cell differentiation. *Mol Immunol* **149**, 13-26 (2022). <https://doi.org/10.1016/j.molimm.2022.06.001>
- 14 Whitley, R. J. & Roizman, B. Herpes simplex virus infections. *Lancet* **357**, 1513-1518 (2001). [https://doi.org/10.1016/S0140-6736\(00\)04638-9](https://doi.org/10.1016/S0140-6736(00)04638-9)
- 15 Grumati, P., Dikic, I. & Stolz, A. ER-phagy at a glance. *J Cell Sci* **131** (2018). <https://doi.org/10.1242/jcs.217364>

Decision Letter, first revision:

Our ref: NCB-A48772A

14th December 2022

Dear Dr. Qi,

Thank you for submitting your revised manuscript "SEL1L-HRD1 endoplasmic reticulum-associated degradation controls STING-mediated innate immunity by limiting the size of activable STING pool" (NCB-A48772A). It has now been seen by the original referees and their comments are below. The reviewers find that the paper has improved in revision, and therefore we'll be happy in principle to publish it in Nature Cell Biology, pending minor revisions to satisfy the referees' final requests and to comply with our editorial and formatting guidelines.

As the current version of your manuscript is in a PDF format, please email us a copy of the file in an editable format (Microsoft Word or LaTeX)-- we can not proceed with PDFs at this stage.

Thank you again for your interest in Nature Cell Biology Please do not hesitate to contact me if you have any questions.

Sincerely,
Daryl

Daryl J.V. David, PhD

Senior Editor, Nature Cell Biology
Consulting Editor, Nature Communications
Nature Portfolio

Heidelberger Platz 3, 14197 Berlin, Germany
Email: daryl.david@nature.com
ORCID: <https://orcid.org/0000-0002-9253-4805>

Reviewer #1 (Remarks to the Author):

The revised manuscript has been greatly improved and my concerns have been addressed in full. This study provides important new findings on the regulation of STING by the HRD11/SEL1L complex and i support its publication in NCB.

Reviewer #2 (Remarks to the Author):

All of my points I had raised has been addressed in this revised manuscript. I have no other concerns.

Reviewer #3 (Remarks to the Author):

This manuscript focussed on an important area of innate immunity, namely control of STING pathway activation, via posttranslational regulation of protein turnover by the HRD/SEL1L complex. Given the importance of STING in host-defense to DNA viruses and anti-tumor immunity a better understanding of STING activation and regulation is needed. The model prpposed by the authors is supported by the data and the revision with extensive additional supporting data makes a strong case. The paper is well written and the findings are important and likely to be of broad interest to the field.

I have no further comments.

Decision Letter, final checks:

Our ref: NCB-A48772A

10th January 2023

Dear Dr. Qi,

Thank you for your patience as we've prepared the guidelines for final submission of your Nature Cell Biology manuscript, "SEL1L-HRD1 endoplasmic reticulum-associated degradation controls STING-mediated innate immunity by limiting the size of activable STING pool" (NCB-A48772A). Please carefully follow the step-by-step instructions provided in the attached file, and add a response in each row of the table to indicate the changes that you have made. Please also check and comment on any additional marked-up edits we have proposed within the text. Ensuring that each point is addressed will help to ensure that your revised manuscript can be swiftly handed over to our production team.

In recognition of the time and expertise our reviewers provide to Nature Cell Biology's editorial process, we would like to formally acknowledge their contribution to the external peer review of your manuscript entitled "SEL1L-HRD1 endoplasmic reticulum-associated degradation controls STING-

mediated innate immunity by limiting the size of activable STING pool". For those reviewers who give their assent, we will be publishing their names alongside the published article.

Nature Cell Biology offers a Transparent Peer Review option for new original research manuscripts submitted after December 1st, 2019. As part of this initiative, we encourage our authors to support increased transparency into the peer review process by agreeing to have the reviewer comments, author rebuttal letters, and editorial decision letters published as a Supplementary item. When you submit your final files please clearly state in your cover letter whether or not you would like to participate in this initiative. Please note that failure to state your preference will result in delays in accepting your manuscript for publication.

Cover suggestions

As you prepare your final files we encourage you to consider whether you have any images or illustrations that may be appropriate for use on the cover of Nature Cell Biology.

Nature Cell Biology has now transitioned to a unified Rights Collection system which will allow our Author Services team to quickly and easily collect the rights and permissions required to publish your work. Approximately 10 days after your paper is formally accepted, you will receive an email in providing you with a link to complete the grant of rights. If your paper is eligible for Open Access, our Author Services team will also be in touch regarding any additional information that may be required to arrange payment for your article.

Please note that *Nature Cell Biology* is a Transformative Journal (TJ). Authors may publish their research with us through the traditional subscription access route or make their paper immediately open access through payment of an article-processing charge (APC). Authors will not be required to make a final decision about access to their article until it has been accepted. Find out more about Transformative Journals

Authors may need to take specific actions to achieve compliance with funder and institutional open access mandates. If your research is supported by a funder that requires immediate open access (e.g. according to Plan S principles) then you should select the gold OA route, and we will direct you to the compliant route where possible. For authors selecting the subscription publication route, the journal's standard licensing terms will need to be accepted, including self-

archiving policies. Those licensing terms will supersede any other terms that the author or any third party may assert apply to any version of the manuscript.

[REDACTED]

Best regards,

Kendra Donahue
Staff
Nature Cell Biology

On behalf of

Daryl J.V. David, PhD

Senior Editor, Nature Cell Biology
Consulting Editor, Nature Communications
Nature Portfolio

Heidelberger Platz 3, 14197 Berlin, Germany
Email: daryl.david@nature.com
ORCID: <https://orcid.org/0000-0002-9253-4805>

Reviewer #1:

Remarks to the Author:

The revised manuscript has been greatly improved and my concerns have been addressed in full. This study provides important new findings on the regulation of STING by the HRD11/SEL1L complex and i support its publication in NCB.

Reviewer #2:

Remarks to the Author:

All of my points I had raised has been addressed in this revised manuscript. I have no other concerns.

Reviewer #3:

Remarks to the Author:

This manuscript focussed on an important area of innate immunity, namely control of STING pathway activation, via posttranslational regulation of protein turnover by the HRD/SEL1L complex. Given the importance of STING in host-defense to DNA viruses and anti-tumor immunity a better understanding of STING activation and regulation is needed. The model prposed by the authors is supported by the data and the revision with extensive additional supporting data makes a strong case. The paper is well written and the findings are important and likely to be of broad interest to the field.

I have no further comments.

Final Decision Letter:

Dear Dr Qi,

I am pleased to inform you that your manuscript, "SEL1L-HRD1 endoplasmic reticulum-associated degradation controls STING-mediated innate immunity by limiting the size of activable STING pool", has now been accepted for publication in Nature Cell Biology.

Please note that *Nature Cell Biology* is a Transformative Journal (TJ). Authors may publish their research with us through the traditional subscription access route or make their paper immediately open access through payment of an article-processing charge (APC). Authors will not be required to make a final decision about access to their article until it has been accepted. Find out more about Transformative Journals

If you have not already done so, we strongly recommend that you upload the step-by-step protocols used in this manuscript to the Protocol Exchange (www.nature.com/protocolexchange), an open online resource established by Nature Protocols that allows researchers to share their detailed experimental know-how. All uploaded protocols are made freely available, assigned DOIs for ease of citation and are fully searchable through nature.com. Protocols and Nature Portfolio journal papers in which they are used can be linked to one another, and this link is clearly and prominently visible in the online versions of both papers. Authors who performed the specific experiments can act as primary authors for the Protocol as they will be best placed to share the methodology details, but the Corresponding Author of the present research paper should be included as one of the authors. By uploading your Protocols to Protocol Exchange, you are enabling researchers to more readily reproduce or adapt the methodology you use, as well as increasing the visibility of your protocols and papers. You can also establish a dedicated page to collect your lab Protocols. Further information can be found at

www.nature.com/protocolexchange/about

With kind regards,

Daryl

Daryl Jason Verzosa David, PhD

Senior Editor, Nature Cell Biology
Nature Portfolio

Heidelberger Platz 3, 14197 Berlin, Germany
Email: daryl.david@nature.com
ORCID: <https://orcid.org/0000-0002-9253-4805>